# A direction-selective cortico-brainstem pathway adaptively modulates innate behaviors

Jiashu Liu[1,2], Yingtian He[1,2], Andreanne Lavoie[1,2], Guy Bouvier[3] & Bao-hua Liu [1,2] ✉

Sensory cortices modulate innate behaviors through corticofugal projections targeting phylogenetically-old brainstem nuclei. However, the principles behind the functional connectivity of these projections remain poorly understood. Here, we show that in mice visual cortical neurons projecting to the optic-tract and dorsal-terminal nuclei (NOT-DTN) possess distinct response properties and anatomical connectivity, supporting the adaption of an essential innate eye movement, the optokinetic reflex (OKR). We find that these corticofugal neurons are enriched in specific visual areas, and they prefer temporo-nasal visual motion, matching the direction bias of downstream NOT-DTN neurons. Remarkably, continuous OKR stimulation selectively enhances the activity of these temporo-nasally biased cortical neurons, which can efficiently promote OKR plasticity. Lastly, we demonstrate that silencing downstream NOT-DTN neurons, which project specifically to the inferior olive—a key structure in oculomotor plasticity, impairs the cortical modulation of OKR and OKR plasticity. Our results unveil a direction-selective cortico-brainstem pathway that adaptively modulates innate behaviors.

Layer 5 (L5) pyramidal neurons (PNs) in mammalian sensory cortices send widespread axon collaterals to phylogenetically older portions of the brain, including various brainstem nuclei that mediate fundamental innate behaviors[1–8]. The functions of these cortical descending outputs have been of long-standing interest to the field since their initial discovery. Projection-specific circuit perturbations have shown that these corticofugal projections can modulate the sensory responses in brainstem nuclei and contribute to the plasticity of corresponding innate motor behaviors[2,3,5,6,9–11], allowing animals to readily adapt to changing environments or challenging conditions. Despite the importance of the corticofugal projections in the plasticity of brainstem circuits and innate behaviors, the underlying circuit mechanisms are still poorly understood. The anatomical and functional characteristics of corticofugal projections are crucial in determining how they contribute to the adaptive changes in innate

behaviors. L5 PNs projecting to the brainstem (cortico-brainstem) do not overlap with L5 PNs projecting to other cortical areas (cortico-cortical) or to the striatum (cortico-striatal)[3,12,13]. Moreover, recent functional studies of the primary visual cortex (V1) revealed that cortico-brainstem PNs possess distinct visual response properties compared to cortico-cortical or cortico-striatal PNs[3,13,14]. Although these findings suggest that corticofugal projections feed unique sensory information to the brainstem, they do not answer whether the signals carried by each individual corticofugal channels to different brainstem nuclei are functionally specialized and match the properties of corresponding innate behaviors.

One behavioral model to address the above question is the optokinetic reflex (OKR), an involuntary eye movement that stabilizes the image on the retina as an animal's head or the visual surround move, and thus plays an essential role in vision[1,15]. This innate behavior

[1]Department of Biology, University of Toronto Mississauga, Mississauga, ON L5L 1C6, Canada. [2]Department of Cell and Systems Biology, University of Toronto, Toronto, ON M5S 3G5, Canada. [3]Université Paris-Saclay, CNRS, Institut des Neurosciences Paris-Saclay, 91400 Saclay, France. ✉e-mail: baohua.liu@utoronto.ca

is an attractive choice, because its neural underpinning is characterized by a distinctive visual response property. The OKR in the horizontal axis is mediated by the closely apposed optic tract and dorsal-terminal nuclei (NOT-DTN) in the brainstem[1,2,15], which receive from the contralateral eye direct retinal input uniquely encoding a temporo-nasal motion relative to this eye. Consistently, the activity of NOT-DTN is also highly biased towards the temporo-nasal visual motion presented to the contralateral eye[1,2,16–18], distinguishing the NOT-DTN from other brainstem nuclei that are innervated by the visual cortex[19–21]. Thus, this temporo-nasal direction bias is an appealing property of the OKR. In addition to the retinal input, the NOT-DTN is also innervated by the corticofugal projections from both the V1 and higher visual areas (HVAs)[1,2,22,23], where neurons respond to gratings or random dot patterns moving in various directions[24–26]. Interestingly, previous studies on monkeys reported that neurons in motion-sensitive areas MT and MST projecting to the NOT-DTN also preferentially respond to the temporo-nasal motion[27,28]. However, NOT-DTN-projecting neurons in other visual areas have yet to be systematically examined; neither known are the anatomy and functionality of the brainstem neurons downstream of the corticofugal projection. Thus, it remains an open question in the mouse to what extent the response properties of these visual corticofugal projections match those of their downstream targets in the NOT-DTN.

Another essential property of the OKR is its plasticity. It enables the amplitude of the eye movements to be adaptively modified through experience or contexts, thereby ensuring image stability throughout life[2,29–31]. For example, a prolonged exposure to OKR stimuli can lead to an increase in the OKR gain[2,30,31], a metric reporting the strength of the behavior. The OKR potentiation is a widely studied model to understand the circuit basis underlying the plasticity of innate behaviors. Previous studies have attributed it mainly to the synaptic and cellular plasticity in subcortical structures including the cerebellum and brainstem nuclei[29,30,32–34]. Nevertheless, a recent study showed that the corticofugal projection connecting the visual cortex to the NOT-DTN can also contribute to OKR potentiation by boosting the NOT-DTN activity[2]. However, how the signals in this descending pathway result in the plasticity of the NOT-DTN and subsequent changes in the OKR remains elusive. In theory, the enhanced NOT-DTN activity could result from several possible circuit, cellular, and synaptic mechanisms. Among them, an attractive mechanism is the elevated responsiveness of corticofugal neurons, since the visual cortex is well-known for its a life-long capacity of plasticity[35–37]. The visually evoked activity in the visual cortex can undergo dramatic changes, when animals learn visually guided behavioral tasks[38,39], or are exposed to visual experience[40–42]. If inheriting the cortical plasticity, the corticofugal projection to NOT-DTN could increase its visual activity to boost the cortical innervation to NOT-DTN and consequently potentiate the OKR. If this were the case, it would provide an insightful perspective on the role of corticofugal projections and expand our view of how the cortical plasticity, known for its role in learned behaviors, can directly contribute to the adaptive plasticity of innate behaviors.

Here, we combined virus-based circuit tracing with in vivo two-photon calcium imaging to examine the connectivity and the response properties of L5 PNs in the mouse visual cortex that project to the NOT-DTN. We found that the NOT-DTN-projecting PNs exhibited a prominent bias towards temporo-nasal visual motion, matching the functional properties of their projection target, the NOT-DTN. Interestingly, we discovered that following OKR potentiation induced by a drum grating oscillating bidirectionally along the azimuth, the activity of NOT-DTN-projecting PNs selectively increased only in the population that prefers the temporo-nasal direction. Furthermore, we found that the visual cortex innervated only one subpopulation of NOT-DTN neurons, which project specifically to the inferior olive (IO) in the brainstem, a structure pivotal in cerebellum-dependent OKR plasticity. This IO-projecting NOT-DTN population also preferred the

temporo-nasal motion, and it was critical for the cortical modulation of OKR and OKR plasticity. Lastly, with modeling we revealed that the temporo-nasal bias and direction-selective plasticity enabled NOT-DTN-projecting PNs to efficiently enhance their input to the NOT-DTN to support the OKR potentiation. Overall, our results provide compelling evidence that a direction-selective and plastic descending pathway links the visual cortex to brainstem, which specializes in adaptively modulating the OKR.

## Results

### NOT-DTN-projecting PNs share temporo-nasal bias with their target

Since the most prominent trait of NOT-DTN neurons is their response preference for the temporo-nasal visual motion[1,2,16–18,43,44], we examined whether the visual cortical neurons that project to the NOT-DTN (NOT-DTN-projecting PNs) have a similar direction preference to their projection target. We first performed extracellular recordings from the NOT-DTN in anesthetized mice with silicon probes, while presenting sinusoidal gratings drifting in eight directions to the animals' contralateral eye (Fig. 1a). The majority of isolated NOT-DTN units responded preferentially to visual stimuli moving in the temporo-nasal direction (Figs. 1b and d **open bars**), consistent with previous reports on response properties of NOT-DTN neurons[1,2,16,17,43]. Next, to evaluate the direction preference of NOT-DTN-projecting PNs, we performed two-photon calcium imaging in awake mice (Fig. 1a). We conditionally expressed GCaMP6s exclusively in NOT-DTN-projecting PNs by locally injecting a retrograde AAV-Cre virus (Retro-Cre) into the NOT-DTN of *Ai14* tdTomato reporter mice (see Methods; Supplementary Fig. 1a, b **left**). Using in vivo two-photon imaging, we observed a sparse population of fluorescence-labeled PNs exclusively in L5 of the ipsilateral visual cortex (0.64% of L5 neurons; Supplementary Fig. 1c left, d). We examined the calcium dynamics of these corticofugal neurons in response to the above sinusoidal gratings presented to the contralateral eye (see Methods; Fig. 1c). The NOT-DTN-projecting PNs were sensitive to moving stimuli. More than half of them, pooled from both V1 and HVAs, responded to drifting gratings of at least one direction (777 out of 1482 neurons, 52%; Supplementary Fig. 2a top). A direction selectivity index (DSI) was computed to evaluate the strength of direction selectivity. We found that 54% of the responsive neurons had DSIs of at least 0.33 (the maximum response is more than double of the response to the null direction, Supplementary Fig. 2a). Next, we determined the preferred direction of NOT-DTN-projecting PNs (Supplementary Fig. 2b, c). Strikingly, there were many more neurons that prefer the temporo-nasal direction than the ones preferring any of the other 7 directions (27% for temporo-nasal direction vs 8–14% for the other 7 directions; Fig. 1d **solid bars**). Overall, these results indicate that the NOT-DTN-projecting PNs share the same direction preference as their brainstem target.

Is the temporo-nasal direction preference a unique feature of the NOT-DTN-projecting PNs or a common feature among all brainstem-projecting PNs? To address this question, we injected a large volume of the Retro-Cre virus into the midbrain (see Methods), which covered several brain regions, including the NOT-DTN, the pretectal nuclei, the superior colliculus (SC), as well as the medio-caudal part of lateral posterior thalamic nucleus (LP) (Supplementary Fig. 1b right). In comparison to the focal injection into the NOT-DTN, a much denser L5 population (8.74% of L5 PNs; Supplementary Fig. 1c right) was labeled with GCaMP6s, and we referred to these neurons as midbrain-projecting PNs. This PN population consisted of a similar percentage of responsive neurons compared to the NOT-DTN-projecting population (2124 out of 3879 neurons, 55%; Supplementary Fig. 2a bottom). However, it did not present any temporo-nasal bias (13% for temporo-nasal direction vs 10–15% for the other 7 directions; Fig. 1e) as seen in the NOT-DTN-projecting PNs (Fig. 1d **solid bars**). Such difference between these two PN populations did not depend on the DSI

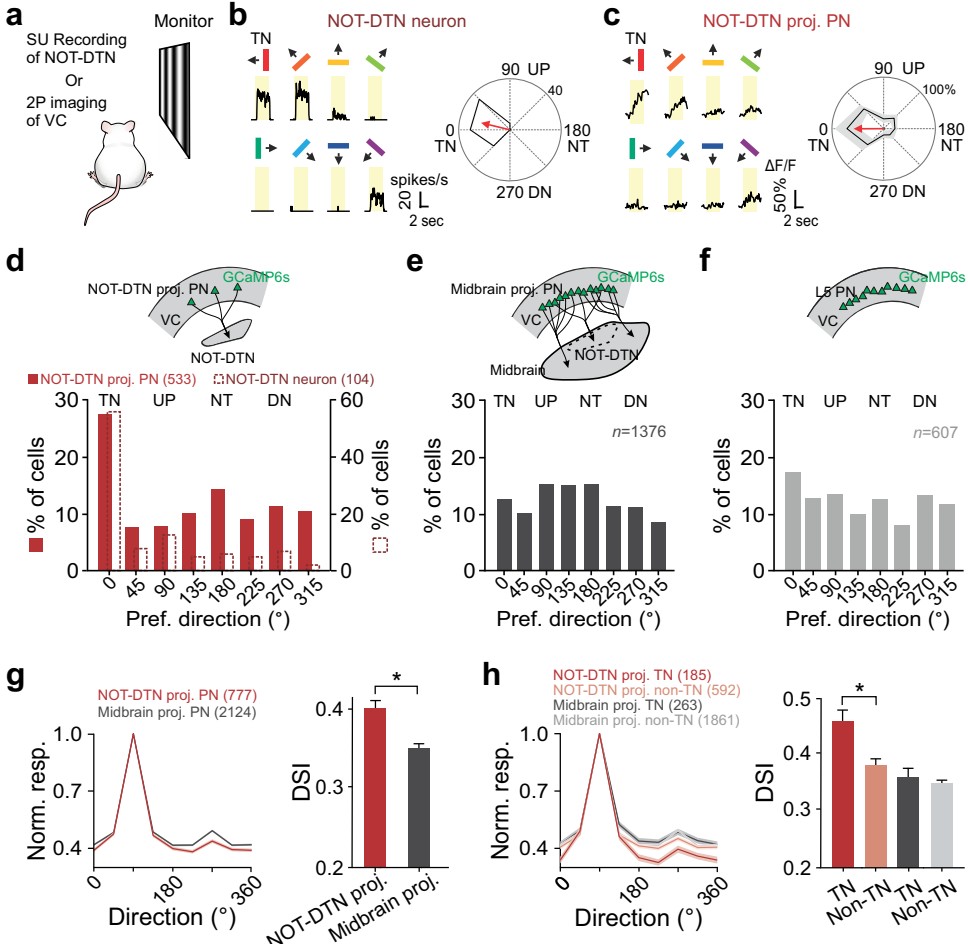

**Fig. 1 | NOT-DTN-projecting PNs in visual cortex share a bias towards temporo-nasal visual motion with the NOT-DTN. a** Schematic of experimental setup. Visual stimuli are presented to animal's eye contralateral to the recording site. SU, single unit; VC, visual cortex. **b** Single-unit activity of an example NOT-DTN neuron. Left, post-stimulus time histograms (PSTHs) of spiking responses evoked by gratings moving in various directions (*n* = 15 trials). Arrows and colors of bars indicate motion directions. Yellow shades indicate the durations of visual stimulation. Right, polar plot of the direction tuning curve. Thickness, s.e.m. Red arrow indicates the preferred direction. TN, temporo-nasal; NT, naso-temporal; UP, up; DN, down. **c** Two-photon calcium imaging of an example NOT-DTN-projecting L5 pyramidal neurons (PNs) in visual cortex. Data are presented as in **b**, except that the traces represent trial-averaged calcium responses (ΔF/F). **d** Histograms of the preferred direction of NOT-DTN-projecting PNs (solid bars, *n* = 15 mice, schematic on top) and NOT-DTN neurons (dashed open bars, *n* = 20 mice). Note that more neurons

prefer temporo-nasal direction in both populations (*P* < 1e-06, one-sided randomization test). The numbers in brackets, sample sizes. **e, f** Histograms of preferred direction of midbrain-projecting L5 PNs (**e**, *n* = 9 mice) or general L5 PNs (**f**, *n* = 6 mice). Data are presented as in **d**. **g** Left, averages of peak-aligned and normalized direction tuning curves of two L5 PN populations. Thickness, s.e.m. Note that the NOT-DTN-projecting PNs have greater difference between the responses of preferred and null directions than the midbrain-projecting PNs. Right, summary of their direction selectivity indices (DSI) (*, *P* = 5.5e-6; one-sided Wilcoxon rank-sum test). Data shown as mean ± s.e.m. **h** Direction turning curves (left) and summary of DSI (right) of neurons preferring temporo-nasal motion (TN) or non-temporo-nasal motions (non-TN). Data are presented as in **g**. *, *P* = 0.0002, one-sided Wilcoxon rank-sum test. The numbers in brackets of **g** and **h**, sample size. Schematics in **d, e** and **f** are adapted from Liu et al. (2016)[2].

(Supplementary Fig. 2d). Similarly, the general L5 PN population in the visual cortex (see Methods; Supplementary Fig. 1e, f), which includes cortico-cortical, cortico-striatal, and cortico-brainstem PNs, neither favored the temporo-nasal motion (Fig. 1f). These results demonstrate that the temporo-nasal bias is indeed a unique feature of the NOT-DTN-projecting PNs. Moreover, the NOT-DTN-projecting PNs possessed both a distinctly higher DSI and a greater percentage of direction-selective neurons, in comparison to the midbrain-projecting PNs (DSI = 0.40 ± 0.01 vs 0.35 ± 0.01; 54% vs 45% with DSI ≥ 0.33; Fig. 1g, Supplementary Fig. 2a, e). This higher degree of direction selectivity of the NOT-DTN-projecting PNs largely came from those preferring the temporo-nasal motion (DSI = 0.46 ± 0.02 vs 0.38 ± 0.01, temporo-nasal vs non-temporo-nasal; Fig. 1h). Taken together, these results indicate that the NOT-DTN-projecting PNs form a distinct pathway among cortico-brainstem projections, which sends functionally relevant information to the brainstem OKR circuit.

Lastly, we examined the SF tuning and TF tuning of the NOT-DTN-projecting and midbrain-projecting PNs by varying the SF or TF values of temporo-nasally moving gratings (Supplementary Fig. 3). As the NOT-DTN neurons preferentially responded to intermediate SFs and low TFs (Supplementary Fig. 3a, e)[2,17], a higher percentage of the NOT-DTN-projecting PNs preferred intermediate SFs of 0.08 and 0.16 cycles per degree (cpd) and a low TF of 0.5 Hz, compared to the midbrain-projecting PNs or general L5 PNs (Supplementary Fig. 3c–d, g–h). These results further support the idea that the projection from the visual cortex to the NOT-DTN tends to match its downstream target in response properties.

## Temporo-nasal bias of NOT-DTN-projecting PNs in V1 and posterior HVAs

Since individual visual cortical areas are functionally specialized[24,26,45–48], it is possible that only selective visual areas transmit neural signals

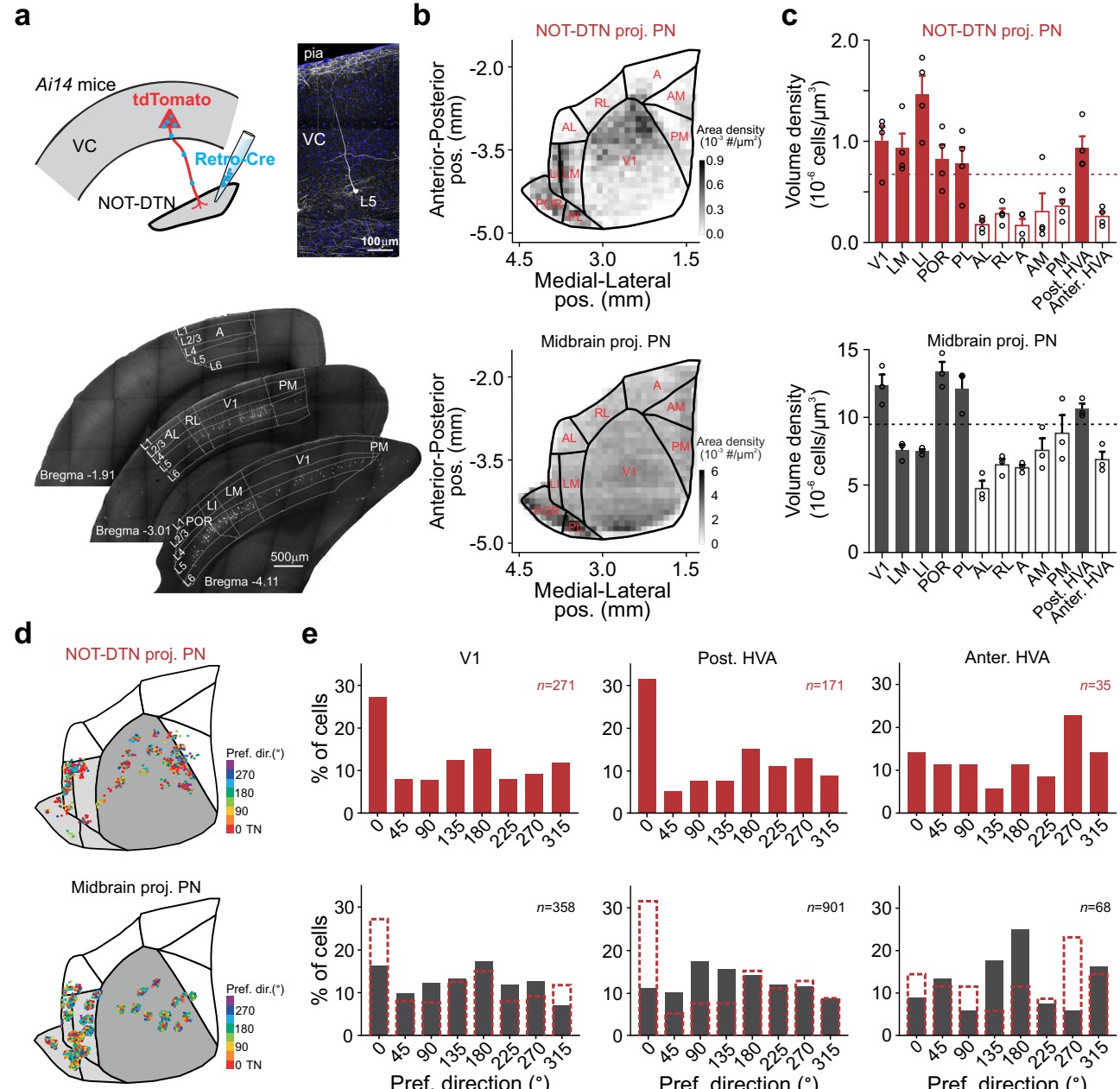

**Fig. 2 | NOT-DTN-projecting PNs in both V1 and posterior HVAs are biased towards temporo-nasal visual motion. a** Top left, schematic of experimental setup. Schematic is adapted from Liu et al. (2016)[2]. Top right, confocal image of an example NOT-DTN-projecting pyramidal neuron (PN). Blue, DAPI; white, tdTomato. VC, visual cortex. Bottom, spatial distribution of NOT-DTN-projecting PNs on three coronal slices containing visual cortex. Such pattern was observed in all repeats (4 mice). Boundaries between primary and higher visual areas are delineated according to Allen mouse brain atlas (atlas.brain-map.org). **b** Area density map of NOT-DTN-projecting PNs (top, $n = 4$ mice) or midbrain-projecting PNs (bottom, $n = 3$ mice). **c** Volumetric density of NOT-DTN-projecting PNs (top, $n = 4$ mice) or midbrain-projecting PNs (bottom, $n = 3$ mice) in individual visual areas. Dashed lines indicate average densities across the visual cortex. Solid Bars, V1 and posterior higher visual areas (HVAs); open bars, anterior HVAs. PM, posteromedial; AM,

anteromedial; A, anterior; RL, rostrolateral; AL, anterolateral; LI, laterointermediate; LM, lateromedial; POR, postrhinal; PL, posterolateral. Data shown as mean ± s.e.m. **d** Cortical locations of NOT-DTN-projecting PNs (top, $n = 477$ neurons) or midbrain-projecting PNs (bottom, $n = 1327$ neurons) recorded in 2-photon calcium imaging. Each dot is one neuron; colors encode preferred directions. Only neurons with DSI ≥ 0.1 are included. Dark grey, V1; light grey, posterior HVAs; white, anterior HVAs. **e** Histograms of preferred direction of the two corticofugal populations in different visual areas. Note that among NOT-DTN-projecting population (red), more neurons prefer temporo-nasal direction in V1 ($P < 1e-6$) and posterior HVAs ($P = 1e-6$), but not in anterior HVAs ($P = 0.45$). In contrast, midbrain-projecting population (black) has much fewer neurons with the bias towards the temporo-nasal direction than NOT-DTN-projecting population (V1: $P = 5e-04$; posterior HVAs: $P < 1e-06$). One-sided randomization test.

encoding temporo-nasal motion to the NOT-DTN. To explore this possibility, we first investigated which visual area(s) send descending projection to the NOT-DTN. After locally injecting the Retro-Cre virus in the NOT-DTN of *Ai14* mice (Fig. 2a **top**, Supplementary Fig. 1b left), on coronal slices we observed a sparse population of tdTomato-positive L5 PNs across both V1 and HVAs exclusively in the ipsilateral hemisphere

(Fig. 2a **bottom**, Supplementary Fig. 4a), consistent with the results from two-photon imaging (Supplementary Fig. 1c left, d). Interestingly, the density of these PNs was remarkably inhomogeneous throughout the visual cortex. For instance, we found two patches of visual areas where the NOT-DTN-projecting neurons were highly enriched: one patch was located at the anterior portion of V1 and the other one

occupied HVAs posterolateral to V1 (LI, LM, POR, and PL; referred to as posterior HVAs; Fig. 2b **top**, c **top solid bars**, Supplementary Fig. 4b, c solid bars). In comparison, these PNs appeared much sparser in the rest of visual cortical areas, i.e., the posterior portion of V1 and HVAs anteromedial or anterolateral to V1 (AL, RL, A, AM, and PM; referred to as anterior HVAs; Fig. 2b **top**, c **top open bars**, Supplementary Fig. 4b, c open bars). However, the midbrain-projecting PNs, which were labeled with a widespread injection of the Retro-Cre virus (Supplementary Fig. 1b right), were much more populated and more uniformly distributed in the visual cortex (average density=8.7e-06 vs 0.6e-06 cells/μm$^3$, coefficient of variation of density=0.32 vs 0.67; Fig. 2b **bottom**, c **bottom**). Thus, these results demonstrate that the corticofugal projection to the NOT-DTN primarily originates from the anterior V1 and posterior HVAs. Moreover, the area-specific enrichment of NOT-DTN-projecting PNs also suggests that they come from a distinct population, instead of random samples of the midbrain-projecting PNs.

The appearance of NOT-DTN-projecting PNs in two patches indicates that there are two streams of cortical innervation, one from the anterior V1 and the other from the posterior HVAs. Do both routes carry the same information about the temporo-nasal motion? To answer this question, we investigated the direction selectivity of the corticofugal neurons located in three areas: V1 (mostly anterior V1), posterior HVAs, and anterior HVAs (mainly areas PM and AL) (Fig. 2d, Supplementary Fig. 5a). In both V1 and posterior HVA, but not the anterior HVA, there were significantly more neurons within the NOT-DTN-projecting population that prefer the temporo-nasal motion (V1: 27% vs 8–15%; posterior HVAs: 32% vs 5–15%; anterior HVAs: 14% vs 5–23%; Fig. 2e **top**). As controls, we examined the midbrain-projecting PNs in V1 and posterior HVAs, and found much fewer neurons with a bias towards the temporo-nasal direction (V1: 16% vs 27%; posterior HVAs: 11% vs 32%; Fig. 2e **bottom**). Furthermore, in both V1 and posterior HVA groups, the NOT-DTN-projecting PNs possessed stronger direction selectivity (V1: $0.42 \pm 0.01$ vs $0.35 \pm 0.01$; posterior HVAs: $0.39 \pm 0.02$ vs $0.35 \pm 0.01$; Supplementary Fig. 5b) and a higher percentage of direction-selective neurons (V1: 55.0% vs 45.9%; posterior HVAs: 52.3% vs 45.1%; Supplementary Fig. 5c) than the midbrain-projecting PNs. Thus, these results indicate that the corticofugal projections from both V1 and posterior HVAs conduct the signal concerning temporo-nasal motion to the NOT-DTN.

## Posterior HVAs impact the OKR more strongly than anterior HVAs

The corticofugal projection to the NOT-DTN is the biological basis for the visual cortex to modulate the OKR[2]. The striking difference in the density of NOT-DTN-projecting PNs among HVAs made us wonder whether posterior HVAs have a stronger impact on the OKR behavior than anterior HVAs. To evaluate the OKR, a mouse was head-fixed at the center of a virtual drum displaying a vertical grating, and its right eye was monitored by infrared video-oculography[2] (see Methods; Fig. 3a). When the grating drifted horizontally in a sinusoidal trajectory (oscillation amplitude 5°), the animal's eye tracked the grating motion in an oscillatory manner, a signature of the OKR behavior (Fig. 3b). The strength of the behavior was quantified by calculating an OKR gain, which is the ratio between the amplitude of the eye trajectory and the amplitude of the grating trajectory. OKR gain was dependent on the SF of the grating, reaching a maximum at 0.16 cpd (Fig. 3c), as described previously[2].

To examine the impact of the visual cortex on the OKR, we optogenetically silenced its activity by photoactivating GABAergic inhibitory neurons which express Channelrhodopsin2 (ChR2), a light-sensitive cation channel (Fig. 3d **bottom**). Consistent with previous reports[2,49], bilaterally silencing the visual cortex led to a reduction in OKR gain, indicating a facilitatory effect of cortical input on the OKR (Fig. 3d **top left**). We referred to this percentage reduction as the cortical contribution to OKR gain. As controls, illuminating the visual

cortex of wild-type mice had no effect on OKR gain (Fig. 3d **top right**, Supplementary Fig. 6a). With this method we compared the impact of anterior HVAs and posterior HVAs on the OKR, by shining blue light on either one of these two areas in the same mice while covering the other one (see Methods; Fig. 3e **top**). Consistent with the density difference of NOT-DTN-projecting PNs in HVAs, silencing posterior HVAs decreased the OKR gain significantly more than silencing anterior HVAs ($8.7 \pm 1.2\%$ vs $5.0 \pm 1.0\%$; Fig. 3e, f, Supplementary Fig. 6b). This data demonstrates that posterior HVAs indeed contribute more to the OKR than anterior HVAs, indicating the functional specialization of HVAs in modulating the OKR. Are these HVAs involved in the plasticity of the OKR differently? We induced OKR potentiation by exposing head-fixed mice to a drum grating oscillating bidirectionally along the azimuth continuously for 45 min (see Methods, Fig. 3g), a classic experimental paradigm to study OKR plasticity[2,30,31]. Consistent with previous reports[2,30,31], continuous OKR stimulation significantly increased the OKR gain, but the exposure to a grey screen (blank stimulus) did not ($0.21 \pm 0.03$ vs $0 \pm 0.03$; Fig. 3g, h). Upon the induction of OKR potentiation, the contribution of posterior HVAs to OKR gain almost doubled ($7.9 \pm 1.9\%$ to $13.9 \pm 1.5\%$), but the contribution of anterior HVAs barely changed ($3.7 \pm 1.2\%$ to $5.2 \pm 1.4\%$, Fig. 3i, Supplementary Fig. 6c). As a result, now posterior HVAs contributed even more strongly to the OKR than anterior HVAs (Supplementary Fig. 6d). These results demonstrate the differential role of posterior and anterior HVAs in the OKR plasticity. Moreover, when the posterior HVAs were silenced, the amount of OKR potentiation on average reduced by 33% ($0.24 \pm 0.06$ to $0.15 \pm 0.06$, Supplementary Fig. 6e), indicating that a substantial portion of OKR potentiation may depend on the posterior HVAs.

## Direction-dependent cortical impact on the OKR

Since the NOT-DTN-projecting PNs have a unique temporo-nasal direction bias (Fig. 1d **solid bars**), next we wondered whether the effect of silencing the visual cortex on the OKR also depends on the direction of drum gratings. The OKR in mice is highly asymmetric[50]. When one eye was occluded (see Methods), only temporo-nasal motion with respect to the viewing eye (equivalently, ipsiversive to the contralateral NOT-DTN) effectively triggered the OKR behavior, but naso-temporal motion did not (norm. OKR gain: $0.05 \pm 0.03$, Supplementary Fig. 7a-c). This result indicates that when an oscillating grating is presented to a binocularly viewing mouse, each pair of the eye and the corresponding contralateral NOT-DTN alternately drive the OKR (Supplementary Fig. 7d left, e left). To estimate the direction-dependent cortical contribution to the OKR, we unilaterally silenced the left visual cortex with the forementioned optogenetic method (Fig. 3d) while presenting drum gratings moving either temporo-nasally or naso-temporally in reference to the right eye of binocularly viewing mice (ipsiversively or contraversively to the left visual cortex and NOT-DTN, Supplementary Fig. 7d left, e left). The cortical silencing evidently decreased the gain of the ipsiversive OKR behavior ($27.5 \pm 1.84\%$, Supplementary Fig. 7d right, f left), but to our surprise it instead enhanced the gain of the contraversive OKR ($-11.8 \pm 3.34\%$, Supplementary Fig. 7e right, f right). These results indicate that the visual cortex facilitates the OKR mediated by the NOT-DTN in the same hemisphere (Supplementary Fig. 7d left), but suppresses the behavior mediated by the NOT-DTN in the opposite hemisphere (Supplementary Fig. 7e left). The cortical suppression of contraversive OKR (Supplementary Fig. 7e right, f right) suggests that the weak activation of NOT-DTN-projecting PNs by naso-temporal motion (Fig. 1d) may innervate the non-operating NOT-DTN and in turn counteract the OKR mediated by the counterpart NOT-DTN. Lastly, we measured the cortical contribution to monocularly driven OKR evoked by the temporo-nasal motion (or ipsiversive), with (open-loop) or without (closed-loop) keeping the retinal slips at constant speeds (see Methods). Consistent with the effect of cortical silencing on ipsiversive binocular

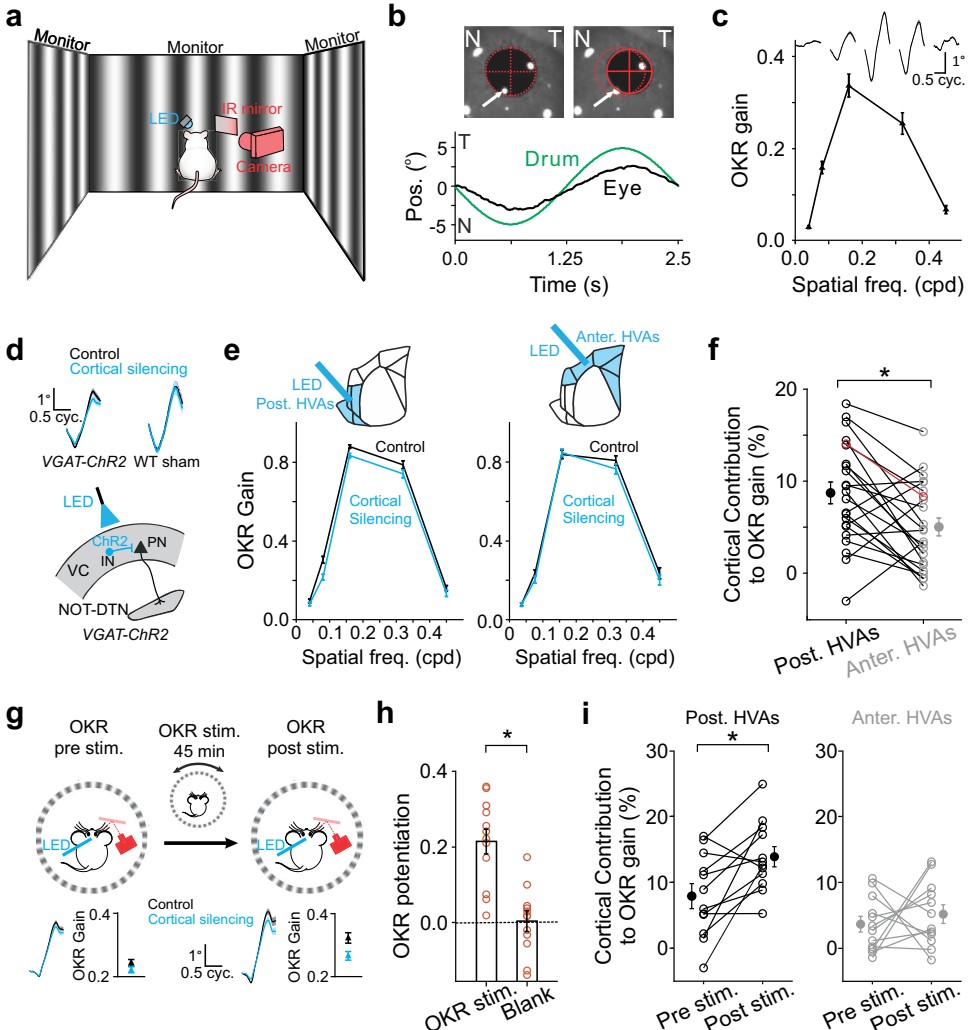

**Fig. 3 | Posterior HVAs impact the OKR more strongly than anterior HVAs.**
**a** Schematic of experimental setup. **b** Top, snapshots of nasal (N; left) and temporal (T; right) eye positions taken during OKR stimulation. Red ellipses, fitting to pupil profile. Red crosses, pupil centers. Arrows, corneal reflection of a reference LED. Bottom, cycle-averaged eye trajectory overlaid with drum trajectory. **c** Spatial frequency tuning curve of OKR gain from one animal. Top trace, cycle-averaged OKR trajectories (*n* = 192 cycles) elicited by gratings of various spatial frequencies. Thickness, s.e.m. **d** Top, cycle-averaged OKR trajectories (*n* = 240 cycles) elicited by moving gratings with (cortical silencing) or without blue LED (control). Thickness, s.e.m. Bottom, schematic of optogenetic silencing of visual cortex. IN, inhibitory neuron; VC, visual cortex; PN, pyramidal neuron. **e** Spatial frequency tuning curves of OKR gain from one mouse when its posterior (left) or anterior (right) higher visual areas (HVAs) are silenced (*n* = 40 trials). Blue and black curves, with and without cortical silencing, respectively. Top schematics, locations of cortical

silencing. **f** Summary of the contribution of posterior or anterior HVAs to OKR gain (*n* = 23 mice; *, *P* = 0.005, one-sided Wilcoxon signed-rank test). Red data points come from the animal in **e**. **g** OKR of one example mouse before (pre stim.) and after (post stim.) OKR potentiation induced by continuous horizontal OKR stimulation. Top, schematic of experimental design. Bottom, cycle-averaged OKR trajectories (Thickness, s.e.m.) and the corresponding OKR gains (*n* = 24 cycles). Blue and black, with and without cortical silencing. **h** Summary of OKR potentiation induced by continuous OKR stimulation (OKR stim., *n* = 12 mice, *P* = 2.4e-4) or a grey screen (Blank, *n* = 12 mice, *P* = 0.45). *, *P* = 4.9e-4, one-sided Wilcoxon signed-rank test. **i** Summary of cortical contribution of posterior HVAs (left, *, *P* = 0.005) and anterior HVAs (right, *P* = 0.29) to OKR gain before (pre stim.) and after (post stim.) continuous OKR stimulation (*n* = 12 mice), one-sided Wilcoxon signed-rank test. Data in **c**, **e**–**i** shown as mean ± s.e.m. Schematics in **d** and **g** are adapted from Liu et al. (2016)[2].

OKR (Supplementary Fig. 7d right, f left), silencing the left visual cortex reduced the gain of ipsiversive monocular OKR in both closed-loop and open-loop conditions (17.0 ± 1.7% vs 16.7 ± 2.8%, Supplementary Fig. 7g-i). This direction-dependent cortical impact on the OKR is aligned with previous lesioning studies on primates and cats[49,51] and supports the idea that the visual cortex modulates OKR behavior through its ipsilateral projection to the NOT-DTN (Fig. 2a, Supplementary Fig. 4a).

**Direction-selective plasticity in the NOT-DTN-projecting PNs**
The corticofugal projection can potentiate the amplitude of the OKR by boosting the activity of the NOT-DTN[2]. Next, we explored one of several potential mechanisms that account for the change in NOT-DTN

activity: the NOT-DTN-projecting PNs may become more responsive following OKR potentiation to supply stronger cortical input to the NOT-DTN. To test this, we measured the activity of corticofugal neurons evoked by four motion directions (tempo-nasal, naso-temporal, up, and down) as well as the OKR gain before and after OKR potentiation (Fig. 4a). Interestingly, along with the emergence of OKR potentiation (0.16 ± 0.04; Fig. 4b), the amplitudes of calcium responses of the NOT-DTN-projecting neurons were strengthened (Fig. 4c). To quantify the change in cortical activity, we used a plasticity index (PI, see Methods), with positive or negative values representing an increase or decrease of activities, respectively. After continuous OKR stimulation, 67.2% of the NOT-DTN-projecting PNs increased their responses to moving gratings (Supplementary Fig. 8a, b).

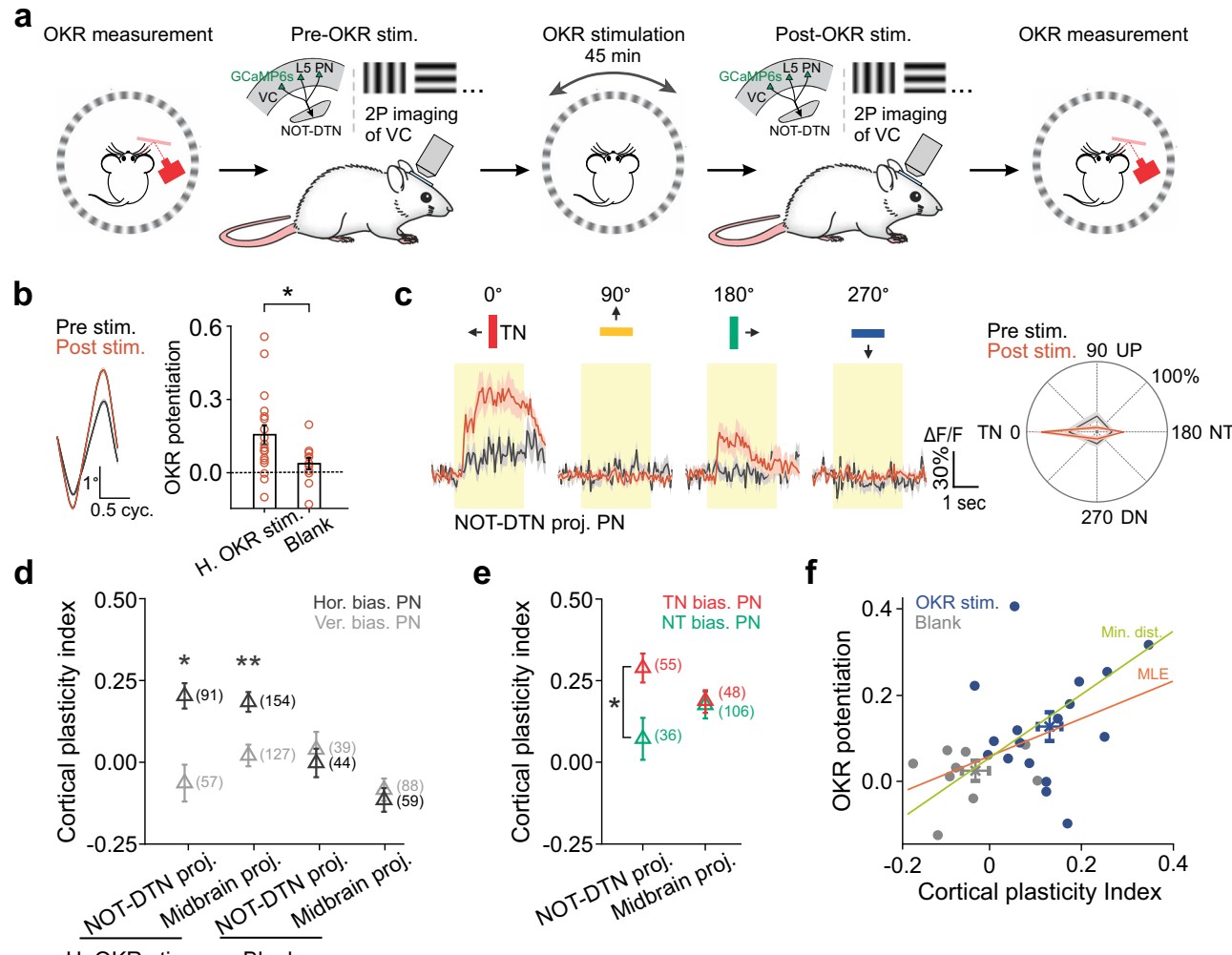

**Fig. 4 | Direction-selective plasticity of NOT-DTN-projecting PNs following OKR potentiation. a** Schematic of experimental design. VC, visual cortex; PN, pyramidal neuron. Schematic is adapted from Liu et al. (2016)[2]. **b** Left, cycle-averaged OKR trajectories from one example animal before (Pre stim.) and after (Post stim.) continuous horizontal OKR stimulation. Thickness, s.e.m. Right, summary of OKR potentiation after the exposure to continuous OKR stimulation (H. OKR stim., $n = 18$) or a grey screen (Blank, $n = 11$). *, $P = 0.01$, one-sided Wilcoxon rank-sum test. **c** Calcium responses (ΔF/F) of an example NOT-DTN-projecting PN before and after OKR potentiation. Left, averaged calcium dynamics ($n = 15$ trials) sorted according to the directions of moving gratings. Arrows and colors of bars indicate motion directions. Yellow shades indicate the duration of visual stimulation. Right, polar plot of direction tuning curves. Thickness, s.e.m. TN, temporo-nasal; NT, naso-temporal; UP, up; DN, down. **d** Summary of plasticity index of temporo-nasally evoked activity among corticofugal PNs that prefer the horizontal motions (Hor. bias. PN, black, 0° and 180°) or vertical motions (Ver. bias. PN, grey, 90° and 270°). The numbers in

brackets, sample sizes. $P = 1.2e-06$ (*) vs 0.83; $P = 1.3e-09$ (**) vs 0.37; horizontally vs vertically biased PNs; one-sided Wilcoxon signed-rank test. **e** Summary of plasticity index of temporo-nasally evoked activity among corticofugal PNs that prefer temporo-nasal (TN bias. PN, red, 0°) or naso-temporal directions (NT bias. PN, green, 180°). The numbers in brackets, sample sizes. $P = 0.003$ (*) vs 0.28; NOT-DTN-projecting vs midbrain-projecting; one-sided Wilcoxon rank-sum test. **f** Correlation between OKR potentiation and cortical plasticity index of calcium responses evoked by temporo-nasal direction. Each data point comes from one recording. Cortical plasticity index is averaged among all neurons from one recording. $n = 17$ (OKR stimulation) and 9 (Blank stimulation) recording sessions. Orange line, the best-fit line of linear regression using maximum likelihood estimation (MLE, $R^2 = 0.19$). Green line, the best-fit line of linear regression using the minimum vertical distance method (Min. dist., $R^2 = 0.06$). Blue and grey crosses, summary of OKR potentiation and cortical plasticity index. Data in **b**, **d**–**f** shown as mean ± s.e.m.

Next, we compared the plasticity index among cortical neurons with different preferred directions. Considering the relevance of temporo-nasal motion to OKR behavior, we focused on the activity evoked by this particular direction. Following OKR potentiation, the NOT-DTN-projecting neurons that preferred horizontal motions became more responsive (PI = 0.20 ± 0.05), but the ones that preferred vertical motions did not (PI = -0.06 ± 0.06; Fig. 4d, Supplementary Fig. 8c left). Interestingly, although a bidirectionally moving grating was used to induce OKR potentiation, the NOT-DTN-projecting PNs that preferred the temporo-nasal motion showed a much higher increase in the amplitude of calcium responses than those preferring the naso-temporal motion (referred to as direction-selective plasticity;

PI = 0.29 ± 0.04 vs 0.07 ± 0.06; Fig. 4e, Supplementary Fig. 8c left). In comparison to the NOT-DTN-projecting population, the plastic change in the activity of the midbrain-projecting population was significantly less (Supplementary Fig. 8b), and it showed no difference between temporo-nasally biased and naso-temporally biased neurons (PI = 0.19 ± 0.03 vs 0.18 ± 0.04; Fig. 4e, Supplementary Fig. 8c right). When a grey screen (blank stimulus) replaced drum gratings, cortical activity of neither NOT-DTN-projecting nor midbrain-projecting populations increased (Fig. 4d). A horizontal drum grating oscillating vertically for 45 min did not increase the gain of horizontal OKR, neither the activity of NOT-DTN-projecting cortical neurons in response to temporo-nasal motion (Supplementary Fig. 8d-f). Altogether, these

results demonstrate that prolonged horizontal OKR stimulation selectively enhances the cortical activity of NOT-DTN-projecting PNs that prefer the temporo-nasal direction, which will lead to a cell-type specific enhancement of cortical input to the NOT-DTN and as a result boost NOT-DTN activity. Considering the NOT-DTN-projecting PNs are necessary for OKR potentiation[2], the direction-selective plasticity suggests that these PNs form a functionally distinct population in promoting OKR potentiation.

Lastly, we analyzed the relationship between the strength of cortical plasticity and OKR potentiation. Remarkably, we found a positive correlation between the plasticity index of cortical activity evoked by the temporo-nasal motion and the OKR potentiation (Fig. 4f). Animals exposed to continuous OKR stimulation had larger values of both the plasticity index and OKR potentiation, than the control animals exposed to a grey screen (blank stimulus) (Fig. 4f **blue and grey crosses**). This result highlights the significance of the plastic change in corticofugal activity as a mechanism underlying OKR potentiation.

## The visual cortex innervates a specific NOT-DTN population

Although at the population level the NOT-DTN neurons show a preference for temporo-nasal visual motion, more than 46% of them still prefer non-temporo-nasal directions (Fig. 1d**, dashed open bars**). That led to the question of whether the NOT-DTN neurons that receive cortical innervation have the same temporo-nasal direction bias as their presynaptic partners—the NOT-DTN-projecting PNs in the visual cortex. To visualize these cortical-recipient NOT-DTN neurons, we injected an anterograde trans-synaptic Cre virus (Antero-Cre) into the visual cortex and a Cre-dependent tdTomato virus into the NOT-DTN, which conditionally expressed tdTomato in downstream NOT-DTN neurons (Fig. 5a). The NOT-DTN neurons project to multiple downstream structures in the brainstem, including the inferior olive (IO), SC, medial terminal nucleus (MTN), contralateral NOT-DTN and sparsely to the nucleus prepositus hypoglossi (NPH) (Supplementary Fig. 9a, b)[52–54]. And they can be divided into largely non-overlapping neuronal populations based on their projection targets[52]. Thus, to determine the identity of these cortical-recipient NOT-DTN neurons, we examined their axonal projections. Surprisingly, axons of these anterogradely labeled NOT-DTN neurons were manifested in the ipsilateral IO and sparsely seen in the ipsilateral NPH, but not in the other three downstream structures, SC, MTN, and the contralateral NOT-DTN (Fig. 5b). Indeed, we confirmed that the NOT-DTN neurons that project to the IO, labeled by an injection of Retro-Cre virus into the IO, sent only sparse collaterals to the ipsilateral NPH and did not project to other downstream targets (Supplementary Fig. 9c, d), indicating that they belong to a distinct population. Taken together, these results demonstrate that the descending projection from the visual cortex selectively impinges on the population of NOT-DTN neurons that project to the IO.

To further validate the connectivity of the visual corticofugal projections with IO-projecting NOT-DTN neurons, we performed rabies virus-based retrograde trans-synaptic tracing. Following conditional expression of TVA receptor and rabies glycoprotein in IO-projecting NOT-DTN neurons (starter cells, Fig. 5j, Supplementary Fig. 9e), the injection of EnvA-pseudotyped glycoprotein-deleted rabies virus (RV-GCaMP6s) in the NOT-DTN indeed trans-synaptically labeled L5 PNs in the ipsilateral visual cortex (referred to as NOT-DTN-innervating PNs, Fig. 5k **left**, Supplementary Fig. 9f). And these trans-synaptically labeled PNs were obviously more condensed in anterior V1 and posterior HVAs than the rest of visual cortex (Supplementary Fig. 9f-h). This finding agrees with the results from AAV-based retrograde tracing and anterograde trans-synaptic tracing (Figs. 2b, c, 5b). In addition, we noted that a small fraction of retinal ganglion cells (RGCs) in the contralateral eye were also labeled in the same animals (referred to as NOT-DTN-innervating RGCs, $187.4 \pm 28.4$ cells/mm²,

5.7% of the total RGC population in mice[55], Supplementary Fig. 9i). Interestingly, these RGCs were clearly more populated in the superior portion of the retina than the inferior portion ($229.6 \pm 40.4$ vs $148.3 \pm 19.0$ cells/mm², Supplementary Fig. 9j). Since both the superior retina and the anterior V1 monitor the visual space below the horizontal meridian, we conclude that retinotopically matched retinal and cortical inputs converge in IO-projecting NOT-DTN neurons. Next, to determine what types of RGCs innervate IO-projecting NOT-DTN neurons, we analyzed their dendritic morphology and dendritic stratification in the inner plexiform layer (IPL) of the retina (Supplementary Fig. 9k). NOT-DTN-innervating RGCs had relatively small dendritic fields ($200.1 \pm 7.7$ μm), and their dendrites highly branched; most of these RGCs (74%) had their dendritic arbors stratifying in both ON and OFF sublaminar layers of the IPL, while some (26%) had dendritic stratification mainly in the ON sublamina (Supplementary Fig. 9k-m). These morphological characteristics suggest that they are ON-OFF or ON direction-selective RGCs, which is consistent with a recent report that both ON direction-selective RGCs and ON-OFF direction-selective RGCs in mice project to the NOT-DTN[56].

Next, we examined the direction preference of IO-projecting NOT-DTN neurons. We combined single unit recording and optogenetic antidromic stimulation to identify the IO-projecting NOT-DTN neurons that expressed Chronos, a ChR2 variant (see Methods; Fig. 5c). Brief pulses (5 ms duration) of blue light delivered on top of the IO elicited spikes in axons coming from the NOT-DTN (Fig. 5c-f). These antidromic spikes recorded in the NOT-DTN were characterized by a small jitter ($0.40 \pm 0.05$ ms) and short onset latency ($5.52 \pm 0.62$ ms; Fig. 5d, e), which allowed us to unequivocally identify IO-projecting NOT-DTN neurons. Remarkably, all IO-projecting neurons (100%, $n = 22$) responded robustly to sinusoidal gratings moving in the temporo-nasal direction (0°), but were minimally active or even suppressed when the gratings moved in the opposite direction (180°; Fig. 5f-h). In contrast, among the NOT-DTN neurons that were not antidromically stimulated (presumably non-IO-projecting neurons), only 40.6% preferred the temporo-nasal direction (Fig. 5g). Moreover, in comparison to non-IO-projecting neurons, the IO-projecting ones had a much stronger direction bias (DSI = $0.95 \pm 0.03$ vs $0.57 \pm 0.03$; Fig. 5i), and they were more likely tuned to intermediate SF values (91.7% vs 58.0% preferring 0.08 and 0.16 cpd, Supplementary Fig. 10).

The NOT-DTN-projecting PNs had much less temporo-nasal bias than the NOT-DTN neurons (Fig. 1d) and the subpopulation, IO-projecting NOT-DTN neurons (Fig. 5g). Such difference may result from the possibility that the retrograde AAV virus may infect axons of passage that do not innervate the NOT-DTN[57]; this technical limitation could contaminate the dataset of NOT-DTN-projecting PNs with midbrain-projecting PNs and thus result in an underestimation of direction bias in NOT-DTN-projecting population. To eliminate this ambiguity, we used the same rabies virus-based trans-synaptic tracing method to express GCaMP6s in the visual cortical neurons that innervate IO-projecting NOT-DTN neurons (NOT-DTN-innervating PNs, Fig. 5j). Two-photon calcium imaging revealed that these PNs had a similar degree of temporo-nasal direction bias (29.4% for temporo-nasal direction vs 6.9–15% for the other 7 directions; Fig. 5k, l) and direction selectivity ($0.38 \pm 0.02$; Fig. 5m) to the NOT-DTN-projecting PNs. Taken together, these above results (Figs. 5g, l, 1d) indicate that the IO-projecting NOT-DTN neurons share similar visual feature selectivity with their cortical input, revealing a functionally specific disynaptic pathway that connects the visual cortex to the IO in the brainstem.

## The NOT-DTN to IO projection is required for cortical modulation of OKR

The corticofugal projection to the NOT-DTN is necessary for the visual cortex to influence the OKR behavior[2]. As this projection selectively innervates the IO-projecting NOT-DTN neurons (Fig. 5b), we

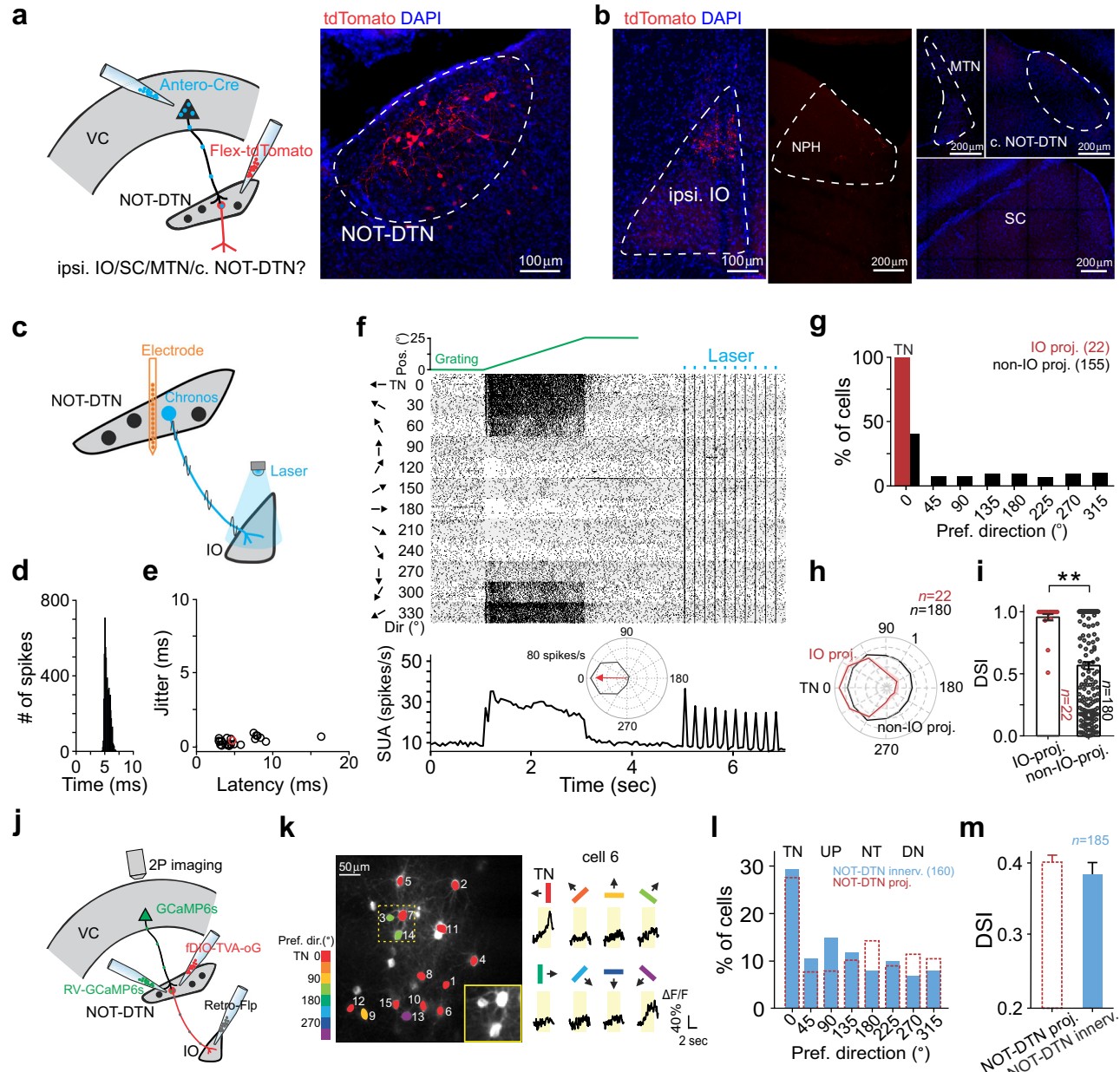

**Fig. 5 | Cortical-recipient NOT-DTN neurons exhibit a bias towards temporo-nasal visual motion. a** Left, schematic of labeling NOT-DTN neurons that receive cortical innervation. VC, visual cortex. Right, coronal slice of the NOT-DTN. **b** Coronal slices of brainstem structures downstream of NOT-DTN (*n* = 1 mouse). Ipsi. IO, ipsilateral inferior olive; NPH, nucleus prepositus hypoglossi; MTN, medial terminal nucleus; c. NOT-DTN, contralateral NOT-DTN; SC, superior colliculus. **c**–**i** Single-unit recording from IO-projecting NOT-DTN neurons. **c** Schematic of experimental setup. **d** Post-stimulus time histogram (PSTH) of optogenetically generated antidromic spikes from an example neuron. **e** Summary of onset latency and jitter of antidromic spikes (*n* = 24 neurons). Red circle, neuron in **d**. **f** Direction tuning of an example neuron. Top, trajectory of moving grating (green) and timing of laser pulses (blue). Middle, raster plots of single-unit activity. Arrows and numbers, directions of moving gratings. Bottom, PSTH of the same neuron. Inset, direction tuning curve. Red arrow, preferred direction. **g** Histogram of the preferred direction of IO-projecting or non-IO-projecting NOT-DTN neurons with DSI ≥ 0.1. Note that more IO-projecting NOT-DTN neurons prefer temporo-nasal

(TN) direction (*P* < 1e-06, one-sided randomization test). Numbers in brackets, sample size. **h**, **i** Population averaged normalized curves of direction tuning (**h**) and summary of DSI (**i**) of two NOT-DTN populations. **\*\***, *P* = 7.5e-6, one-sided Wilcoxon rank-sum test. **j**–**m** Two-photon calcium imaging of NOT-DTN-innervating pyramidal neurons (PNs). **j** Schematic of experimental setup. **k** Left, example two-photon image. Colors, preferred directions of neurons with DSI ≥ 0.1. Inset, higher magnification of the enclosed area. Right, trial-averaged calcium responses (ΔF/F, *n* = 15 trials) of an example neuron. Arrows and colors of bars, motion directions. Yellow shades, durations of visual stimulation. **l** Histogram of the preferred direction (solid blue bars, *n* = 12 mice). Note that more neurons prefer temporo-nasal direction (*P* < 1e-06, one-sided randomization test). Numbers in brackets, sample size. For the comparison purpose, data of the NOT-DTN-projecting PNs from Fig. 1d is superimposed (red dashed bars). **m** Summary of DSI. For the comparison purpose, data of the NOT-DTN-projecting PNs from Fig. 1g is superimposed. Data in **f**, **h**, **i** and **m** shown as mean ± s.e.m. Schematics in **a**, **c** and **j** are adapted from Liu et al. (2016)[2].

hypothesized that this specific NOT-DTN population is responsible for the cortical influence on the OKR. To test this, we used a similar intersectional strategy aforementioned (Supplementary Fig. 1a) to conditionally express hM4Di, a chemogenetic silencer, in the

IO-projecting NOT-DTN neurons in both hemispheres (see Methods; Fig. 6a). We confirmed the effectiveness of hM4Di in chemogenetically silencing the IO-projecting NOT-DTN neurons by staining c-Fos, an immediate early gene activated in response to neuronal activity. The

intraperitoneal injection of clozapine N-oxide (CNO), an agonist of HM4Di, effectively suppressed the c-Fos activity in IO-projecting NOT-DTN neurons evoked by 60 min of intermittent OKR stimulation, compared to saline injection (0.29e4 ± 0.01e4 vs 0.58e4 ± 0.03e4; Supplementary Fig. 11a, b).

We examined the cortical contribution to OKR gain by optogenetically silencing the visual cortex while chemogenetically suppressing the activity of IO-projecting NOT-DTN neurons (Fig. 6a). When saline was injected, cortical silencing reduced the OKR gain significantly across various SF values (10.6 ± 3.03%; Fig. 6b-c **saline**, Supplementary Fig. 11c), consistent with our previous report[2]. Crucially, when CNO was instead injected to suppress the activity of IO-projecting NOT-DTN neurons, cortical silencing no longer affected the OKR gain (-0.6 ± 2.7%; Fig. 6b-c **CNO**, Supplementary Fig. 11c). As controls, when tdTomato was expressed in the IO-projecting NOT-DTN neurons (Supplementary Fig. 9c), the administration of CNO had no influence on the cortical contribution to OKR gain (13.6 ± 2.1% vs 16 ± 2.3%, saline vs CNO; Fig. 6c **right**). These results indicate that the IO-projecting NOT-DTN neurons are required for the visual cortex to modulate OKR behavior. Consistently, optogenetically silencing the IO-projecting NOT-DTN neurons in the left hemisphere, which conditionally expressed inhibitory opsin ArchT (Fig. 6d, Supplementary Fig. 11d-e), reduced the gain of ipsiversive OKR (temporo-nasal), but not the gain of contraversive OKR (naso-temporal, 14.4 ± 3.5% vs -0.07 ± 1.5%, Fig. 6d-f). These results resemble the effect of silencing the left visual cortex (Supplementary Fig. 7d-f). Thus, the IO-projecting NOT-DTN population is the hub through which the ipsilateral visual cortex directly innervates the brainstem OKR circuit and consequently exerts its impact on the OKR behavior.

Lastly, since the NOT-DTN to IO projection is downstream of the retinal and cortical inputs, both of which are essential for OKR plasticity, we examined the role of this brainstem pathway in OKR potentiation. We applied the same chemogenetic method to suppress the activity of IO-projecting NOT-DTN neurons during the induction of OKR plasticity (Fig. 6g). Disrupting this pathway by CNO injection effectively prevented OKR potentiation, but saline injection did not (-0.01 ± 0.04 vs 0.33 ± 0.06; Fig. 6h,i), which is consistent with the blockage of OKR plasticity by lesioning the IO[30]. Therefore, the NOT-DTN to IO projection pathway is indispensable for the OKR potentiation.

## Temporo-nasal bias and direction-selective plasticity promote cortical innervation

So far, we showed that the NOT-DTN-projecting PNs are characterized by temporo-nasal bias and direction-selective cortical plasticity. To understand the roles of these functional properties in the cortical innervation of the NOT-DTN, we built a firing-rate-based feedforward model in which the NOT-DTN as a single node receives convergent inputs from a collection of PNs in the visual cortex (Fig. 7a). Model parameters such as the proportion of cortical neurons that are biased towards one particular direction (Fig. 1d **solid bars**), and their firing rates before and after OKR potentiation (Supplementary Fig. 12a), were estimated from our experimental data (see Methods). The simulation showed that upon cortical plasticity the synaptic input provided by the NOT-DTN-projecting PNs increased much more than that provided by the midbrain-projecting PNs (121.8 versus 62.7; Fig. 7b, c **top left**). This result indicates that the NOT-DTN-projecting population is more efficient in enhancing cortical drive to the NOT-DTN than the midbrain-projecting population, to support the OKR potentiation.

Next, we asked what circuit features matter to the transformation from cortical activity into synaptic input to the NOT-DTN. We varied three circuit features of the NOT-DTN-projecting PNs (temporo-nasal bias, strength of direction selectivity, and direction-selective cortical plasticity) one at a time. First, we replaced the distribution of preferred direction in the model with the non-temporo-nasal bias from the

midbrain-projecting PNs (Figs. 1e, 7c **top right**). Second, we used the responses of midbrain-projecting PNs (Fig. 7c **bottom right**, Supplementary Fig. 12a bottom), which have less direction selectivity than the NOT-DTN-projecting PNs (Fig. 1g). Lastly, we disrupted the direction-selective cortical plasticity of NOT-DTN-projecting PNs in three ways: (1) equalize plasticity strengths among the PN populations biased towards different directions (uniform plasticity, Fig. 7c **bottom left**); (2) swap the plasticity strength of temporo-nasally biased PNs with those preferring other directions (Supplementary Fig. 12b); (3) use the plasticity strength of midbrain-projecting PNs instead (Supplementary Fig. 12c). In all above scenarios of perturbations, the amount of extra cortical input generated by the cortical plasticity dramatically reduced. Thus, temporo-nasal direction bias, direction selectivity, and direction-selective plasticity are all required for the efficient integration of cortical input.

Do the above three circuit features interact with each other or work independently in the integration of cortical input? To address this question, we systematically varied the direction bias profile of the NOT-DTN-projecting population (Fig. 7d **top schematic**). When we included more neurons with a temporo-nasal bias, extra cortical drive upon cortical plasticity increased monotonically (Fig. 7d **magenta curve**). Interestingly, it rose rapidly when the fraction of temporo-nasally preferring neurons was low, and reached 85% of the maximum (Fig. 7d **horizontal dashed line**) at the actual fraction in the NOT-DTN-projecting population (Fig. 7d **magenta vertical line**). This result indicates that a moderate amount of temporo-nasal bias is enough for efficient cortical integration. Next, to examine the interaction between the strength of direction selectivity and temporo-nasal bias, we substituted the firing rate of cortical neurons with the responses of midbrain-projecting PNs. Consequently, the new curve (Fig. 7d **black curve**) reached the maximum value more slowly, indicating that the strength of direction selectivity affects how cortical input depends on temporo-nasal bias. Furthermore, to examine the interaction between direction-selective plasticity and temporo-nasal bias, we tweaked the original model by using uniform plasticity. The resulting flat curve (Fig. 7d **grey curve**) indicates that the specificity of cortical plasticity indeed dramatically impacts the relationship between cortical input and temporo-nasal bias, highlighting the functional importance of differential plasticity. Lastly, we explored the interaction between the strength of direction selectivity and direction-selective cortical plasticity. The increment of cortical input rose linearly (Fig. 7e **magenta curve**) with the plasticity strength of temporo-nasally preferring PNs. Interestingly, the midbrain-projecting PNs, which have less direction selectivity, gave rise to a shallower slope than the NOT-DTN-projecting PNs (Fig. 7e **black curve**). Collectively, these modeling results demonstrate that direction bias profile, strength of direction selectivity, and direction-selective cortical plasticity interact with each other, synergistically facilitating the integration of cortical input to the NOT-DTN.

## Discussion

To understand the functional organization of the cortico-brainstem projections, we examined the response properties and connectivity of NOT-DTN-projecting PNs using two-photon calcium imaging and virus-based circuit tracing. Our results revealed unprecedented specificities in this corticofugal population in comparison to the general midbrain-projecting population. Functionally, the NOT-DTN-projecting PNs are biased to temporo-nasal visual motion, matching the response properties of the NOT-DTN neurons they innervate (Fig. 1 & 5). In addition, the activities of these temporo-nasally biased corticofugal neurons are selectively enhanced upon the induction of OKR potentiation (Fig. 4e). Anatomically, the NOT-DTN-projecting PNs appear to be clustered in the anterior V1 and posterior HVAs (Fig. 2b, c), and they specifically innervate the NOT-DTN neurons that project to the IO (Fig. 5b). Altogether, we uncovered a visual pathway which transmits behaviorally

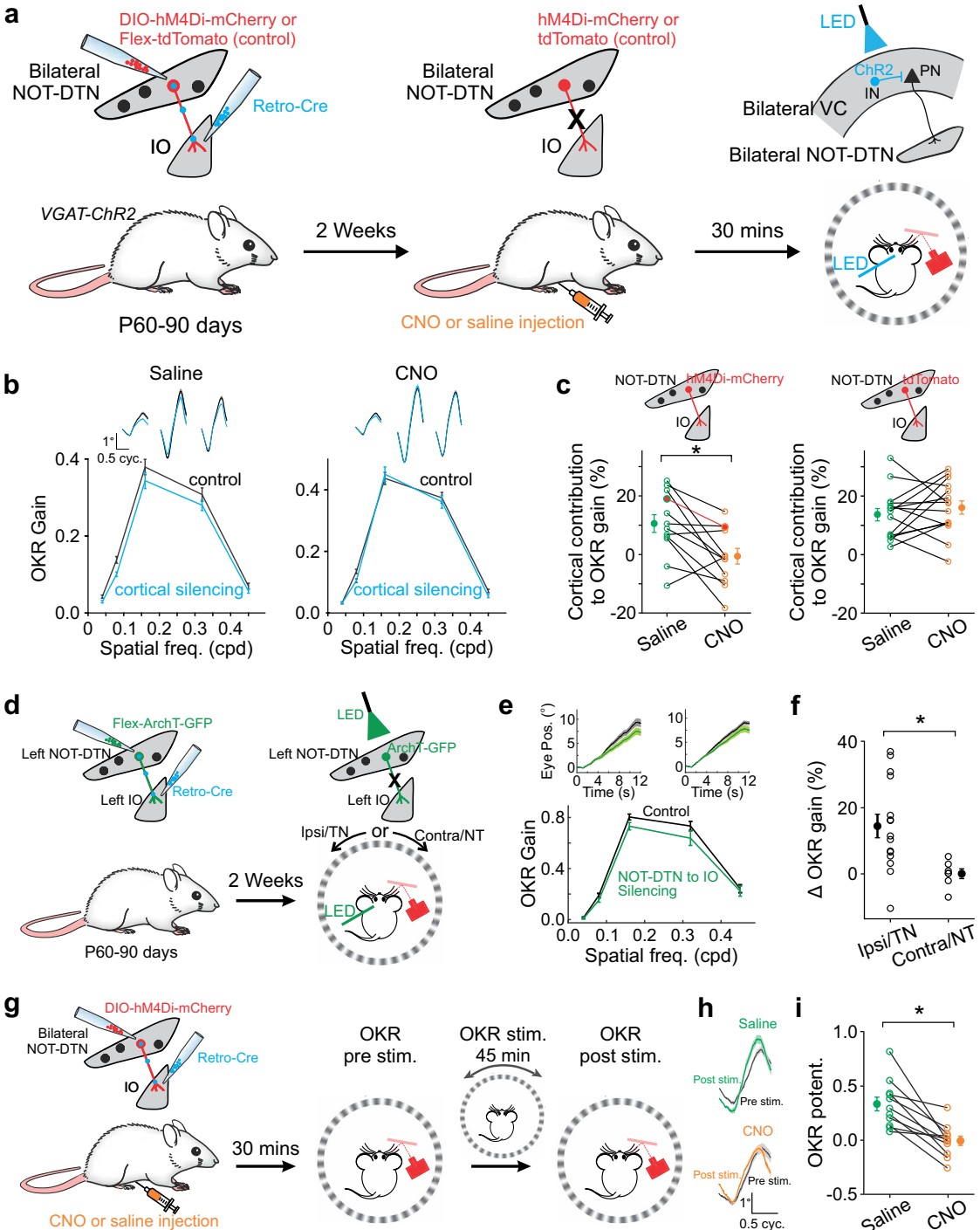

**Fig. 6 | IO-projecting NOT-DTN neurons are necessary for the cortical modulation of OKR. a–c** Cortical impact on baseline OKR requires the NOT-DTN to IO projection. **a** Schematic of experimental design. IO, inferior olive; VC, visual cortex; IN, inhibitory neuron; PN, pyramidal neuron. **b** Example animal. Spatial frequency (SF) tuning curves of OKR gain with intact (saline) or silenced (CNO) IO-projecting NOT-DTN neurons. Blue and black curves, with and without cortical silencing, respectively. Top traces, cycle-averaged OKR trajectories ($n = 48$ cycles). Thickness, s.e.m. **c** Summary of cortical contribution to OKR gain after saline or CNO injection. hM4Di-mCherry (left, $n = 12$ mice, *, $P = 0.006$) or tdTomato (right, $n = 15$ mice, $P = 0.82$) are expressed in IO-projecting NOT-DTN neurons (averaged over all tested SFs). $P = 0.006$ (saline) and 0.66 (CNO) for the HM4Di group; $P = 3e\text{-}05$ (saline) and 6e-05 (CNO) for the tdTomato control group. All statistic tests in **c**, one-sided Wilcoxon signed-rank test. Red, the example in **b**. **d–f** The NOT-DTN to IO projection modulates OKR behavior. **d** Schematic of optogenetically silencing IO-projecting NOT-DTN neurons in the left hemisphere. **e** Example animal. Top traces, averaged OKR trajectories ($n = 24$ trials) evoked by ipsiversively moving drum gratings of SF 0.08 cpd (left) or 0.32 cpd (right). Note that the nystagmus has been removed. Thickness, s.e.m. Bottom, SF tuning curves of OKR gain. Green and black curves, with and without optogenetically silencing, respectively. **f** Summary of the percentage change in the gain of OKR behavior evoked by ipsiversive (Ipsi/TN, $n = 15$, $P = 6e\text{-}4$) or contraversive (Contra/NT, $n = 7$, $P = 0.34$) motions when silencing the NOT-DTN to IO pathway (one-sided Wilcoxon signed-rank test). *, $P = 0.002$, one-sided Wilcoxon rank-sum test. **g–i** The NOT-DTN to IO projection is required for OKR potentiation. **g** Schematic of experimental design. **h** Example animal. Cycle-averaged OKR trajectories ($n = 40$ cycles) before (black, pre stim.) and after (colored, post stim.) inducing OKR potentiation. Thickness, s.e.m. **i** Summary of OKR potentiation with intact (saline, $P = 4.9e\text{-}4$) or silenced (CNO, $P = 0.58$) the NOT-DTN to IO projection ($n = 11$ mice). *, $P = 4.9e\text{-}4$. All statistic tests in **i**, one-sided Wilcoxon signed-rank test. Data in **b–c**, **e–f**, **i** shown as mean ± s.e.m. Schematics in **a**, **c**, **d**, and **g** are adapted from Liu et al. (2016)[2].

relevant motion signals from selective visual areas to the brainstem to promote the plasticity of OKR behavior.

A long-standing effort in the field has been to understand the logics of anatomical connectivity and response properties of the long-range projections along the visual pathways. It is well-documented that in the retinofugal, thalamocortical, or cortico-cortical pathways, the projections to different downstream targets carry distinct types of visual information, supporting the existence of anatomically and functionally separate channels or 'labeled lines'. Our results highlight the exquisite functional specificity of output channels from the visual cortex to the brainstem, suggesting a precise 'division of labor' among different axonal projections within the cortico-brainstem pathway. Brainstem nuclei innervated by the visual cortex have various response properties and are involved in different visual functions[1,19,21,58–61]. The unique temporo-nasal bias of NOT-DTN-projecting PNs (Fig. 1) suggests that cortico-brainstem projections to different downstream targets may follow the same principle of functional connectivity: the response properties of target-specific cortico-brainstem projections match those of the brainstem circuits they innervate. Thus, each cortico-brainstem projection as a separate channel or 'labeled line' may send distinct behaviorally relevant signals to the corresponding brainstem nucleus. This functional specificity provides an efficient solution for the visual cortex to adaptively modulate innate behaviors.

Our data show that the NOT-DTN-projecting PNs have less degree of temporo-nasal bias than their postsynaptic targets, NOT-DTN neurons (Fig. 1d). It is unlikely that this difference can be fully accounted for by the nonspecific infection of axons of passage that innervate other structures, but not NOT-DTN, since trans-synaptically labeled PNs that synapse on the IO-projecting NOT-DTN neurons also have a substantial portion of non-temporo-nasally biased neurons (Fig. 5l). Moreover, such difference between NOT-DTN-projecting PNs and NOT-DTN neurons was also uncovered in primates[17,18,27,28]. Thus, the differential level of temporo-nasal direction preference between NOT-DTN neurons and their cortical input may be conserved in mammalian species. What is the functional role of those non-temporo-nasally biased PNs? One potential function is to impact the contraversive OKR. The OKR is driven by the left and right NOT-DTNs in a push-pull manner; that is to say, the difference in activity between the operating (ipsiversive) and non-operating (contraversive) NOT-DTNs determines the strength of OKR[62]. Since some of the NOT-DTN-projecting PNs preferentially respond to naso-temporal motion, they could provide innervation to the non-operating NOT-DTN (Supplementary Fig. 7e left), and thus depress the expression of OKR behavior. This idea is supported by our result that silencing the visual cortex on one side enhances the gain of the contraversive OKR but reduces the gain of ipsiversive OKR (Supplementary Fig. 7d-f). This finding is reminiscent of previous studies[49,51] reporting that unilateral lesion of monkey MT and MST or cat visual cortex enhanced the OKR running away from the side of lesion, i.e. contraversive OKR, but impaired ipsiversive OKR. Therefore, combining the facilitatory effect on ipsiversive OKR and suppressive effect on contraversive OKR, NOT-DTN-projecting PNs with a diverse direction bias allow animals to bidirectionally modulate innate behaviors.

The response properties of visual cortical neurons in adult animals undergo dramatic changes when animals learn visually guided behavioral tasks[38,39,63,64], or experience repetitive visual exposure[40–42]. An important step of transforming cortical plasticity to behavioral changes is the integration of cortical input in motor centers downstream of the corticofugal projections. We discovered that the NOT-DTN-projecting PNs become more responsive following OKR potentiation, with the maximum change observed in the temporo-nasally biased population. This cortical plasticity, at least in part, explains the enhanced cortical drive to the NOT-DTN, which in turn contributes to OKR potentiation. Remarkably, computational modeling shows that

the direction-selective cortical plasticity works synergistically with the temporo-nasal direction bias to efficiently integrate cortical input in the NOT-DTN neurons (Fig. 7c-e, Supplementary Fig. 12b, c). Overall, the modeling results provide a mechanistic framework to understand how corticofugal neurons modulate the brainstem activity in order to adapt innate behaviors.

In addition to cortical plasticity, the plastic changes in subcortical circuits have also been shown to play a critical role in the plasticity of oculomotor behaviors[29,30,32–34,65]. In a classical model, the OKR plasticity is induced by the retinal slip, a type of motor error signal conducted by the NOT-DTN projection to the IO and further down to the cerebellum[32,66,67]. IO neurons give rise to the climbing fibers that form synaptic connections with Purkinje cells in the cerebellar cortex[32,66,67]. Through these connections, the retinal slip can shape the plasticity of the synapses between the parallel fibers and Purkinje cells[68,69], a cerebellar mechanism underlying the adaptation of OKR. Interestingly, our data show that the projection from NOT-DTN to IO is well-suited for carrying the signals of retinal slips associated with horizontal environmental or head movements, because all the IO-projecting NOT-DTN neurons prefer the temporo-nasal motion (Fig. 5g) and they are required for OKR plasticity (Fig. 6i). More importantly, we found that the IO-projecting NOT-DTN neurons receive temporo-nasally biased cortical input (Fig. 1 & 2), indicating that in addition to the retinal input, the NOT-DTN → IO projection can also transmit cortical signals of retinal slips to the cerebellum. This result suggests that the visual cortex may also have an impact on the cerebellum-dependent OKR potentiation, which would imply an interaction between the cortical and cerebellar mechanisms of OKR plasticity. Thus, the IO-projecting NOT-DTN population is a key node that integrates the retinal signals and cortical signals of retinal slips and is essential for the adaptive plasticity of OKR.

What is the functional role of the visual cortex→NOT-DTN → IO pathway in the OKR? Under the baseline condition without OKR plasticity, this pathway has weak impact on the OKR behavior[2] (Figs. 3d-f, 6b-f, Supplementary Fig. 7). Instead, the retinal input is the driver of the OKR behavior, determining the basic level of OKR including its tuning curves (the set point) likely via NOT-DTN projections to pre-oculomotor structures different from the IO, such as the projections to the medial vestibular nucleus and NPH[53,54]. When the protocols of OKR potentiation are applied, NOT-DTN-projecting PNs become more active (Fig. 4d-e). Correspondingly, now the visual cortex contributes to the OKR gain more strongly and perturbing various stages of the visual cortex to IO pathway severely compromises the OKR potentiation, while largely preserving the OKR behavior and its tuning[2] (Figs. 3i, 6g-i). These findings support the idea that in the context of OKR plasticity the cortical input works as a modulator to fine-tune the behavioral responses around the set point. Thus, the cortico-brainstem pathway may play an important role mainly in the plasticity of innate behaviors.

Like monkeys and carnivores[70], the primary and higher visual areas in mice are hierarchically organized[24,45,46,71–73]. Based on inter-areal connectivity between visual areas, it has been proposed that those areas can be divided into functionally specialized ventral and dorsal streams[72,74]. However, considering the controversy in defining these two streams in mice[46,75,76], we used the cortical locations to name the two groups of HVAs (anterior vs posterior), which differ in the density of NOT-DTN-projecting PNs. Notably, the posterior HVAs (equivalent to the ventral stream areas) send significantly more projections to the NOT-DTN (Fig. 2b, c) and contribute more robustly to the OKR behavior and its plasticity (Fig. 3e, f, i, Supplementary Fig. 6b, c), compared to the anterior HVAs (equivalent to the dorsal stream areas). These results align with the idea that the mouse visual cortex, as in other mammalian species, also has two parallel streams of information processing. Moreover, the specific contribution from the posterior HVAs is reminiscent of previous findings that the motion sensitive areas in monkeys (MT and MST) project strongly to the NOT-

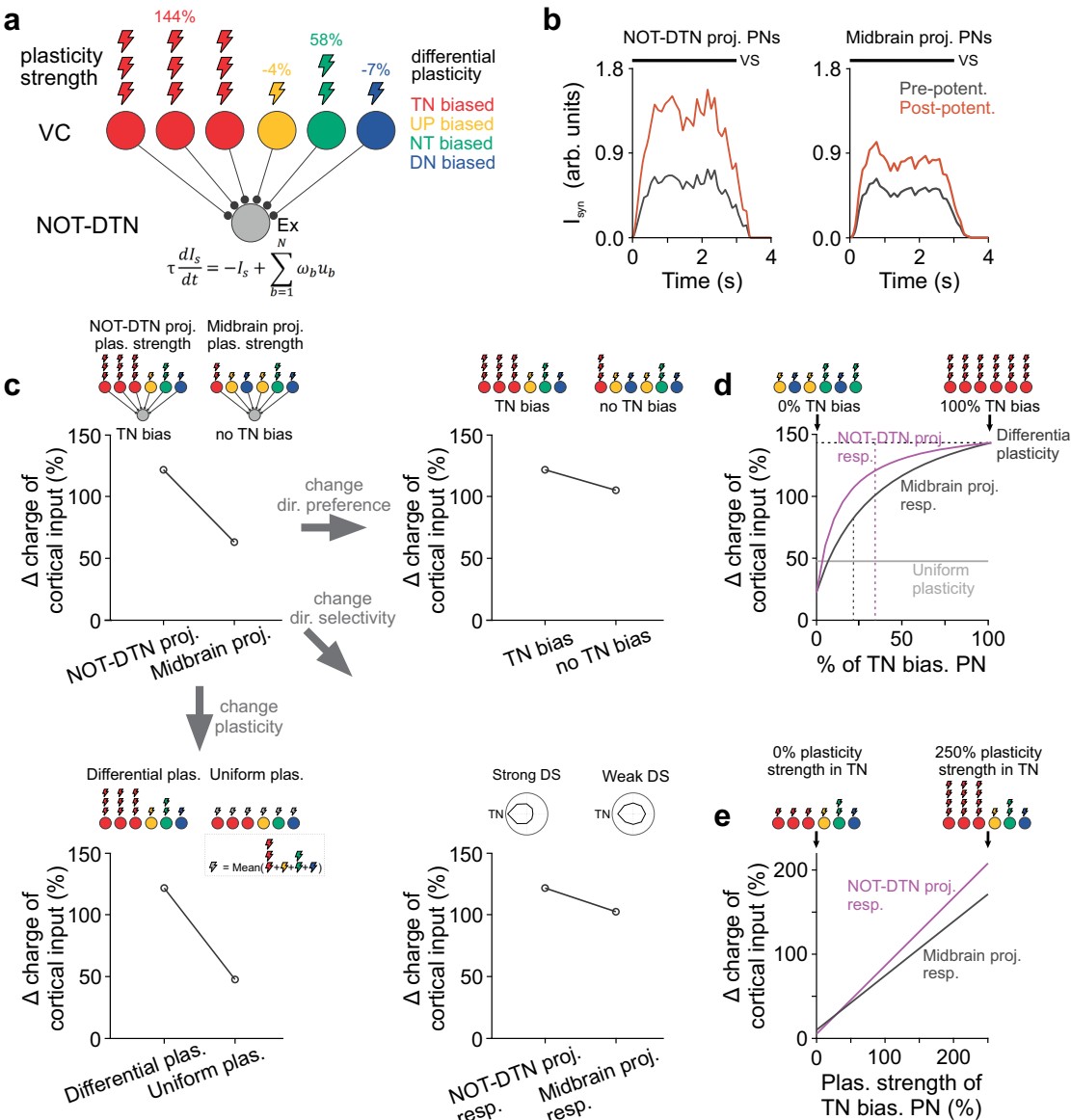

**Fig. 7 | Temporo-nasal bias and direction-selective plasticity promote the integration of cortical input. a** Feedforward model of the cortical input to the NOT-DTN in response to temporo-nasal visual motion. Colors encode preferred directions. The number of cortical neurons that prefer one of the four directions is determined according to data in Fig. 1d **solid**. The number of lightning bolts indicates cortical plasticity strength (data from Supplementary Fig. 8c). Note that the temporo-nasally biased pyramidal neurons (PNs) have stronger plasticity than PNs preferring other directions (differential plasticity). The firing rates are derived from the calcium responses in Fig. 1d–e. VC, visual cortex; Ex, excitation; TN, temporo-nasal; NT, naso-temporal; UP, up; DN, down. **b** Synaptic current provided by NOT-DTN-projecting (left) or midbrain-projecting (right) PNs. Black bar, the duration of visual stimulation (VS). **c** Percentage change in cortical input (charge) received by the NOT-DTN upon cortical plasticity. Schematics on top of each panel illustrate the parameters used in modeling. Top left, original models where the parameter values from either NOT-DTN-projecting PNs (left) or midbrain-projecting PNs (right) are used. Top right, the direction preference chosen from either NOT-DTN-projecting PNs (TN bias) or midbrain-projecting PNs (no TN bias). Bottom left, the plasticity strength chosen from either differential plasticity or averaged plasticity strength of four direction-biased populations (uniform plas.). Bottom right, cortical responses chosen from either NOT-DTN-projecting PNs (strong DS) or midbrain-projecting PNs (weak DS). **d** Plot of percentage change in cortical input (charge) vs the percentage of temporo-nasally biased PNs in three scenarios: responses of NOT-DTN-projecting PNs are used (magenta curve); responses of midbrain-projecting PNs are used (black curve); uniform plasticity is used (grey curve). Vertical dashed lines, actual percentages of temporo-nasally biased PNs in NOT-DTN-projecting (magenta) or midbrain-projecting (black) populations. Horizontal dashed line, the maximum percentage change in cortical input. **e** Plot of percentage change in cortical input (charge) vs plasticity strength of temporo-nasally biased PNs, when responses of NOT-DTN-projecting PNs (magenta curve) or midbrain-projecting PNs (black curve) are used.

DTN[77,78], and these projections share the temporo-nasal direction bias[27,28]. However, we recognize the discrepancy between mice and monkeys: MT and MST belong to the dorsal stream of monkey visual cortex; but the anterior HVAs in mice, the analog to the dorsal stream, barely project to the NOT-DTN and have little influence on the OKR (Figs. 2b, c, 3e, f, i). This discrepancy may reflect the species difference. Notwithstanding, a recent view considers the POR (one of the posterior HVAs) in mice to be the homolog of the MT area in monkeys[75] because they are both located on the temporal cortex and receive strong innervation from the ancient extrageniculate pathway (retina→SC → LP)[76,79]. This view reconciles the above discrepancy; thus, the area specific connectivity between the visual cortex and the NOT-DTN may stand for a circuit feature of corticofugal projections shared by mammalian species.

In conclusion, this study reveals three functionally relevant circuit features (strong temporo-nasal direction bias, area-specific origin, and direction-specific plasticity) that distinguish the corticofugal projection to the NOT-DTN from other cortico-brainstem projections. These specific functional properties endow cortico-brainstem projections with the capability of efficiently innervating brainstem targets to adaptively modulate innate motor behaviors.

## Methods

### Mice

All experimental procedures performed in this study were approved by the Biological Sciences Local Animal Care Committee, in accordance with guidelines established by the University of Toronto Animal Care Committee and the Canadian Council on Animal Care (protocol # 20012152 and 20012125).

We used the following mouse lines: *Ai14* tdTomato reporter[80] (Jackson Laboratory #007914), *VGAT-ChR2-EYFP*[81] (Jackson Laboratory #014548), *C57BL/6J* (Jackson Laboratory #000664), and *CD-1* (Charles River #022). Since there was no report of sex dimorphism in the cortical modulation of OKR, the sex was not considered in the experimental design. Mice of both sexes were used for experiments. Experimental mice were bred by crossing homozygous *VGAT-ChR2-EYFP*, homozygous *Ai14* or *C57BL/6J* males with wild-type *CD-1* females. Mice were housed in a vivarium with a reversed light cycle (12 h day/ 12 h night), ambient temperature at 22 °C, and humidity at 50%. *CD-1* x *C57BL/6J* hybrid mice were used in experiments where wild-type mice were needed unless otherwise noted.

### Viral injections

We used the following adeno-associated viruses (AAV):

For Cre recombinase (Cre) dependent expression of GcaMP6s: AAV9-Syn-Flex-GcaMP6s -WPRE-SV40[82] (Flex-GcaMP6s, Addgene 100845; Upenn Vector Core)

For retrograde expression of Cre: AAVrg-pmSyn1-EBFP-Cre[57] (Retro-Cre, Addgene 51507-AAVrg) or AAVrg-hSyn-Cre-WPRE-hGH (Retro-Cre, Addgene 105553-AAVrg)

For Cre-dependent expression of tdTomato: AAV1-CAG-Flex-tdTomato-WPRE-bGH (Flex-tdTomato, Allen Institute 864; Upenn Vector Core)

For Cre-dependent expression of hM4D: AAV9-hSyn-DIO-hM4D(Gi)-mCherry[83] (DIO-hM4Di- mCherry, Addgene 44362-AAV9)

For the local and anterograde expression of Cre: AAV1-hSyn-Cre-WPRE-hGH[84] (Antero-Cre, Addgene 105553-AAV1)

For the expression of Chronos: AAV1-Syn-Chronos-GFP[85] (Addgene 59170-AAV1)

For the expression of tdTomato: AAV1-CAG-tdTomato-WPRE-SV40 (CAG-tdTomato, Upenn Vector Core)

For Cre-dependent expression of ArchT: AAV1-CAG-Flex-ArchT-GFP (Flex-ArchT, UNC Vector Core)

For retrograde expression of Flp recombinase (Flp): AAV2/retro-hSyn-FlpO (Retro-Flp, neurophotonics construct-1683)

For FLPo-dependent expression of TVA-2A-oG: AAV2/8-CAG-fDIO-TVA-mCherry-2A-oG (fDIO-TVA-mCherry-oG, neurophotonics construct-1400)

For Cre-dependent expression of TVA-2A-oG: AAV9-hSyn-Flex-TVA-P2A-eGFP-2A-oG (Flex-TVA-eGFP-oG, Addgene 85225; Salk Viral Vector Core)

The following rabies viruses were used:

For retrograde trans-synaptic expression of GCaMP6s: RV-EnvA-SADΔG-GCaMP6s (RV-GCaMP6s, gift from Dr Georg Keller at FMI)

For retrograde trans-synaptic expression of mCherry: RV-EnvA-SADΔG-mCherry (RV-mCherry, Addgene 32636; Salk Viral Vector Core)

To retrogradely label L5 PNs in the visual cortex that project to the NOT-DTN or midbrain, 2–4-month-old *Ai14* mice were anaesthetized with 1.5–2% isoflurane (vol/vol) in O$_2$. The depth of anesthesia

was monitored with the toe-pinch response. Eyes were protected with lubricant eye ointment (Systane, Alcon). The animal's body temperature was maintained by a heating pad (HTP-1500, Kent Scientific). Carprofen (Rimadyl, Pfizer) was administered subcutaneously at a dose of 5 mg/kg to reduce pain. Furs over the head were shaved and the exposed skin was disinfected with 70% isopropyl alcohol and iodine solution. The scalp was cut open and part of the skull over the NOT-DTN (~300−500 μm diameter) was thinned until it became soft enough to allow the penetration of the beveled glass pipette (diameter 22–27 μm). To selectively label NOT-DTN-projecting PNs, Retro-Cre virus[57] was injected in the NOT-DTN of the left hemisphere (coordinates: anteroposterior axis (AP) relative to bregma, 2.60 mm; mediolateral axis relative to midline, 1.10 mm; depth, 1.62 mm; the actual coordinates were calculated according to a standard bregma-lambda distance of 4.21 mm) by an iontophoresis pump (+ 4 μA, 7 s on/7 s off, for 3 min, BAB-600, Kation Scientific). To non-selectively label midbrain-projecting PNs, a bolus of Retro-Cre virus (120 nL) was injected into the NOT-DTN (the same coordinates as labeling the NOT-DTN-projecting PNs) by a microinjection pump (UMP3T, WPI). This method allowed the virus to go beyond the NOT-DTN and infect neighboring nuclei. The scalp was sutured with a few stitches of sterile 6-0 silk suture (667 S, CP Medical). Mice were subcutaneously injected with lactated ringer's solution at a rate of 1 ml/h (JB2324, BAXTER) and recovered on the head pad before returning to the home cage. Mice were transcardially perfused 3 weeks after the virus injection.

To functionally image NOT-DTN/midbrain-projecting, or non-selective L5 PNs in the visual cortex, AAV-Flex-GCaMP6s[82] was injected into the left visual cortex of *Ai14* pups by a Nanoliter Injector (NANOLITER2010, WPI). Pups were anaesthetized by hypothermia and placed on a molded platform. The viral injection was done at two sites along the media-lateral axis of the visual cortex. At each site, a bolus of GCaMP6s virus (42 nL was injected at each of two depths (0.35 and 0.5 mm from the skin). Then, at postnatal 2–4 months, Cre virus was injected. To image NOT-DTN-projecting or midbrain-projecting PNs, Retro-Cre virus was stereotactically injected in the left NOT-DTN with an iontophoresis pump or with a microinjection pump respectively, as described above. Alternatively, to image non-selective L5 PNs a bolus of AAV1-hSyn-Cre-WPRE-hGH virus (100 nL, 100x dilution with PBS) was injected at each of six sites in the left visual cortex, at a depth of 700 μm from the pial surface (along the axis perpendicular to the cortex). Experiments were performed 3 weeks after the injection of Cre viruses.

To anterogradely label the NOT-DTN neurons that are innervated by the visual cortex, a bolus of AAV-Flex-tdTomato virus (50 nL) was stereotactically injected in the left NOT-DTN of adult WT mice, as described above. Subsequently, a bolus of Antero-Cre[84] (200nL) was injected into layer 5 of the left visual cortex (depth 0.7 mm from the pial surface along the axis perpendicular to the horizontal plane) at each of eight sites. To maximize the efficiency of the anterograde tracing, we chose the injection sites where NOT-DTN-projecting PNs are populated (Fig. 2b, coordinates: site 1 (AP, 2.8 mm; ML, 2.5 mm); site 2 (AP, 3.3 mm; ML, 2.5 mm); site 3 (AP, 3.7 mm; ML, 2.3 mm); site 4 (AP, 3.7 mm; ML, 2.8 mm); site 5 (AP, 3.83 mm; ML, 2.52 mm); site 6 (AP, 3.9 mm; ML, 3.7 mm); site 7 (AP, 4.3 mm; ML, 3.7 mm); site 8 (AP, 4.6 mm; ML, 3.8 mm)). Mice were transcardially perfused 3 weeks after the virus injection.

To trace the axonal projections of NOT-DTN neurons, a bolus of AAV-CAG-tdTomato (40nL) was stereotactically injected in the left NOT-DTN at postnatal 5 months, as described above. Mice were transcardially perfused 3 weeks after the virus injection.

To chemogenetically silence IO-projecting NOT-DTN neurons, a bolus of AAV-DIO-hM4Di-mCherry[83] (100 nL) was stereotactically injected into the NOT-DTN of VGAT-ChR2-EYFP mice bilaterally at postnatal 2–4 months (as described above) and a bolus of Retro-Cre

(350 nL) was subsequently injected into the IO at each of two sites (coordinates: AP, 5.5 and 6.0 mm; ML, 0.0 mm; depth, 5.0 mm; the actual coordinates were calculated according to a standard bregma-lambda distance of 4.21 mm). This combination selectively expressed hM4Di in NOT-DTN neurons that project to the IO. In control experiments, in replacement of AAV-DIO-hM4Di-mCherry, 100 nL of AAV-Flex-tdTomato was instead injected into the NOT-DTN. Experiments were performed 2 weeks after the viral injection.

To retrogradely label IO-projecting NOT-DTN neurons, a bolus of AAV-Flex-tdTomato (100 nL) was stereotactically injected into the left NOT-DTN of *CD-1 x C57BL/6J* mice and a bolus of Retro-Cre (100 nL) was subsequently injected into the IO at each of two sites at postnatal 2–4 months (as described above).

To optogenetically silence IO-projecting NOT-DTN neurons, a bolus of AAV-Flex-ArchT (150 nL) was stereotactically injected into the left NOT-DTN of *CD-1 x C57BL/6J* mice and a bolus of Retro-Cre (200 nL) was subsequently injected into the IO at each of two sites at postnatal 2–4 months (as described above).

To functionally image L5 PNs in the visual cortex that innervate IO-projecting NOT-DTN neurons, and to analyze the spatial distribution of the visual cortical neurons or retinal ganglion cells that innervate IO-projecting NOT-DTN neurons, a bolus of AAV-fDIO-TVA-mCherry-oG (150 nL) was stereotactically injected into the left NOT-DTN of *CD-1 x C57BL/6J* mice and a bolus of Retro-Flp (200 nL) was subsequently injected into the IO at each of two sites at postnatal 2–4 months (as described above). 3 weeks after the AAV injections, a bolus of RV-GCaMP6s (150 nL) was stereotactically injected into the left NOT-DTN. Two-photon imaging was performed 8–11 days after the injection of rabies virus. And then mice were transcardially perfused immediately. To retrogradely label retinal ganglion cells, mice were transcardially perfused, and the retinas were dissected 10 days after the injection of rabies virus.

To analyze the dendritic morphology of retinal ganglion cells that innervate IO-projecting NOT-DTN neurons, the same procedure described above was implemented, except that Flex-TVA-eGFP-oG, Retro-Cre, and RV-mCherry were used.

To antidromically stimulate IO-projecting NOT-DTN neurons, a bolus of AAV1-Syn-Chronos-GFP[85] (300 nL) was stereotactically injected into the left NOT-DTN of adult WT mice as described above. Experiments were performed 3–4 weeks after the viral injection.

## Two-photon calcium imaging of L5 PNs in the visual cortex[86]

To implant headplates and cranial glass windows, the scalp over the visual cortex was removed. Then, we applied several drops of the solution of 1% lidocaine and 1:100,000 epinephrine (XYLOCAINE, Dentsply Sirona) on the skull to minimize pain and bleeding. The temporalis muscle was separated from the skull to further increase the contact surface for attaching headplates. We scraped the skull to remove the fascia and cleaned it with PBS. After being dried with compressed air, the exposed skull was then covered with a thin layer of superglue (All Purpose, Krazy Glue). Then, a headplate (Supplementary Fig. 1a right) was glued onto the skull, with its opening centering on V1 or posterior HVAs of the left hemisphere (center coordinates: 3.8 mm from the midline, 0.7 mm anterior to the lambda suture for imaging V1; 4.4 mm from the midline, 0.5 mm anterior to the lambda suture for imaging HVAs) with acrylic resin (B1356, Lang Dental) mixed with black paint (iron oxide, Alpha Chemicals). Next, a craniotomy of ~3 mm in diameter was made at the center of the headplate opening, and the dura was removed. After rinsing off the bone debris with saline and stopping bleeding, a 3 mm round cover glass sealed the craniotomy with superglue (Gel Control, Loctite) and acrylic resin (B1356, Lang Dental) mixed with black paint (iron oxide, Alpha Chemicals). Carprofen (dose 5 mg/kg) was administered daily for 5 days after the surgery. Animals were allowed to recover for 3 weeks before the two-photon calcium imaging.

All visual stimuli used in this study were generated with the PsychoPy (1.90.2) running in Python 2. Visual stimuli for 2p imaging were displayed on a computer monitor (U2415, Dell, refresh rate 60 Hz, gamma-corrected), which was placed 20 cm from the animal's right eye and covered ~60 × 75° of the visual space. To map retinotopic receptive fields of L5 PNs neurons, we divided a monitor into a 3×4 grid and presented contrast-modulated Gaussian noise movies[87] in one location at a time in a random sequence. We used a spatial frequency spectrum with a sharp cutoff at 0.2 cpd and a temporal frequency spectrum with a sharp low-pass cutoff at 10 Hz to construct those movies. To evaluate direction selectivity, we presented sinusoidal gratings drifting in eight evenly spaced directions to the animal's contralateral eye with the temporo-nasal direction defined as 0° (spatial frequency (SF), 0.08 cpd; temporal frequency (TF), 1 Hz; contrast, 100%; mean luminance, 40 cd/m²). In each trial, the grating moved for 3 sec with a 6-sec interstimulus interval which is defined as the duration between the offset of one stimulus and the onset of the next stimulus. In blank trials, a grey screen (luminance, 40 cd/m²) was presented to measure the spontaneous activity of L5 PNs. To evaluate SF or TF tuning, we systematically varied the SF value (0.02–0.64 cpd, TF fixed at 1 Hz) or TF value (0.25–8 Hz, SF fixed at 0.08cpd) of a temporo-nasally drifting grating. Each stimulus was repeated 15 times and the order of stimuli in each repetition was randomized.

Several days before two-photon imaging, mice were familiarized with head fixation in the imaging setup at least three times. The animal's head was tilted by 5° when imaging posterior HVAs and 10° when imaging V1 so that the cranial window was orthogonal to the vertically oriented objective. No visual stimulation was given. Two-photon calcium imaging was performed on awake head-fixed mice with a customized two-photon microscope (MOM, Sutter instrument), equipped with a water dipping objective (CFI75 LWD 16X W, Nikon). We used an infrared laser of 940 nm (35–70 mW, Insight X3, Spectra Physics) to excite both GCaMP6s and tdTomato. The fluorescence lights were filtered by a dichroic mirror (565dcxr, Chroma Tech) and two emission filters (ET525/70m-2p, ET605/70m-2p, Chroma Tech) and detected by GaAsP PMTs (H10770PA-40 SEL, Hamamatsu). ScanImage 2018 (Vidrio Technologies) was used to record two-photon imaging. Two planes in layer 5 (30 μm apart, 512 ×512 pixels, covering 416.8×416.8 μm²) were imaged simultaneously at 10 Hz with the control of a piezo stage (nPfocus 400, npoint). Retrogradely labeled PNs appeared at a depth of 435–550 μm from the pial surface. Recording started 1 sec before the visual stimulation and lasted until 1 sec after the visual stimulation finished (totally 5 sec). The space between the objective and the cranial window was sealed to prevent the light from the monitor from contaminating the two-photon fluorescence signal.

## Identifying visual areas of L5 PNs revealed by two-photon imaging

We used the pattern of blood vessels as a landmark to locate visual cortical neurons revealed by two-photon imaging (Supplementary Fig. 5a). This process includes four steps. First, we overlaid a 2p image of the pial surface which contains a network of blood vessels on a 2p image of the L5 PNs at the same location. Second, according to the blood vessels, we superimposed the above 2p image on top of a wide-field image of the cranial window from a live animal taken by a Stereo Microscope (SteREO Discovery V12, ZEISS). Third, 2 h after euthanizing the mice with CO₂, we dissected the brain and then superimposed the resulting image from step 2 on top of a top-view wide-field image of the whole-mount brain according to the vasculature. Fourth, outlines of visual areas in the Allen mouse brain atlas were projected onto the image of the whole brain by matching the profile of the standard mouse brain in the atlas with that of the sample brain. Lastly, Suite2p software (python version, v0.7.1) was used to define individual neurons based on tdTomato fluorescence and compute their coordinates. A series of coordinate transformations were done to convert the

coordinates in the 2p image to the coordinates in the Allen mouse brain atlas. Then, we determined which visual areas L5 PNs belong to.

## Assessment of OKR behavior

The horizontal OKR was stimulated by a 'virtual drum' grating[2,88]. Three computer monitors (B247W, Acer, refresh rate 60 Hz, gamma-corrected) were mounted orthogonally to each other to form a square enclosure. The enclosure covered ~270° of the visual space along the azimuth and 63° vertically. With a discrete graphic card (Geforce GTX 1650 or Quadro P620, NVIDIA), three monitors were merged into a single display to ensure frames were synchronized across multiple monitors. A vertical sinusoidal grating with a constant spatial frequency was generated by adjusting the period of grating stripes throughout the azimuthal plane[2], as if the grating was projected onto the surface of a 'virtual drum'. Mice were head-fixed at the center of the drum with their eyes positioned 14° below the vertical center of the monitors. Along the horizontal axis, the grating drifted bidirectionally or unidirectionally. In bidirectional stimulation, the grating rotated clockwise or counterclockwise in an oscillatory manner for 10 sec (oscillatory amplitude ±5°; SF 0.04–0.45 cpd; oscillation frequency 0.4 Hz; contrast 80%; mean luminance 40 cd/m$^2$) with an interstimulus interval of 8 sec. In unidirectional stimulation, the grating drifted in one direction at a constant speed for 10 sec (SF 0.04–0.45 cpd; TF 0.4 Hz; contrast 80%; mean luminance 40 cd/m$^2$) with an interstimulus interval of 10 sec. To evaluate the spatial frequency tuning, we varied the spatial frequency of the grating between 0.04 cpd and 0.45 cpd. Each stimulus was repeated 20–40 times. In cortical silencing experiments, the inter-stimulation interval increased to 20 sec after the silencing trials. To examine the open-loop OKR behavior, the grating (SF 0.04–0.45 cpd; TF 0.4 Hz; contrast 80%; mean luminance 40 cd/m$^2$) drifted counterclockwise, that is to say, temporo-nasal direction in reference to the right eye or ipsiversive to the left NOT-DTN. During the test, the infrared video-oculography system (see below) sent the real-time position of right eye to the visual stimulation software. This allowed the visual stimulation software to modify the speed of grating movement according to the eye position in order to keep the retinal slips at constant speeds[89]. To stimulate OKR monocularly, the animal's left eye was occluded by a piece of clay.

To monitoring eye movements with infrared video-oculography[2,90], a T-shaped head bar was implanted on an animal's skull for head fixation in a similar way to the implantation of headplates for two-photon imaging. Mice were head-fixed in an upright position. Mice were familiarized with head fixation in the setup at least three times. The movements of the right eye were recorded by a high-speed infrared (IR) camera (G3-GM12-M0640, Teledyne Dalsa, frame rate 100 Hz) under the control of custom LabVIEW software (2014, National Instruments). The camera captured the reflection of the eye on an IR mirror that was transparent to visible light (64–471, Edmund Optics). In each trial, the video recording preceded the grating stimulation by 1 sec, and ended 1 s after the visual stimulation stopped. The pupil in the video was detected online by a two-step process: (1) use a thresholding algorithm to estimate the center of mass of the pupil; (2) from the estimated pupil center, run a one-step starburst algorithm to profile the boundary of the pupil. The eye position was calculated based on the distance between the pupil center and the corneal reflection of a reference IR LED placed along the optical axis of the camera. To calibrate the measurement of the eye movements, the camera and the reference IR LED together were moved by ±10° along a circle centered on the image of the eye.

## In vivo extracellular recordings from the NOT-DTN of anesthetized mice

Visual stimuli for extracellular recordings were displayed on three computer monitors (S27R35AFH, Samsung, refresh rate 60 Hz,

gamma-corrected) which were mounted orthogonally to each other to form a square enclosure. To ensure multiple monitors were synchronized, we used AMD Eyefinity Technology (Radeon Pro WX5100, AMD). With the help of a T-shaped head bar, the animal's head was immobilized at the center of the enclosure. To compare the response properties of L5 PNs of the visual cortex and NOT-DTN neurons (Fig. 1a-d, Supplementary Fig. 3), we used the same types of visual stimulation as the ones used in two-photon calcium imaging (see above), except that the interstimulus interval was reduced to 2 sec. To investigate the response properties of IO-projecting NOT-DTN neurons, we adapted the visual stimulation to best activate NOT-DTN neurons. To evaluate direction selectivity (Fig. 5c-i), on the frontal monitor of the 'virtual drum' apparatus (Fig. 3a) sinusoidal gratings drifted in twelve evenly spaced directions (SF, 0.08 cpd; TF, 1 Hz; contrast, 100%; mean luminance, 50 cd/m$^2$), while keeping the left and right monitors in uniform luminance. The temporo-nasal direction in reference to the contralateral eye was defined as 0°. In each trial, the grating moved for 2 sec with a 6-sec interstimulus interval. Individual stimuli were repeated 25 times and delivered in a random sequence. To evaluate SF tuning (Supplementary Fig. 10), we systematically varied the SF value (0.04–0.45 cpd, oscillation frequency fixed at 0.4 Hz) of drum gratings drifting horizontally in an oscillatory manner (see "Visual stimulation in Assessment of OKR behavior"). In each trial, the grating moved for 5 sec with a 6-sec interstimulus interval. Each stimulus was repeated 30 times.

For extracellular recordings, a T-shaped head bar was stereotactically implanted on the mouse skull for head fixation under the guidance of an inclinometer (551-1002-1-ND, Digi-Key electronics)[2]. The inclinometer allowed us to calibrate the inclination of the two axes of the T bar relative to the AP and ML axes of the skull before fixing it to the skull with dental cement. Three reference points with known coordinates were marked on the mouse skull because both bregma and lambda were inevitably masked by the dental cement holding the head bar. The head post on the recording rig was also calibrated with the same inclinometer to ensure that the recording probes were in register with the skull. Animals were anaesthetized with an intraperitoneal injection of urethane (1 g/kg,) in addition to intramuscular injection of the sedative chlorprothixene (0.05 mL of 4 mg/mL). Body temperature was maintained at 37 °C using a feedback-controlled heating pad (50-7220-F, Harvard apparatus). A thin layer of silicone oil (378429, Sigma Aldrich) was applied to both eyes to prevent drying. Lactated Ringer's solution was administered at a rate of 1 mL/h to prevent dehydration. To access the NOT-DTN, we made a craniotomy of 2×2 mm around the coordinates of 2.9 mm (antero-posterior) and 1.3 mm (medio-lateral). A drop of PBS was applied on top of the craniotomy to keep the exposed brain moist. A 32-channel poly2 silicon probe (ASSY-37-H6B, Cambridge Neurotech) mounted on a manipulator (IVM Mini, Scientifica) was slowly advanced into the brain to a depth of 2000–2200 μm. Activity preferring the temporo-nasal motion indicated the proper targeting of the NOT-DTN. Electrophysiological signals were amplified and filtered with an extracellular amplifier (C3315, Intan Technologies) and digitized at 30 kHz by a USB interface board (C3100, Intan Technologies) under the control of Intan RHD 2000 interface V1_5_2. Raw data were stored on a computer hard drive for offline analysis.

To optogenetically identify IO-projecting NOT-DTN neurons with extracellular recording, we made an antero-posteriorly oriented craniotomy ( ~0.5x2mm) above the IO (center coordinates: AP, 7 mm; ML, 0 mm) for photo-stimulating the axons from the NOT-DTN which expressed Chronos[85]. An optical fiber of 100 μm in diameter (MFC_100/125-0.66_12mm_ZF1.25_FLT Doric lenses) was inserted into a depth of 4500–5000 μm from the surface of the brain. A train of blue light pulses (duration=5 ms, frequency=5 Hz) were provided by a 470 nm laser (22–24 mW, MDL-III-470, CNIlaser) to optogenetically stimulate the axons of IO-projecting NOT-DTN neurons for optogenetic tagging.

The antidromically generated action potentials were recorded by electrodes inserted into the NOT-DTN and recognized by their little jitter and short onset latency.

## Optogenetic silencing of the visual cortex

*VGAT-ChR2-EYFP* mice were used in experiments involving optogenetic silencing of the visual cortex[2]. C57 mice were used as a control experiment. To allow an efficient penetration of blue light, we gently thinned the skull above the whole visual cortex or higher visual areas until it became transparent when wetted by PBS solution. After being dried, the cranial window was then covered with a thin layer of Crazy glue and a T-shaped head bar was mounted at the midline of the skull for head fixation. 1 week after recovery, the mice were familiarized with head fixation in the setup three times without visual stimuli. To photo-stimulate ChR2-expressing cortical inhibitory neurons in vivo, a 470 nm fiber-coupled blue LED (1 mm diameter, LEDP-B_PF960-0.50_1m Doric Lenses or M470F3 Thorlabs) was placed ~5–10 mm above the cranial window of each hemisphere. We restricted the photo-stimulation within the desired areas by covering the surrounding areas with dental cement mixed with black paint (iron oxide, Alpha Chemicals). The light power at the end of the fiber was 12–16 mW (corresponding to a power density of 1.2–1.6 mW/mm$^2$). LED was turned on during the whole period of visual stimulation in half of the recording trials. The timing of LED was controlled by the LabVIEW software (2014, National Instruments) which recorded eye movements.

To optogenetically silence anterior HVAs or posterior HVAs, the skull above both anterior and posterior HVAs was thinned. We delineated the boundary of the collection of anterior HVAs with a nonagon defined by nine sets of coordinates (coord. 1: AP, 1.56 mm; ML, 1.31 mm; coord. 2: AP, 1.93 mm; ML, 2.33 mm; coord. 3: AP, 2.30 mm; ML, 3.31 mm; coord. 4: AP, 2.93 mm; ML, 3.95 mm; coord. 5: AP, 2.84 mm; ML, 3.32 mm; coord. 6: AP, 2.71 mm; ML, 2.85 mm; coord. 7: AP, 2.20 mm; ML, 2.39 mm; coord. 8: AP, 3.26 mm; ML, 1.85 mm; coord. 9: AP, 4.31 mm; ML, 1.31 mm). And we delineated the boundary of posterior HVAs with a heptagon defined by seven sets of coordinates (coord. 1: AP, 2.93 mm; ML, 3.95 mm; coord. 2: AP, 3.78 mm; ML, 3.85 mm; coord. 3: AP, 4.00 mm; ML, 4.00 mm; coord. 4: AP, 4.51 mm; ML, 3.70 mm; coord. 5: AP, 4.84 mm; ML, 3.08 mm; coord. 6: AP, 3.96 mm; ML, 3.43 mm; coord. 7: AP, 2.84 mm; ML, 3.32 mm). The actual coordinates were calculated according to a standard bregma-lambda distance of 4.21 mm. To verify the boundary of HVAs, we used the pattern of blood vessels as a landmark as described above (Supplementary Fig. 5a). During the photo-stimulation, the targeted HVAs were exposed to blue LED, while the other HVAs were covered with silicone elastomer (Kwik-Sil, WPI) mixed with black paint (iron oxide, Alpha Chemicals).

## Chemogenetic silencing of IO-projecting NOT-DTN neurons

To suppress the activity of IO-projecting NOT-DTN neurons which expressed an inhibitory chemogenetic tool hM4Di[83], clozapine N-oxide (CNO, HB6149, Hello bio, 0.1 mg/mL in saline) was administered through i.p. injection at a dose of 1 mg/kg. In control experiments, an equal volume of saline was injected to examine the non-specific effect of CNO. OKR behavior and the cortical contribution to OKR gain were examined 30 min after the injection.

## Optogenetically silencing of IO-projecting NOT-DTN neurons

To suppress the activity of IO-projecting NOT-DTN neurons which expressed an inhibitory opsin ArchT, a fiber optic cannula (200um in diameter, 0.39NA, 3 or 3.5 mm length, R-FOC-BL200C-39NA, RWD) was inserted above the left NOT-DTN two weeks after the injection of AAV-Flex-ArchT. 1 week after recovery, the mice were familiarized with head fixation in the setup three times without visual stimuli. To optogenetically suppress the activity of ArchT-expressing NOT-DTN neurons, a 554 nm fiber-coupled green LED (MINTF4, Thorlabs) was

connected to the implanted fiber cannula, through which 2.4 mW of green light was delivered on top of the NOT-DTN. LED was turned on during the whole period of visual stimulation in half of the recording trials. The timing of LED was controlled by the LabVIEW software (2014, National Instruments) which recorded eye movements.

## Histology

Under anesthesia (urethane 1.5 g/kg; intraperitoneal injection; 94300, Sigma Aldrich), mice were perfused transcardially first with PBS and then with 4% paraformaldehyde in PBS (pH 7.4). Brains were dissected from the skull, post-fixed overnight in 4% paraformaldehyde, and then immersed in 30% sucrose in PBS until they sank. Finally, we coronally sectioned brains into 50–100 μm thick slices with a sliding Microtome (HM450, Thermo Scientific).

## Examining the spatial distribution of trans-synaptically labeled retinal ganglion cells

After transcardial perfusion, the temporal side of cornea was marked by a cautery (AA00, Bovie), which provides a reference to individual sectors of the retina (superior, inferior, nasal, temporal). Then, the eyeball was removed from the eye socket and further fixed in 4% paraformaldehyde for at least 1 h. After fixation, the retina was dissected[91].

## Examining the dendritic morphology of retinal ganglion cells labeled by RV-mCherry

After the dissection, the retina was washed 3 times with PBS at room temperature for 5 min each round. Then the retina was incubated with a blocking solution (5% horse serum (H1138, Sigma Aldrich) + 0.3% Triton X-100 (BP151–500, Fisher Scientific) in PBS) at room temperature for 1 h and then incubated with primary antibodies goat anti-VAChT (1:800, Millipore, ABN100), goat anti-ChAT (1:200, AB144P, Millipore), rabbit anti-RFP (1:1000, ab62341, Abcam) at 4 °C for 5 days. Next, the retina was washed 3 times with PBS at room temperature for 30 min each round and then incubated with secondary antibodies Donkey anti-goat-Alexa Fluor 488 (1:500, A32814, Invitrogen) and Donkey anti-rabbit-Alexa Fluor 594 (1:1000, A32754, Invitrogen) at room temperature for 2 h. Then the retina was washed 3 times with PBS at room temperature for 5 min each round and mounted in a home-made mounting medium (80% glycerol (BDH1172, VWR), 1% DABCO (D27802, Sigma Aldrich) in PBS, pH 8.6). The above commercial antibodies have been previously validated by their manufactures and were well characterized in the literature.

## Validating chemogenetic silencing of IO-projecting NOT-DTN neurons with c-Fos immunostaining

Before the experiment, mice underwent dark accommodation overnight to minimize the baseline c-Fos expression. 30 min after i.p. injection of CNO or saline, mice were stimulated by drum gratings drifting along the horizontal axis (oscillatory amplitude ±5°; SF 0.08, 0.16 and 0.32 cpd; oscillation frequency 0.4 Hz; contrast 80%; mean luminance 40 cd/m$^2$; duration 15 s; inter-stimulus interval 5 s; 180 trials in total) for 60 min. 90 min after the beginning of OKR stimulation (30 min after OKR simulation was finished), animals were perfused transcardially and their brains were processed and sectioned as described above. The slices were washed 5 times with PBST (PBS, 0.1% Triton X-100) at room temperature for 5 min each round. Then the slices were incubated with a blocking solution (10% goat serum (16210072, Thermofisher) in PBST) at room temperature for 1 h and then incubated with rabbit anti-c-Fos primary antibody (1:1000, 226003, Synaptic Systems) in the blocking solution at 4 °C overnight. Next, the slices were washed 5 times with PBST at room temperature for 5 min each round and then incubated with a secondary antibody conjugated with Alexa Fluor 633 anti-rabbit (1:500, A21070, Invitrogen) in the blocking solution at room temperature for 2 h. After being washed 5 times with PBST and then 5 times with PBS, the slices were

stained with DAPI (0.05 μg/mL in PBS, D1306, Invitrogen) for 10 min. The slices were washed 3 times with PBS for 5 min each round and mounted in the homemade mounting medium described above. The above commercial antibodies have been previously validated by the manufactures and were well characterized in the literature.

## Validating optogenetic silencing of IO-projecting NOT-DTN neurons with c-Fos immunostaining

Before the experiment, mice underwent dark accommodation overnight to minimize the baseline c-Fos expression. Mice were stimulated by drum gratings unidirectionally drifting counterclockwise (SF 0.08, 0.16 and 0.32 cpd; TF 0.4 Hz; contrast 80%; mean luminance 40 cd/m²; duration 10 s; inter-stimulus interval 5 s) for 60 min. In half of animals, 554 nm LED light was delivered on top of the NOT-DTN during visual stimulation in all trials (as mentioned above). 90 min after the beginning of OKR stimulation (30 min after OKR simulation was finished), animals were perfused transcardially, and their brains were processed and sectioned. Brain slices containing NOT-DTN were immunostained against c-Fos with the same procedure mentioned above.

## Microscopy

Confocal images were acquired on a Zeiss LSM 880 confocal microscope (software: Zeiss Zen 2.3 SP1). Widefield images of cranial windows and dissected brains were captured using a Zeiss Discovery V12 stereoscope (software: AxioVision 4.8.2). Olympus VS200 Slide Scanner (software: OlyVIA 3.4.1) was used to document the NOT-DTN-innervating PNs labeled by rabies virus-based trans-synaptic tracing. ImageJ (National Institutes of Health) was used to process images and the SNT was used to reconstruct the morphologies of trans-synaptically labeled RGCs. RGCs which had 60% or more of their dendritic arbors in the ON sublamina are defined as ON RGCs; the rest are defined as ON-OFF RGCs.

## OKR potentiation

We induced OKR potentiation by presenting continuous OKR stimulation to an animal head-fixed under the two-photon microscope[2,30,31]. An enclosure of three orthogonally mounted computer monitors was tilted to match the tilting of the animal's head (see "Two-photon imaging" above). Presented on the three monitors, A 'drum grating', as described above, drifted horizontally in an oscillatory manner for 45 min (oscillatory amplitude ±5°; SF 0.1 cpd; oscillation frequency 0.4 Hz; contrast 100%; mean luminance 35 cd/m²). In the blank condition, a grey screen was instead presented on the three monitors for 45 min. In the control condition of vertical OKR stimulation, a horizontal 'drum' grating (with the same parameters to the vertical drum grating) was presented on the left and right monitors. The drum grating rotated vertically in an oscillatory manner for 45 min between two rounds of OKR measurement and 2p imaging. Calcium responses of L5 corticofugal neurons to four motion directions (0°, 90°, 180°, 270°) were measured before and after the induction of OKR potentiation. The horizontal OKR behavior triggered by the same visual stimulation parameters (oscillatory amplitude ±5°; SF 0.1 cpd; oscillation frequency 0.4 Hz; contrast 100%; mean luminance 35 cd/m²) was also evaluated before and after the induction of OKR potentiation, as described above.

## Feedforward integration model

To evaluate the impact of direction bias of corticofugal neurons and cortical plasticity on cortical input to the NOT-DTN, we applied a firing-rate-based feedforward integration model[92]. Spiking activity of corticofugal neurons was estimated by deconvolving their calcium responses ($\tau = 1.5$ sec for GCaMP6s) with Suite2p software (python version, v0.7.1)[93]. In both NOT-DTN-projecting population and midbrain-projecting population, neurons were categorized into four groups based on their preferred directions (TN bias, UP bias, NT bias, DN bias). Within each group, the firing rate dynamics of PNs

were averaged after being normalized to the preferred direction, and then scaled to the average firing rates of the population (Supplementary Fig. 12a). The spontaneous firing rate (the averaged firing rate before the onset of visual stimuli) was subtracted. We used the PNs from the evaluation of direction selectivity (Fig. 1) to derive the firing rates before OKR potentiation. And the firing rates after OKR potentiation were estimated by multiplying the firing rates before OKR potentiation and the percentage changes of calcium responses of the corresponding groups (Fig. 7a, Supplementary Fig. 8c). Then, the synaptic current ($I_s$) supplied by the corticofugal neurons was calculated as

$$\tau \frac{dI_s}{dt} = -I_s + \sum_{b=1}^{N} w_b u_b \qquad (1)$$

where $\tau$ is the decay time constant of the synaptic conductance of AMPA glutamate receptors, $w_b$ is the synaptic weight of neuron b, $N$ is the number of neurons, $u_b$ is the firing rate of corticofugal neurons. $\tau$ was estimated to be 3.9 ms based on AMPA receptor mediated EPSC recorded from NOT-DTN neurons on slices when corticofugal axons were optogenetically stimulated (data taken from Liu et al.[2]), consistent with a previous report[94]. $w_b$ was arbitrarily set to be 0.005 for all neurons. 100 neurons ($N$) were used in deriving synaptic current. The number of neurons in each of the four groups (TN bias, UP bias, NT bias, DN bias) was calculated based on the histogram of preferred directions. The charge of cortical input was calculated by integrating synaptic current $I_s$ in a 3.5 s time window starting from the onset of visual stimuli. The percentage change in the *charge* of cortical input ($\Delta C$) was calculated as

$$\Delta C = \frac{(C_{post} - C_{pre})}{C_{pre}} \qquad (2)$$

where $C_{pre}$ and $C_{post}$ are the charge before and after OKR potentiation, respectively.

## Data analysis of neuronal activity

Suite2p (python version, v0.7.1)[93] was used to analyze two-photon imaging data in multiple steps. Motion artifacts in the horizontal plane were corrected frame by frame based on the calcium-insensitive tdTomato fluorescence. Individual neurons, defined as regions of interest (ROIs), were delineated. The GCaMP6s fluorescence was measured by averaging all pixels in a given ROI. 70% of the neuropil signal was subtracted from the fluorescence of one ROI to remove the contamination from the neuropil. Custom written codes in MATLAB (Matlab R2018a, Mathworks) were used to quantify calcium responses. The motion artifact in the vertical axis was first corrected based on the change in calcium-insensitive tdTomato fluorescence. Calcium signals in response to visual stimuli were expressed as:

$$\frac{\Delta F}{F_0} = (F_I - F_0)/F_0 \qquad (3)$$

where $F_I$ is the instantaneous fluorescence signal and $F_O$ is the baseline fluorescence calculated as the mean fluorescence over the 1 sec prior to the visual stimulation. We removed outlier trials in which the amplitude of calcium signals is more than the upper quartile or less than the lower quartile by at least 1.5x interquartile range. Calcium responses were averaged across fifteen repetitions and the amplitude of average responses during the visual stimulation was used to derive tuning curves of visual feature selectivity. Two criteria were used to define responsive neurons: (1) their maximum response amplitudes are at least 6%; (2) the amplitudes of responses evoked by preferred stimuli are significantly higher than those in blank trials with a grey screen presented (two-sample t-test, $P < 0.05$).

To isolate units from extracellular recording, spike waveforms from raw data were sorted and clustered using a spike-sorting algorithm (kilosort2: https://github.com/MouseLand/Kilosort)[95]. Then, clusters were manually merged, split, and cleaned with phy 2.0 beta 1 (https://github.com/cortex-lab/phy/) based on the similarity in the waveform, correlogram, channel position, and visual feature selectivity. Clusters with refractory period violations of less than 0.2% were considered as single units. The multi-unit activity was defined as all spiking events exceeding the detection threshold after the removal of electrical noise or movement artifact by the sorting algorithm. All units were assigned a depth according to the electrode sites where the amplitudes of their spikes were the largest. A strong temporo-nasal preference of multi-unit activity was used to determine the boundary of the NOT-DTN. Only single units located within the NOT-DTN were used in the analysis. For both single-unit activity and multi-unit activity, the visual response was computed as the mean firing rate during visual stimulation after baseline subtraction. Responsive NOT-DTN units were identified when their firing rates evoked by preferred stimuli were significantly higher than their spontaneous activity (paired t-test, $P < 0.01$). The onset latency of optogenetically triggered antidromic spikes of IO-projecting NOT-DTN neurons was determined as the time lag between the beginning of the LED illumination and the time point at which the evoked firing rate reached 3-fold the standard deviation of spontaneous activity. Trial-by-trial jitter of the antidromic spikes was calculated as the standard deviation of the spike timing.

The preferred directions of individual neurons were determined in two ways. For orientation-selective neurons (OSI ≥ 0.1, Supplementary Fig. 2c left), we first determined the preferred orientation using the argument of a response-weighted vector summation of all orientations:

$$\theta_{pref} = \arg\left( \sum_{k} R(\rho_k) \times e^{2i\rho_k} \right) \qquad (4)$$

where $R(\rho_k)$ is the response amplitude to the kth direction $\rho_k$. Then, at the sample orientation of drifting grating (0°, 45°, 90°, 135°) closest to the preferred orientation, the direction which evoked a stronger calcium response was defined as the preferred direction. For the neurons that were not orientation selective (Supplementary Fig. 2c right), we calculated the preferred direction with the argument of the response-weighted vector summation of all directions:

$$\rho_{pref} = \arg\left( \sum_{k} R(\rho_k) \times e^{i\rho_k} \right) \qquad (5)$$

The direction opposite to the preferred direction was defined as the null direction. The sample direction of the drifting grating (0°, 45°, 90°, 135°, 180°, 225°, 270°, 315°) closest to the preferred direction was used to calculate the direction selectivity (DSI):

$$DSI = \frac{(R_{pref} - R_{null})}{(|R_{pref}| + |R_{null}|)} \qquad (6)$$

where $R_{pref}$ is the response amplitude to the preferred direction, and $R_{null}$ is the response amplitude to the null direction. The denominator is the summation of absolute values of $R_{pref}$ and $R_{null}$, since some visually-driven activities are negative. For calculating DSI of single units, evoked firing rates were used.

For the spatial frequency or temporal frequency tuning, we defined the preferred spatial frequency or temporal frequency as the one that evoked the highest response amplitude (Supplementary Fig. 3b, f). The spatial frequency or temporal frequency selectivity was calculated as

$$\text{Spatial or temporal frequency selectivity} = \frac{(R_{pref} - R_{null})}{(R_{pref} + R_{null})} \qquad (7)$$

where $R_{pref}$ is the response amplitude to preferred spatial or temporal frequency, and $R_{base}$ is the average of two minimum response amplitudes.

## Density maps of NOT-DTN-projecting or midbrain-projecting PNs

Confocal images of coronal slices containing the visual cortex were overlaid onto standard coronal slices of the Allen mouse brain atlas. ImageJ was used to count the number of fluorescent L5 PNs in individual visual areas and record their distance from the midline (that is the ML coordinate). The AP coordinate of each coronal slice was estimated based on the Allen mouse brain atlas. To calculate the volume density of individual areas, we divided the total number of labeled neurons in a visual area by the volume of the area. To draw a density map of corticofugal neurons, the whole visual cortex was divided into a grid of 100μm x 100μm squares, and area density was calculated for each square. Since the boundaries of the visual area on the surface were used in the density map, labeled L5 PNs were projected onto the pial surface along an axis perpendicular to the surface. The map of density ratio (Supplementary Fig. 4b) was the ratio between the density of NOT-DTN-projecting PNs and that of midbrain-projecting PNs, calculated pixel by pixel:

$$\text{Density ratio} = \frac{D_{NOT-DTN}}{D_{midbrain}} \qquad (8)$$

where $D_{NOT\text{-}DTN}$ was the density of NOT-DTN-projecting PNs, and $D_{midbrain}$ was the density of midbrain-projecting PNs.

## Data analysis of OKR behavior

To quantify OKR gain, eye movements in the horizontal axis were analyzed with custom-written codes running in MATLAB (Matlab R2018a, Mathworks). Saccade-like fast eye movements were detected as surges in the velocity curve (the temporal derivative of the eye position) and replaced by linear interpolation. Consequently, only slow eye movements were used to quantify the OKR behavior. Then, we used the Fourier transform of the trajectories of eye movements to derive the amplitudes of the OKR behavior. The OKR gain was calculated as

$$\text{OKR gain} = \frac{Amp_{eye}}{Amp_{drum}} \qquad (9)$$

where $Amp_{eye}$ is the amplitude of eye movement, and $Amp_{drum}$ is the amplitude of the drum grating movement. Thus, the OKR gain equals to 1 if the eye perfectly tracks the trajectory of drum grating and equals to 0 if it does not.

The cortical contribution to OKR gain was defined as the percentage reduction in OKR gain resulting from cortical silencing and calculated as

$$\triangle V(\%) = \frac{(V_{control} - V_{silencing})}{V_{control}} \qquad (10)$$

where $V_{control}$ and $V_{silencing}$ are the values of the OKR gain measured under control conditions or during optogenetic cortical silencing, respectively. Cortical contribution to OKR gain was computed and included in the summary for OKR gain of at least 0.02.

OKR potentiation was quantified as the amount of change in OKR gain and calculated as

$$\text{OKR potentiation} = \frac{(V_{\text{post}} - V_{\text{pre}})}{V_{\text{pre}}} \quad (11)$$

where $V_{\text{pre}}$ and $V_{\text{post}}$ are the values of OKR gain before and after the induction of OKR potentiation.

Cortical plasticity index was calculated as

$$\text{Cortical plasticity index} = \frac{(R_{\text{post}} - R_{\text{pre}})}{(R_{\text{post}} + R_{\text{pre}})} \quad (12)$$

where $R_{\text{pre}}$ and $R_{\text{post}}$ are the amplitudes of calcium responses before and after the induction of OKR potentiation.

Response change to temporo-nasal direction (cortical plasticity strength) was calculated as

$$\triangle R = \frac{(R_{\text{post}} - R_{\text{pre}})}{R_{\text{pre}}} \quad (13)$$

where $R_{pre}$ is the response amplitude to the temporo-nasal direction before OKR potentiation, and $R_{post}$ is the one after OKR potentiation. Then the neurons were categorized by their direction preference and the cortical plasticity strength of each population was calculated by averaging ΔR.

### Statistical analysis

Statistical analyses were done with the statistics toolbox in MATLAB (R2018a, Mathworks). All error bars were presented as mean ± s.e.m. unless otherwise noted. For small data sizes or the data that did not meet assumptions of parametric tests, statistical significance was assessed with Wilcoxon signed rank test or Wilcoxon rank sum test unless otherwise noted. Randomization test was used for identifying the biased distribution of direction preference. Kolmogorov-Smirnov test was used for testing whether the SF or TF preference distribution of the NOT-DTN-projecting population and midbrain population come from the same distribution. To perform linear regression with the maximum likelihood estimation (Fig. 4f, **orange line**), R function 'mle2' in bbmle package was used (R version 3.6.1). To perform linear regression with the minimum vertical distance of individual data points to the fitting line (Fig. 4f, **green line**), MATLAB function 'fminsearch' was used.

### Reporting summary

Further information on research design is available in the Nature Portfolio Reporting Summary linked to this article.

### Data availability

All data analyzed for this study are presented in the article, supplementary figures, and source data. The Allen Mouse Brain Atlas (https://atlas.brain-map.org/) was used to determine the boundaries of visual areas. Paxinos and Franklin's *The Mouse Brain in Stereotaxic Coordinates* (Elsevier, 2012) was used to determine the boundaries of subcortical structures. Because the raw and pre-processed datasets that support the findings of this study within the article and its supplementary materials are huge and presented in highly diverse formats, they are available from the corresponding author upon request. Source data are provided with this paper.

### Code availability

Custom codes used in data analysis and modeling are available on a publicly available repository GitHub at https://github.com/liulabutm/Codes-of-corticofugal-paper.git.

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

## Acknowledgements

We are thankful to A. Resulaj for sharing the *VGAT-ChR2-EYFP* line, to G. Keller for sharing RV-GCaMP6s virus, to D. Li for advice on statistics, to Y. Peng for advice on the characterization of RGC morphology, to S. Chen, A. Resulaj, M. Scanziani, M. Xue and S. Harris for providing feedback on the manuscript, to M. Cheng, J. Levine, X. Yin, and members of the Liu and Resulaj laboratories for advice during the course of the study, to P. Duggan and M, Szreder for machining and electronic engineering. We thank the UTM Imaging Facility and K. Harris-Howard for the use of confocal microscopes. This work was supported by grants from the Canadian Foundation of Innovation and Ontario Research Fund (CFI/ORF project no. 37597, B.L.), NSERC (RGPIN-2019-06479, B.L.), CIHR (Project Grant 437007, B.L.), Connaught New Researcher Awards (B.L.), and Ontario Graduate Scholarships (A.L.).

## Author contributions

J.L. and B.L. designed the study. J.L. conducted all experiments and data analysis except the electrophysiological recordings. Y.H. performed and analyzed electrophysiological recordings. A.L. performed the anterograde trans-synaptic tracing experiment. J.L., G.B. and B.L. wrote the paper.

## Competing interests

The authors declare no competing interests.
