## [Peer Review File · Nature Communications]

A Direction-selective Cortico-Brainstem Pathway Adaptively Modulates Innate BehaviorsEditorial Note: Parts of this Peer Review File have been redacted as indicated to remove third-party material where no permission to publish could be obtained.

REVIEWER COMMENTS

Reviewer #1 (Remarks to the Author):

General comments:

This is an impressive study applying optogenetic tools, electrophysiological and 2photon calcium imaging recordings to the mouse optokinetic system to confirm earlier studies from primates and to report a number of important additional original findings. In my opinion the authors should modify their statement in introduction (p.2 line 15-18) „...it remains unknown whether individual corticofugal channels to different brainstem nuclei each carry functionally specialized signals that match the properties of corresponding innate behaviors“ and (p 2-3 line 29 ff) „However, the direction preference of these corticofugal projections remains to be determined. “ to acknowledge published findings in primates by Distler et al JCN 2002, Hoffmann et al. JNP 2002, Hoffmann et al. EJM 2009). It is well established in primates that response properties of NOT-DTN projecting PNs resemble those of its downstream target neurons and for a comparative neurobiologist it is not too surprising that it is true also in the mouse. The authors also express this view by their statement at the end of discussion „the area specific connectivity between the visual cortex and the NOT-DTN may stand for a circuit feature of corticofugal projections shared by mammalian species“

Building on this basic finding the authors provide compelling evidence that

- 1) NOT-DTN projecting cortical layer V pyramidal cells have an area specific origin
- 2) become more responsive following OKR potentiation, with the maximum change observed in the temporo-nasally biasing population.
- 3) the IO projecting NOT-DTN population is a key node that integrates the retinal signals and cortical signals of retinal slips, and is essential for the adaptive plasticity of OKR.

I strongly recommend the publication of this study in Nature Comm. after appropriate revision.

Specific comments

P 2 Line 2 primitive ? better: phylogenetically older

P 2 Line 5-7 may be you could add one of the first findings in this respect. Wickelgren & Sterling's paper on the effects of cortical lesions on direction selectivity in the superior colliculus in the cat JNP 1969

P2 | 16-18 and 30 ff These statements should be corrected to acknowledge published findings ; see Hoffmann et al 2002, 2009

P 2 Line 29 why only gratings. Random dot patterns are equally or even more effective

P3 line 6-7 replace amplitude by gain

P3 | 17 this principle was already demonstrated in ref 2

P3 | 24 not surprising if you only stimulated in that direction

P4 | 5 was first described by Hoffmann & Schoppmann 1975

Fig 1 less could be more , difficult to appreciate the miniature figures. Omit schemes in a1 and a3

P 4 | Comparing panels a1-3 and b1-3 in fig 1 shows that cortical NOT-DTN projecting neurons are significantly more responsive to non t-n directions than NOT-DTN neurons. Is this due to a sampling bias in your NOT-DTN recordings?

P 7 fig 3. To really assess the cortical contribution to OKN quantitatively open loop OKN gain should be measured. Also monocular OKN would be more informative.

I have some difficulty understanding why stimulation was always binocular (maybe I missed some methodological detail). I do not know how large the binocular visual field is in mice. Even if the monitor was in „front“ of one eye the other would also be stimulated. Of course temporo-nasal in one eye would be naso-temporal in the other. Are NOT-DTN neurons to some extent binocular? Please clarify. Ipsiversive versus contraversive would be the better terms especially in the OKR measurements.

Fig 4f The real line of best fit (2 dimensional regression or shortest distance to each dot) would give a better impression than the simple regression line

P 9 | 14 I am astonished that you did not see a projection to the nucleus prepositus hypoglossi like shown in the cat? Magnin et al Visual Neuroscience , Volume 3 , Issue 1 , July 1989 , pp. 53 - 58

P9 | 22-24 Do I understand correctly, only IO projecting NOT-DTN (as a fraction of NOT-DTN

cells) receive cortical PN input. That would be very different from the monkey where almost all NOT-DTN neurons were activated by electrical stimulation of the visual cortex. However almost all NOT-DTN cells could be antidromically identified from the inferior olive

P 10 NOT-DTN PN have many non T-N neurons. Which are their target cells

P 10 | 23 ff Do the IO NOT-DTN neurons provide nothing to OKN gain ? Doesn't lesioning of the dorsal cap reduce OKN gain?

P 13 | 3 comparing fig 1 a3 and b3 the PNs have a much broader range of preferred directions. Which are the properties of PNs innervating the IO projecting NOT-DTNs ?

P29 | 18 frontal monitor ?? 12 directions

Reviewer #2 (Remarks to the Author):

Summary:

Previous work carried out by Liu et al, (2016) established that projections from the mouse visual cortex to the accessory optic system promote the adaptive plasticity of the optokinetic reflex (OKR) that is observed in response to vestibular impairment. Such plasticity relies on an enhanced drive exerted by the visual cortex onto the accessory optic system. Here, in a similar vein, using sophisticated techniques including single-unit recordings and cell-specific viral labeling strategies the properties of the OKR circuit are further investigated.

First, it is demonstrated that the visual cortical neurons (V1 and HVA) that project to the NOT-DTN, an area that is known to be activated preferentially by temporal-nasal motion, have similar tuning properties as their targets. Silencing the visual cortex (posterior more than the anterior parts) has a subtle but statistically significant effect on the OKR gain.

Forty-five-minute OKR training sessions enhance the direction selectivity of projection neurons as well as the OKR gain, suggesting the potential for causal linkage.

Further characterization of the cortical-recipient NOT-DTN neurons shows that they are more biased for temporo-nasal motion compared to the general population of NOT-DTN neurons. Moreover, these cortical-recipient neurons selectively project to the inferior olive (IO), rather than other known NOT-DTN targets, and silencing them impacts OKR.

In summary, this study uncovers various aspects of a circuit pathway from VC/HVA->

NOT/DTN-> IO, showing that it is dedicated to carrying information about temporal-nasal motion, and could modulate the OKR.

Major comments

This study represents a careful and thorough investigation of the OKR circuit with cutting-edge tools. The viral/optical approaches used here enable these authors to realize the tuning properties of select circuits embedded in densely populated regions of the brain, which could not be accomplished by conventional methods. However, there are several points that need to be addressed, which I feel would significantly increase the impact of this study.

- One of the main claims of this study is that the cortical-NOT-DTN-IO circuitry is important for OKR gain control. However, in the presence of the vestibular ocular reflex, cortical modulation of OKR is marginal, as shown by these authors previously (Liu et al.,). Indeed, the effects of silencing VC/or NOT/DTN on OKR gain is tiny (~2 %? Fig. 3e and Fig. 6d). This subverts the impact of the study.
- Given the effects of silencing are small, it is worth performing adequate control experiments. Describing the effects of silencing VC/NOT-DTN when the direction of the grating is reversed is an important control that is missing. With the drifting grating direction reversed, the opposite DTN/NOT should be engaged and unilateral silencing should no longer be effective. Showing the directional dependence of silencing would enable behaviors to be compared in individual mice and would be much more compelling.
- The authors nicely demonstrate large related changes in OKR gain and cortical responsivity after 45-minute OKR training. It would be important to show to what extent changes in OKR gain occur when this circuit is silenced at different levels.
- In Figure 4, they show that after OKR potentiation, the cortical neurons are also potentiated, and conclude that: "NOT-DTN projecting PNs are a functionally distinct population in promoting OKR potentiation". This experiment doesn't suggest that PNs promote potentiation. It is still possible that the opposite is true, i.e. that there is some mechanism by which the OKR potentiates the PN responses. In this experiment (Fig. 4) the effect size is larger (Fig. 4d), which to me might suggest that the PN to IO circuit isn't so important for establishing the increase in gain that comes with changing spatial frequency (i.e. small effect in Fig. 3e), but is more important for potentiating responses around that set point.

- It would be helpful if the authors clearly state what are the significant advances being made in this study. How is the top-down modulation of the DTN-NOT different from what was shown earlier by Liu et al., (2016)? Given that cortical control of OKR gain is weak with the vestibular system intact, the authors should discuss what the major contributor to the OKR gain is.

Reviewer #3 (Remarks to the Author):

In their study, Liu et al. describe the organization and function of a direction-selective cortico-brainstem pathway in mice. The authors find that corticofugal layer 5 pyramidal neurons (L5 PN) labeled retrogradely from NOT and DTN, two brainstem nuclei that prefer temporo-nasal (TN) visual motion and control horizontal gaze-stabilizing eye movements via the optokinetic reflex (OKR), share the direction preference and strong selectivity of their targets. Liu et al. show that L5 PNs providing input to NOT/DTN are enriched in anterior primary visual cortex (V1) and posterior higher visual areas (HVAs). Consistent with this enrichment, they demonstrate that optogenetic silencing of posterior HVAs reduces the OKR gain more than optogenetic silencing of anterior HVAs. Next, Liu et al. discover that the OKR potentiation observed after prolonged visual stimulation is correlated with an increase in the responses of NOT/DTN-projecting L5 PNs, particularly those preferring TN motion. Neurons in NOT/DTN project to several downstream targets. Liu et al. show that the DTN/NOT neurons that receive input from L5 PNs selectively innervate one of these, the inferior olive (IO). All IO-projecting NOT/DTN neurons prefer TN motion. Liu et al. pharmacogenetically suppress the IO-projection NOT/DTN neurons, which cancels the impact of optogenetically silencing visual cortex on the OKR gain. Together these findings indicate that visual cortex modulates the OKR gain through a direction-selective (TN) pathway from visual cortex via NOT/DTN to IO. Finally, the authors use a computational model to explore how directional bias and response enhancement of L5 PNs may drive OKR potentiation through the downstream pathway.

Overall, this exciting study reveals the organization and function of a direction-selective pathway from visual cortex via NOT/DTN to IO that modulates a highly conserved gaze-stabilizing reflex. The authors combine an impressive array of sophisticated techniques, and

the quality of the data and their analysis is exceptional. The results are presented clearly in the figures and the text. Addressing the following specific comments would help to clarify a few important points.

Specific comments

1.) Do retinal ganglion cells and L5 PNs converge onto IO-projecting NOT/DTN neurons? The discussion suggests they do. Has this been shown in mice? If so, what is the spatial and cell-type distribution of the retinal ganglion cells, and does it align with the retinotopic position of the L5 PNs in the anterior V1? A retrograde tracing strategy could address these questions (e.g., injecting a retrogradely transported helper AAV into IO followed by injection of a rabies virus into NOT/DTN).

2.) The authors show that pharmacogenetic silencing of IO-projecting NOT/DTN neurons suppresses the impact of cortical silencing on the OKR gain. How does silencing the IO-projecting NOT/DTN neurons alone affect the OKR gain and its stimulus-drive plasticity?

3.) The authors show that OKR plasticity is accompanied by enhanced responses in TN-preferring DTN/NOT-projection L5 PNs. They show that both phenomena require patterned visual stimulation (i.e., a horizontally oscillating grating vs. a gray screen). Are both phenomena motion axis selective, or do vertically oscillating gratings have the same effects, does the stimulus selectivity of the two phenomena diverge for vertically oscillating gratings?

To all reviewers:

We would like to thank the reviewers for their helpful comments on the manuscript. We made major revisions of the manuscript in the following aspects:

1. We have combined the rabies transsynaptic tracing and 2-photon (2p) calcium imaging to examine the direction selectivity of corticofugal neurons that innervate IO-projecting NOT-DTN neurons (**page 12, line 19-28**).
2. With rabies transsynaptic tracing, we have examined the heterogeneous distributions of the corticofugal neurons and retinal ganglion cells (RGCs) that innervate IO-projecting NOT-DTN neurons. We also analyzed the dendritic morphology of those RGCs to identify their cell types (**page 11, line 14-32; page 12, line 1-4**).
3. With optogenetics, we have examined the effects of unilaterally silencing visual cortex on unidirectional OKR (ipsiversive vs contraversive with respect to the side of cortical silencing; **page 8, line 15-31; page 9, line 1-12**).
4. With optogenetics, we have examined the roles of higher visual areas in OKR potentiation (**page 8, line 2-13**).
5. With optogenetics and chemogenetics, we have examined the role of the NOT-DTN to IO projection pathway in OKR and OKR plasticity (**page 13, line 23-27; page 14, line 1-7**).
6. We have investigated the effects of continuously rotating a horizontal drum grating in the vertical direction on horizontal OKR behavior and the responsiveness of NOT-DTN-projecting neurons to temporonasal motion (**page 10, line 9-11**).
7. We have revised the manuscript (highlighted in green) and modified figures to address reviewers' comments and to integrate new data into the story.

Here are our replies to reviewers' comments.

Reviewer #1 (Remarks to the Author):

General comments:

This is an impressive study applying optogenetic tools, electrophysiological and 2photon calcium imaging recordings to the mouse optokinetic system to confirm earlier studies from primates and to report a number of important additional original findings.

1. *"In my opinion the authors should modify their statement in introduction (p.2 line 15-18) „...it remains unknown whether individual corticofugal channels to different brainstem nuclei each carry functionally specialized signals that match the properties of corresponding innate behaviors“ and (p 2-3 line 29 ff) „However, the direction preference of these corticofugal projections remains to be determined. “ to acknowledge published findings in primates by Distler et al JCN 2002, Hoffmann et al. JNP 2002, Hoffmann et al. EJN 2009). It is well established in primates that response properties of NOT-DTN projecting PNs resemble those of its downstream target neurons and for a comparative neurobiologist it is not too surprising that it is true also in the mouse. The authors also express this view by their statement at the end of discussion „the area specific connectivity between the visual cortex and the NOT-DTN may stand for a circuit feature of corticofugal projections shared by mammalian species“"*

We thank the reviewer for the comments and suggestions. The two monkey studies (*Hoffmann et al., J Neurophysiol 2002; Hoffmann et al., Eur J Neurosci 2009*) indeed showed that MT and MST neurons projecting to the NOT-DTN are biased to temporo-nasal motion. We modified the introduction to acknowledge these two studies and to introduce the remaining question properly (**page 2, line 15-18; page 2, line 30-32; page 3, line 1-4**). Please see the detailed discussion in comment 4.

Building on this basic finding the authors provide compelling evidence that

1) NOT-DTN projecting cortical layer V pyramidal cells have an area specific origin

2) become more responsive following OKR potentiation, with the maximum change observed in the temporo-nasally biasing population.

3) the IO projecting NOT-DTN population is a key node that integrates the retinal signals and cortical signals of retinal slips, and is essential for the adaptive plasticity of OKR.

I strongly recommend the publication of this study in Nature Comm. after appropriate revision.

Specific comments:

2. “P 2 Line 2 primitive ? better: phylogenetically older”

Following reviewer’s suggestion, we have changed the wording accordingly (**page 2, line 3**).

3. “P 2 Line 5-7 may be you could add one of the first findings in this respect. Wickelgren & Sterling’s paper on the effects of cortical lesions on direction selectivity in the superior colliculus in the cat JNP 1969”

Following reviewer’s suggestion, we have added this paper in the citation. (**page 2, line 7, ref# 11**).

4. “P2 l 16-18 and 30 ff These statement should be corrected to acknowledge published findings ; see Hoffmann et al 2002, 2009”

We thank the reviewer for the comment. First of all, in the first submission, we specifically meant that these 2p functional imaging studies did not answer whether the corticofugal projections to different brainstem nuclei send functionally specialized signal. We revised the sentence to clarify it (**page 2, line 15-18**). Second, we agree with the reviewer that the two monkey studies (*Hoffmann et al., J Neurophysiol 2002; Hoffmann et al., Eur J Neurosci 2009*) provided valuable insight into functional specificity of corticofugal projections. However, the technique of antidromic stimulation, which was used in these studies to identify cortical neurons projecting to the NOT-DTN, can non-specifically activate axons of passage that innervate non-NOT-DTN structures. Moreover, NOT-DTN-projecting neurons in other visual areas or in other species, and the brainstem neurons downstream of the corticofugal projection have not been systematically examined yet. Therefore, the answer to the question of to what extent the response properties of the visual corticofugal projections match those of their downstream targets in the NOT-DTN remains incomplete. Please see the revision in **page 2, line 30-32 and page 3, line 1-4**.

5. “P 2 Line 29 why only gratings. Random dot patterns are equally or even more effective”

We agree with the reviewer that random dot patterns are also effective to activate both V1 and higher visual areas, as shown in *Samonds et al., J Neurosci 2019 and Sit & Goard, Nat Commun 2020*. We added it to the sentence in **page 2, line 30**.

6. “P3 line 6-7 replace amplitude by gain”

Done. We have replaced ‘amplitude’ by ‘gain’ in **page 3, line 7-8**.

7. “P3 | 17 this principle was already demonstrated in ref 2”

This principle has not been directly demonstrated by ref 2 yet (*Liu et al., Nature 2016*). This previous study showed that upon the induction of OKR potentiation, the impact of visual cortex on the activity of NOT-DTN neurons (evaluated by silencing the visual cortex) became stronger and this cortical drive of NOT-DTN activity could explain the cortical contribution to the behavioral change in OKR potentiation. Although this study demonstrated the necessary role of the corticofugal projection in OKR potentiation, it did not examine how the corticofugal projection increased the cortical drive of NOT-DTN neurons. In theory, the enhanced cortical impact can result from several possible mechanisms, including, but not limited to i) an increase in the firing of NOT-DTN-projecting corticofugal neurons, ii) a strengthening of the synapse between corticofugal axons and NOT-DTN neurons or iii) a decrease in feed-forward inhibition received by NOT-DTN neurons. Since multiple possibilities exist, we cannot deduce a conclusion favoring the first mechanism, enhanced responsiveness of corticofugal neurons. Thus, in the study we decided to experimentally test this mechanism. We chose this mechanism rather than others, because it fits the theme of this study, the response properties of the corticofugal neurons, and because there is compelling evidence of cortical plasticity across supragranular and infragranular layers, including layer 5 where corticofugal neurons reside (*Schoups et al., Nature 2001; Gilbert & Li, Neuron 2012; Hofer et al., Nature 2009; Diamond et al., Science 1994; Asokan et al., Nat Commun 2018*). We revised the manuscript accordingly (**page 3, line 14-17; page 9, line 16**).

References not cited in the paper:

Hofer et al., Experience leaves a lasting structural trace in cortical circuits. Nature 457(7227):313-7 (2009).

Diamond, M. E., Huang, W. & Ebner, F. F. Laminar comparison of somatosensory cortical plasticity. Science 265, 1885-1888 (1994).

Asokan et al., Sensory overamplification in layer 5 auditory corticofugal projection neurons following cochlear nerve synaptic damage. Nat Commun 9(1):2468 (2018).

8. “P3 | 24 not surprising if you only stimulated in that direction”

In response to the reviewer’s comment, we replaced ‘surprisingly’ with ‘interestingly’ (**page 3, line 27**). It is worth noting that in the protocol of inducing OKR potentiation, the drum grating alternated between temporo-nasal and naso-temporal directions. However, the plasticity occurred only in the population that prefers the temporo-nasal direction, but not in the one preferring naso-temporal direction. Our finding indicates that the plasticity depends on not only the directions of visual motion, but also the functional relevance of the neuronal population. Therefore, in some senses it is a surprise to see a bidirectional motion induced the plasticity exclusively in the NOT-DTN-projecting PNs population preferring one motion direction, the temporo-nasal.

9. “P4 | 5 was first described by Hoffmann & Schoppmann 1975”

Done. We have added this paper to the citations in **page 4, line 10, ref# 25**.

10. “Fig 1 less could be more , difficult to appreciate the miniatur figures. Omit schemes in a1 and a3”

We agree with the reviewer that old figure 1 had too many panels to be appreciated. Following reviewer’s suggestion, we removed the schemes in a1 and a3. To make the figure clearer, we further merged the histograms in a3 and b3 into the new panel d.

11. “P 4 | Comparing panels a1-3 and b1-3 in fig 1 shows that cortical NOT-DTN projecting neurons are significantly more responsive to non t-n directions than NOT-DTN neurons. Is this due to a sampling bias in your NOT-DTN recordings?”

The reviewer has raised an excellent point. First, it is unlikely that this difference in direction selectivity between NOT-DTN neurons and NOT-DTN-projecting cortical neurons resulted from a sampling bias in NOT-DTN recordings, because the recordings were done from various locations and depths in the NOT-DTN (see images below).

Legend: Example coronal brain slices containing the NOT-DTN, which came from *in vivo* NOT-DTN recordings. Slices are arranged according to their anterior to posterior locations in the NOT-DTN (Left, anterior; right, posterior). Red dashed line, boundary of NOT-DTN. Note that the electrode tracks were labeled in red.

Second, such difference is neither accounted for by the possibility that retrograde-Cre virus may infect nonspecifically the axons of passage which do not innervate the NOT-DTN. This nonspecific infection would contaminate the dataset of NOT-DTN-projecting cortical neurons with midbrain-projecting ones, which has no temporo-nasal bias. To rule out this possibility, we used the rabies retrograde transsynaptic labeling to express GCaMP6s exclusively in the visual cortical neurons that innervate the IO-projecting NOT-DTN neurons (NOT-DTN-innervating PNs, **Fig. 5j**). Using 2p imaging, we observed a similar distribution of preferred direction between NOT-DTN-innervating PNs and NOT-DTN-projecting PNs; both populations had a substantial fraction of non-temporo-nasally biased neurons (**Fig. 5l**). Please see details in **page 12, line 19-28**.

Last, such difference was also observed in primates. Previous studies reported that in this higher mammalian species all NOT-DTN neurons were biased towards temporo-nasal motion (*Hoffmann & Distler, J Neurophysiol 1989; Ilg & Hoffmann, Eur J Neurosci 1996*), while some NOT-DTN-projecting cortical neurons in motion sensitive areas MT and MST preferentially responded to non-temporo-nasal motion (*Hoffmann et al., J Neurophysiol 2002; Hoffmann et al., Eur J Neurosci 2009*). Thus, we think that this difference in the degree of temporo-nasal bias between presynaptic cortical neurons and NOT-DTN neurons may be conserved in mammals.

We discussed this difference between NOT-DTN neurons and NOT-DTN-projecting PNs and the possible functional implication in **page 16, line 24-31** and **page 17, line 1-9**.

12. "P 7 fig 3. To really assess the cortical contribution to OKN quantitatively open loop OKN gain should be measured. Also monocular OKN would be more informative."

We agree with the reviewer's suggestions, since the open loop visual stimulation can keep the retinal slips at a constant speed and monocular OKR stimulation can control which side of the NOT-DTN and visual cortex are activated. Their combination allows a simpler experimental design and a higher explanatory power. Thus, we performed a series of experiments to examine the cortical contribution to OKR under those two conditions. Considering that the visual cortex projects exclusively to the ipsilateral NOT-DTN, we hypothesized that the visual cortex facilitates only the ipsiversive OKR, but not the contraversive OKR with respect to the side of cortical silencing. If this were proven true, it would support the idea that the cortical impact on the OKR is achieved by the ipsilateral corticofugal projection.

First, we evaluated the monocular OKR elicited by ipsiversive motion (equivalent to temporo-nasal) or by contraversive motion (equivalent to naso-temporal). Consistent with previous reports on rodent OKR, the gain

of monocular OKR in response to ipsiversive motion was much stronger than that to contraversive motion (gain ratio=20:1, **Supplementary Fig. 7c**). This asymmetry indicates that in mice the OKR is primarily driven by the eye seeing temporo-nasal motion and the contralateral NOT-DTN (**Supplementary Fig. 7a left**). This result is also consistent with the anatomy that RGC axons barely innervate the ipsilateral NOT-DTN (*Dhande et al., J Neurosci 2013*) and with the limited degree of cortical binocularity in mice (*Hübener, Curr Opin Neurobiol 2003*). In contrast, in primates both ipsiversive motion and contraversive motions effectively elicit monocular OKR (*Koerner & Schiller, Exp Brain Res 1972*). This symmetric OKR results from two factors: **i**) a substantial amount of ipsilateral retinal projection to the NOT-DTN (*Telkes et al., Eur J Neurosci 2000*), and **ii**) a high degree of binocularity in the visual cortex of primates (*LeVay et al., J Comp Neurol 1980*).

Second, we asked how the cortical contribution to OKR gain depends on the direction of grating motion. We examined the contribution of the left visual cortex to the binocular OKR elicited by either the ipsiversive or contraversive motions. As predicted by the anatomy of visual circuits (contralateral retinal input and ipsilateral cortical input, **Supplementary Fig. 7d left**) and the temporo-nasal direction bias of retinal (*Dhande et al., J Neurosci 2013*) and cortical inputs (**Fig. 1d**), silencing the visual cortex in one hemisphere reduced the OKR gain in response to ipsiversive motion ($27.5 \pm 1.84\%$, **Supplementary Fig. 7d right, f left**). Surprisingly, the same unilateral cortical silencing slightly increased the OKR gain in response to contraversive motion ($-11.8 \pm 3.34\%$, **Supplementary Fig. 7e right, f right**). This can be explained by **i**) the same anatomical connectivity of visual circuits, and **ii**) the weak activity of the NOT-DTN-projecting PNs elicited by the contraversive motion (**Fig. 1d**). Therefore, this direction-dependent cortical impact on the OKR supports the idea that the visual cortex modulates OKR behavior through its ipsilateral projection to the NOT-DTN (**Fig. 2a, Supplementary Fig. 4a**).

Last, we compared the cortical contribution to OKR gain under open-loop or closed-loop conditions. In open-loop OKR, the speed of retinal slips (the actual visual inputs driving the OKR) was kept constant by an engineering approach and in consequence the OKR gain became much higher than 1, which is consistent with previous reports (*Koerner & Schiller, Exp Brain Res 1972; Behrens et al., Prog Brain Res 1989*). In contrast, in the closed-loop OKR the speed of retinal slips was affected by the instantaneous speed of eye movement and changed moment by moment. Despite the difference between closed-loop and open-loop conditions, we found that the cortical contribution to OKR gain was similar under those two conditions ($17.0 \pm 1.7\%$ vs $16.7 \pm 2.8\%$, **Supplementary Fig. 7g-i**).

Please see details in the results (**page 8, line 15-31; page 9, line 1-12**) and discussion (**page 16, line 24-31; page 17, line 1-9**).

References not cited in the paper:

Hübener. Mouse visual cortex. Curr Opin Neurobiol 13(4):413-20 (2003).

Telkes, I., Distler, C. & Hoffmann, K. P. Retinal ganglion cells projecting to the nucleus of the optic tract and the dorsal terminal nucleus of the accessory optic system in macaque monkeys. Eur J Neurosci 12, 2367-2375 (2000).

LeVay, S., Wiesel, T. N. & Hubel, D. H. The development of ocular dominance columns in normal and visually deprived monkeys. J Comp Neurol 191, 1-51 (1980).

13. *"I have some difficulty understanding why stimulation was always binocular (maybe I missed some methodological detail). I do not know how large the binocular visual field is in mice. Even if the monitor was in „front“ of one eye the other would also be stimulated. Of course temporo-nasal in one eye would be naso-*

temporal in the other. Are NOT-DTN neurons to some extent binocular? Please clarify.”

The choice of binocular stimulation in this study is related to the induction protocol of OKR plasticity. Oscillating drum gratings had been shown to be an effective way to increase the OKR gain, and this experimental paradigm of OKR potentiation had been used by many labs to investigate the molecular, synaptic, and circuit mechanisms underlying oculomotor learning (Katoh et al., PNAS 1998; Faulstich et al., Vision Res 2004; Liu et al., Nature 2016; Shutoh et al., Euro J of Neurosci 2003; Takeuchi et al., Plos one 2008). Therefore, we decided to use this well-established method to evaluate OKR and to induce OKR potentiation. It is worth noting that for a binocularly viewing mouse, in any moment, only the eye that sees temporo-nasal motion and the corresponding contralateral NOT-DTN drive the OKR effectively (**Supplementary Fig. 7d left**). Its OKR works in this way because the NOT-DTN neurons in mice are exclusively innervated by temporo-nasally biased RGCs in the contralateral eye (Dhande et al., J Neurosci 2013, **Supplementary Fig. 7a left, b left**). When the oscillatory drum grating reverses, the counterpart eye and NOT-DTN drive the OKR in the opposite direction (**Supplementary Fig. 7e left**). In other words, the two sets of eye-NOT-DTN systems take turns mediating the OKR in one of the two phases of drum motion, clockwise or counterclockwise. If we present an oscillating drum grating to monocularly viewing mice with one eye closed, due to the strong asymmetry of monocular OKR (**Supplementary Fig. 7a-c**), the eye trajectory will substantially drift towards the nose, which complicates the analysis in the amplitude of OKR eye movement and OKR gain (see the figure below).

Legend: Cycle-averaged trajectory of monocular OKR (black) elicited by an oscillating drum grating (its trajectory in green). The downward direction is temporo-nasal. Note that in comparison to [FIGURE REDACTED] trajectories of binocular OKR (Fig 3c, 4b, 6b), the eye did not move much during the phase when the drum moved naso-temporally (the flat middle part of the eye trajectory), which caused the eye to drift towards the nose.

References not cited in the paper:

Shutoh et al. Role of protein kinase C family in the cerebellum-dependent adaptive learning of horizontal optokinetic response eye movements in mice. *Eur J Neurosci.* 2003 18(1):134-42.

Takeuchi et al. Enhancement of both long-term depression induction and optokinetic response adaptation in mice lacking delphilin. *PLoS One.* 2008 3(5):e2297.

14. “Ipsiversive versus contraversive would be the better terms specially in the OKR measurements.”

Following the reviewer’s suggestion, we have now clarified the definitions of visual motion directions and standardized their usage throughout the paper. ‘temporo-nasal’ or ‘naso-temporal’ were used to describe the directions of the grating motion with respect to the right eye (for example, **page 2, line 24** and **page 8, line 19**), since all electrophysiological recording and 2p imaging were done from the left NOT-DTN or left visual cortex, which receive the primary input from the right eye. But if we used the NOT-DTN or the visual cortex as the reference, especially when unilateral circuit perturbation was involved, the ‘ipsiversive’ or ‘contraversive’ were chosen to describe the directions of the visual motion and eye movement (for example, **page 8, line 19** and **page 13, line 25**).

15. “Fig 4f The real line of best fit (2 dimensional regression or shortest distance to each dot) would give a better impression than the simple regression line”

Following the reviewer’s suggestion, we used the shortest distance of each dot to the regression line as the criterion to do the fitting (**green line in Fig 4f**). Moreover, we also used the maximum likelihood estimation to do the linear regression (**red line in Fig 4f**). Both results were added to the **Fig 4f**, in comparison to the original regression line with least squares method (**black line in Fig 4f**).

16. “P 9 | 14 I am astonished that you did not see a projection to the nucleus prepositus hypoglossi like shown in the cat? Magnin et al *Visual Neuroscience*, Volume 3, Issue 1, July 1989, pp. 53 – 58”

The reviewer is correct about the existence of NOT-DTN to NPH projection. Thanks to reviewer’s suggestion, we went back to examine the brain slices from the experiments in which the IO-projecting NOT-DTN neurons were labeled anterogradely (**Fig. 5a-b**), retrogradely (**Supplementary Fig. 9c-d**), or by direct somatic infection (**Supplementary Fig. 9a-b**). In all three experiments, we indeed found very sparse axons in the nucleus prepositus hypoglossi. We added confocal images of coronal slices containing NPH to **Fig. 5b**, **Supplementary Fig. 9b,d**. Please note that because those slices were prepared a few years ago, DAPI fluorescence had faded completely. We also modified the description of these experimental results accordingly (**page 11, lines 3, 7, 10**).

17. “P9 | 22-24 Do I understand correctly, only IO projecting NOT-DTN (as a fraction of NOT-DTN cells) receive cortical PN input. That would be very different from the monkey where almost all NOT-DTN neurons were activated by electrical stimulation of the visual cortex. However almost all NOT-DTN cells could be antidromically identified from the inferior olive”

Yes, our data indicate that only IO-projecting NOT-DTN neurons receive cortical input in mice. This conclusion is supported by our finding that the axons of NOT-DTN neurons that receive cortical innervation (labeled by anterograde transsynaptic tracing) were primarily located in the IO, but not in other downstream targets (**Fig. 5b**). This finding in mice may reflect a commonly shared principle with monkeys. Since in monkeys almost all NOT-DTN neurons receive cortical input (activated by electrical stimulation of the visual cortex) and project to the IO (antidromic stimulation), we can infer that the visual cortex in monkeys also innervates ‘only’ IO-projecting NOT-DTN neurons.

However, we do recognize the difference in NOT-DTN circuits between monkeys and mice. First of all, in mice the optogenetic activation of the visual cortex elicited activity in ~50% of NOT-DTN neurons and photostimulation of cortical axons triggered monosynaptic excitatory postsynaptic currents in ~20% of NOT-DTN neurons (Fig. 2c,d in *Liu et al. Nature 2016*). These findings indicate that a substantial number of NOT-DTN neurons do not receive cortical innervation. Second, neither all NOT-DTN neurons in mice project to the inferior olive. The inhibitory neurons in the NOT-DTN, which make up ~30% of the NOT-DTN neuronal population, do not project to the inferior olive (*Horn & Hoffmann, Brain Res 1987*, see the figure below). Moreover, in mice only a portion of NOT-DTN neurons (~10%) could be activated by antidromically stimulating their axons in the inferior olive (**Fig. 5g-i**). Taken together, although the proportions of NOT-DTN neurons receiving cortical innervation and projecting to the IO differ between mice and monkeys, both species may have a similar connectivity principle governing the corticofugal projection to the NOT-DTN.

[FIGURE REDACTED]

Legend: Inhibitory NOT-DTN neurons project to the superior colliculus, but not to the IO. Adapted from a figure in another our manuscript (He et al., in preparation).

References not cited in the paper:

Horn A.K., Hoffmann, K.P. Combined GABA-immunocytochemistry and TMB-HRP histochemistry of pretectal nuclei projecting to the inferior olive in rats, cats and monkeys. Brain Res. 409(1):133-8 (1987).

18. “P 10 NOT-DTN PN have many non T-N neurons. Which are their target cells”

In another ongoing study in my lab which concerns the functional roles of inhibitory population in the NOT-DTN (manuscript in preparation), we found that the inhibitory NOT-DTN population had a higher percentage of non-temporo-nasally biased neurons and many of these non-temporo-nasally biased inhibitory neurons projected to the superior colliculus (see the figure below). Therefore, at least some of the non-temporo-nasally biased NOT-DTN neurons project to the superior colliculus. Because these data are not directly related to the topic of this study and they have been used in another paper, we won't include them in this manuscript.

[FIGURE REDACTED]

Legend: Histograms of preferred direction of several different NOT-DTN populations: excitatory neurons, inhibitory neurons and inhibitory neurons projecting to the SC. Note that higher percentages of inhibitory neurons and SC projecting inhibitory neurons preferred non-temporo-nasal directions. Adapted from a figure in another our manuscript (He et al., in preparation).

19. “P 10 | 23 ff Do the IO NOT-DTN neurons provide nothing to OKN gain ? Doesn't lesioning of the dorsal cap reduce OKN gain?”

To answer this question, we took an optogenetic approach to silence the IO-projecting NOT-DTN neurons (**Fig. 6d**). We did not consider lesioning the dorsal cap of inferior olive because this structure receives convergent inputs from many places (*Armstrong, Physiol Rev 1974*) and lesioning does not allow projection-specific perturbation. We chose optogenetics instead of chemogenetics, because optogenetic tools allow the circuit to be silenced and unsilenced in interleaved trials. The rapidity and reversibility of optogenetic silencing control for the trial-by-trial variability of the OKR gain; and most importantly, they also control for potential drifts in OKR gain during long recording sessions, since it takes approximately 1.5 hours to generate a SF tuning curve (24 trials per stimulus condition, randomized throughout the session). Thus, we conditionally expressed inhibitory opsin ArchT in IO-projecting NOT-DTN neurons in the left hemisphere (**Fig. 6d**). We found that silencing this specific population reduced the gain of ipsiversive OKR, but did not affect contraversive OKR ($14.4 \pm 3.6\%$ vs $-0.07 \pm 1.5\%$, **Fig. 6f**). This result is consistent with the effects of unilateral cortical silencing (**Supplementary Fig. 7d-f**). In addition, we chemogenetically silenced the IO-projecting NOT-DTN neurons during the induction of OKR plasticity by continuous OKR stimulation (**Fig. 6g**). This circuit perturbation prevented the OKR potentiation (**Fig. 6h-i**). Overall, these results suggest that the NOT-DTN to IO projection plays a modulatory role, by fine-tuning the strength of OKR according to behavioral needs and visual experience. Furthermore, the abolishment of OKR potentiation upon silencing the NOT-DTN to IO pathway highlights its pivotal role in OKR plasticity. We have added the results of these two experiments to **page 13, line 23-27** and **page 14, line 1-7**.

References not cited in the paper:

Armstrong, D.M. Functional significance of connections of the inferior olive. Physiol Rev 54(2): 358-417 (1974)

20. “P 13 | 3 comparing fig 1 a3 and b3 the PNs have a much broader range of preferred directions. Which are the properties of PNs innervating the IO-projecting NOT-DTNs ?”

The reviewer raised a good point. Considering the stronger temporo-nasal bias of IO-projecting NOT-DTN neurons than the overall NOT-DTN population, the cortical neurons innervating the IO-projecting NOT-DTN neurons may have a stronger temporo-nasal bias than the cortical neurons projecting to the NOT-DTN. To examine this possibility, we utilized the rabies transsynaptic tracing to exclusively express GCaMP6s in cortical neurons that innervate IO-projecting NOT-DTN neurons (**Fig. 5j**). We found that this corticofugal population

had similar direction preference to NOT-DTN-projecting cortical neurons: more neurons were biased towards the temporo-nasal direction than other directions; still a substantial portion of neurons preferred non-temporo-nasal directions (29.4% for temporo-nasal direction vs 6.9-15% for the other 7 directions; **Fig. 5I**). And this population also had a direction selectivity index similar to that of the NOT-DTN-projecting cortical population (0.38 ± 0.02 , **Fig. 5m**). Thus, we conclude that the cortical input to the NOT-DTN has a weaker temporo-nasal bias than its postsynaptic neurons in the NOT-DTN, which is reminiscent of reports from monkey studies (*Hoffmann & Distler, J Neurophysiol 1989; Ilg & Hoffmann, Eur J Neurosci 1996; Hoffmann et al., J Neurophysiol 2002; Hoffmann et al., Eur J Neurosci 2009*). The fact that some non-temporo-nasally biased corticofugal neurons innervate the NOT-DTN might be perceived as suboptimal. However, it enables the visual cortex to suppress the OKR in the contraversive direction, in addition to facilitate ipsiversive OKR (**Supplementary Fig. 7d-f**). Therefore, the diverse direction bias of NOT-DTN-projecting PNs may allow animals to bidirectionally modulate innate behaviors. We have added the results of this experiment to **page 12, line 19-28**. We also discussed this topic in **page 16, line 24-31** and **page 17, line 1-9**.

21. "P29 I 18 frontal monitor ?? 12 directions"

We did extracellular recording from NOT-DTN neurons on the same OKR behavior rig to evaluate their direction, SF and TF tuning. When examining the direction selectivity, we presented the drifting gratings (12 sample directions) only on the frontal monitor of the 'virtual drum' apparatus (**Fig. 3a**), while keeping the left and right monitors in uniform luminance. We revised the method to clarify it (**page 36, line 24-26**).

Reviewer #2 (Remarks to the Author):

Summary:

Previous work carried out by Liu et al, (2016) established that projections from the mouse visual cortex to the accessory optic system promote the adaptive plasticity of the optokinetic reflex (OKR) that is observed in response to vestibular impairment. Such plasticity relies on an enhanced drive exerted by the visual cortex onto the accessory optic system. Here, in a similar vein, using sophisticated techniques including single-unit recordings and cell-specific viral labeling strategies the properties of the OKR circuit are further investigated.

First, it is demonstrated that the visual cortical neurons (V1 and HVA) that project to the NOT-DTN, an area that is known to be activated preferentially by temporal-nasal motion, have similar tuning properties as their targets. Silencing the visual cortex (posterior more than the anterior parts) has a subtle but statistically significant effect on the OKR gain. Forty-five-minute OKR training sessions enhance the direction selectivity of projection neurons as well as the OKR gain, suggesting the potential for causal linkage.

Further characterization of the cortical-recipient NOT-DTN neurons shows that they are more biased for temporo-nasal motion compared to the general population of NOT-DTN neurons. Moreover, these cortical-recipient neurons selectively project to the inferior olive (IO), rather than other known NOT-DTN targets, and silencing them impacts OKR.

In summary, this study uncovers various aspects of a circuit pathway from VC/HVA-> NOT/DTN-> IO, showing that it is dedicated to carrying information about temporal-nasal motion, and could modulate the OKR.

Major comments

This study represents a careful and thorough investigation of the OKR circuit with cutting-edge tools. The viral/optical approaches used here enable these authors to realize the tuning properties of select circuits

embedded in densely populated regions of the brain, which could not be accomplished by conventional methods. However, there are several points that need to be addressed, which I feel would significantly increase the impact of this study.

1. “One of the main claims of this study is that the cortical-NOT-DTN-IO circuitry is important for OKR gain control. However, in the presence of the vestibular ocular reflex, cortical modulation of OKR is marginal, as shown by these authors previously (Liu et al.,). Indeed, the effects of silencing VC/or NOT/DTN on OKR gain is tiny (~2 %? Fig. 3e and Fig. 6d). This subverts the impact of the study.”

The reviewer raised a concern that the “tiny” effects of silencing visual cortex or any downstream pathways subvert the impact of the study. We disagree on the reviewer’s comment, because actually the cortical contribution to OKR, even with intact vestibular ocular reflex, can be moderate under certain conditions and the results from these experiments will provide insight into the circuit mechanisms of cortical contribution to OKR plasticity.

First, in Fig 1g of Liu et al. nature 2016, the cortical contribution to OKR gain was ~11% before vestibular lesion and ~24% after vestibular lesion. The cortical contribution to OKR gain at a suboptimal spatial frequency of 0.04 cpd was even higher (~30% even before vestibular lesion, Extended Data Fig. 3b of the same paper). It is worth noting that those values came from the experiments of silencing the whole visual cortex. However, in Fig. 3e-f of the current study, we silenced only a portion of the visual cortex (either posterior or anterior higher visual areas) and thus expected to have a relatively smaller effect on the OKR gain. Nevertheless, we still got an obvious reduction in OKR gain when silencing the posterior HVAs of one example mouse (5.2-29.1% in Fig. 3e, also see the left figure below; on population average, $8.7 \pm 1.2\%$ in Fig. 3f left and $>10\%$ at suboptimal spatial frequencies in Supplementary Fig. 6b). Similarly, in new Fig. 6b left (old Fig. 6d), cortical silencing reduced the OKR gain of one example mouse by 9.0-31.0% when the NOT-DTN to IO projection was intact (also see the right figure below). And on population average, the visual cortex contributed to 10.6-16.0% of OKR gain (Fig. 6c left, saline and 6c right). Consistently, optogenetically silencing the NOT-DTN to IO pathway reduced the OKR gain by a similar degree ($14.4 \pm 3.6\%$, Fig. 6f left). Therefore, although moderate, the cortical modulation of OKR at the baseline level is noteworthy and significant.

Legend: Left, cortical contribution to OKR gain of the example animal in Fig. 3e left. Right, cortical contribution to OKR gain of the example animal in Fig. 6b left.

Second, in the presence of the vestibular ocular reflex, cortical modulation of OKR was substantially enhanced by continuous OKR stimulation (extended fig 4d in Liu et al. Nature 2016,). Moreover, this paradigm of OKR potentiation also increased the cortical contribution of posterior HVAs, a fraction of the visual cortex, to OKR gain from 7.9% to 13.9% (Fig. 3i).

Third, the moderate level of the cortical contribution to OKR in mice is expected. Even in primates and cats, after the ablation of visual cortex the majority of binocular OKR or monocular OKR evoked by temporo-nasal motion remained (Strong et al., Brain Res 1984; Wood et al., Brain Res 1973; Zee et al., J Neurophysiol 1987;

Dürsteler & Wurtz, *J Neurophysiol* 1988), meaning that the majority portion of the behavior is independent of the visual cortex and instead results from the visual input from the eye. Thus, the retinal input is the driver of the OKR, while the cortical input is the modulator of the OKR. This idea was also raised by the reviewer in comment 5: “which to me might suggest that the PN to IO circuit isn’t so important for establishing the increase in gain that comes with changing spatial frequency (i.e. small effect in Fig. 3e), but is more important for potentiating responses around that set point”. Despite the difference in the quantity of cortical contribution among different animal models, qualitatively speaking the mice and higher mammalian species may share the same circuit principles in modulating innate behaviors. Thus, the study on mouse OKR circuits can benefit the studies on monkey or cat’s OKR circuits.

Last but not the least, although the cortical contribution to OKR at the baseline level is weaker than that after the induction the OKR plasticity, the cortical contributions under both conditions share the same circuit components. **i)** The ablation of the corticofugal projection from the visual cortex to the NOT-DTN not only reduced the cortical contribution to OKR gain at the baseline level, but also prevents the increase in cortical contribution during OKR potentiation (Fig. 3d in Liu et al. *Nature* 2016). **ii)** The amount of cortical impact on NOT-DTN activity explained the cortical contribution to OKR gain before and after the induction of OKR potentiation (Fig. 5c-e in Liu et al. *Nature* 2016). **iii)** Posterior HVAs, which send more projections to the NOT-DTN than the anterior HVAs, contributed more to the cortical modulation of OKR under both baseline and plasticity conditions (**Fig. 3f,i**). **iv)** Silencing the NOT-DTN to IO projection abolished the cortical contribution to OKR gain and the OKR potentiation (**Fig. 6**). These findings demonstrate that the visual cortex→NOT-DTN→IO pathway is shared by the cortical contribution to OKR behavior under both baseline and plasticity conditions. Therefore, studying the circuit mechanisms under baseline condition will provide insights into the mechanisms under plasticity condition.

References not cited in the paper:

Strong et al., *Horizontal Optokinetic Nystagmus in the Cat: Recovery from Cortical Lesions. Brain Res* 315(2):179-92 (1984).

Wood et al., *Direction-specific deficits in horizontal optokinetic nystagmus following removal of visual cortex in the cat. Brain Res* 60(1):231-7 (1973).

Zee et al., *Effects of occipital lobectomy upon eye movements in primate. J Neurophysiol* 58(4):883-907 (1987).

Dürsteler & Wurtz, *Pursuit and optokinetic deficits following chemical lesions of cortical areas MT and MST. J Neurophysiol* 60(3):940-65 (1988).

2. “Given the effects of silencing are small, it is worth performing adequate control experiments. Describing the effects of silencing VC/NOT-DTN when the direction of the grating is reversed is an important control that is missing. With the drifting grating direction reversed, the opposite DTN/NOT should be engaged and unilateral silencing should no longer be effective. Showing the directional dependence of silencing would enable behaviors to be compared in individual mice and would be much more compelling.”

We thank the reviewer for the comment and suggestion. In response, we examined whether the effect of unilateral cortical silencing on OKR depends on the direction of drifting drum gratings. When we silenced the visual cortex in one hemisphere, the OKR gain in response to ipsiversive motion moderately reduced ($27.5 \pm 1.84\%$, **Supplementary Fig. 7d, f left**), which aligns with the anatomy of visual circuits (contralateral retinal input and ipsilateral cortical input, **Supplementary Fig. 7d left**) and the temporo-nasal direction bias of retinal and cortical inputs (**Fig. 1d**). When the direction of drum grating was reversed, this contraversive

motion primarily engaged the opposite NOT-DTN and visual cortex. As predicted, the same unilateral cortical silencing did not reduce the gain of contraversive OKR, but surprisingly it instead increased the gain ($-11.8 \pm 3.34\%$, **Supplementary Fig. 7e, f right**). This effect can be explained by **i**) the same anatomical connectivity of visual circuits, and **ii**) the weak activity of the NOT-DTN-projecting PNs elicited by the contraversive motion (**Supplementary Fig. 7e left**). Therefore, the effect of unilateral cortical silencing on OKR indeed depends on the direction of drifting gratings, and the polarity of this effect is consistent with the connectivity and direction selectivity of NOT-DTN-projecting cortical neurons. We added the results in **page 8, line 23-31** and **page 9, line 1-4**.

3. “The authors nicely demonstrate large related changes in OKR gain and cortical responsivity after 45-minute OKR training. It would be important to show to what extent changes in OKR gain occur when this circuit is silenced at different levels.”

We agree with the reviewer that the contributions of circuit components at different levels to OKR plasticity can provide valuable insight into the mechanisms of such cortical function. First, in Fig. 1e and Extended Fig. 4c of *Liu et al. Nature 2016*, silencing the visual cortex after the induction of OKR potentiation reduced the amount of increase in OKR gain by 47% and 33% for vestibular lesion protocol and continuous OKR stimulation protocol, respectively. These results indicate that the visual cortical activity contributes substantially to the plasticity of OKR behavior. Second, in Fig. 3e of the same study the ablation of NOT-DTN projecting PNs reduced OKR potentiation by $\sim 50\%$, indicating that this corticofugal projection is necessary for a large fraction of OKR potentiation. Third, in the present study we optogenetically silenced posterior HVAs before and after OKR potentiation (**page 8, line 3-13**). We found that even this small fraction of the visual cortex obviously contributed to the OKR potentiation: their cortical contribution almost doubled (from $7.9 \pm 1.9\%$ to $13.9 \pm 1.5\%$, **Fig. 3i**); and silencing these visual areas reduced the OKR potentiation by 34% (**Supplementary Fig. 6e**). Last, in this study we also showed that chemogenetically silencing the NOT-DTN to IO projection prevented the visual cortex from modulating the OKR (**Fig. 6b-c, page 13, line 17-19**) and completely abolished the OKR potentiation (**Fig. 6g-i, page 14, line 1-7**). Therefore, evidence from the above circuit perturbations of the visual cortex, corticofugal projection and NOT-DTN \rightarrow IO projection supports the essential role of the visual cortex \rightarrow NOT-DTN \rightarrow IO pathway in OKR plasticity.

4. “In Figure 4, they show that after OKR potentiation, the cortical neurons are also potentiated, and conclude that: “NOT-DTN projecting PNs are a functionally distinct population in promoting OKR potentiation”. This experiment doesn’t suggest that PNs promote potentiation. It is still possible that the opposite is true, i.e. that there is some mechanism by which the OKR potentiates the PN responses.”

We agree with the reviewer that the experiment in **Fig. 4** itself cannot tell whether NOT-DTN-projecting PNs promote OKR potentiation. However, the results from circuit perturbations collectively can demonstrate the causality. First, silencing the visual cortex substantially reduced the amount of OKR potentiation (Fig. 1f and Extended Data Fig. 4c in *Liu et al. Nature 2016*); even silencing the posterior HVAs, a portion of the visual cortex, lead to smaller OKR potentiation (from $23.5 \pm 6.1\%$ to $15.5 \pm 5.7\%$, **Supplementary Fig. 6e**). Second, more importantly the ablation of NOT-DTN-projecting PNs or silencing their downstream targets (IO-projecting NOT-DTN neurons) severely impaired the OKR potentiation and prevented the cortical contribution to this phenomenon (Fig. 3e in *Liu et al. Nature 2016*, **Fig. 6**). In addition to the circuit perturbations, the plasticity of NOT-DTN-projecting PNs and the changes in NOT-DTN activity together can also explain OKR potentiation. The direction-selective enhancement of corticofugal activity will lead to stronger cortical inputs to the NOT-DTN (**Fig. 7**) and in consequence more robust NOT-DTN activity (Fig 2c,d of *Liu et al. Nature 2016*), which can account for, at least partly, the increased OKR gain in OKR potentiation (Fig. 5c of *Liu et al. Nature 2016*).

Therefore, we think it is a fair speculation that NOT-DTN-projecting PNs are a functionally distinct population in promoting OKR potentiation. We revised the manuscript accordingly (**page 10, line 15-16**).

5. *“In this experiment (Fig. 4) the effect size is larger (Fig. 4d), which to me might suggest that the PN to IO circuit isn’t so important for establishing the increase in gain that comes with changing spatial frequency (i.e. small effect in Fig. 3e), but is more important for potentiating responses around that set point.”*

We thank the reviewer for raising this good point and agree to the reviewer’s comment on the functional role of the visual cortex→NOT-DTN→IO pathway. Under the baseline condition (without OKR plasticity), the responsiveness of NOT-DTN-projecting PNs was low (**Fig. 4**) and silencing the visual cortex had small effects on the OKR gain and NOT-DTN activity (**Fig. 3e-f, 6b-c**, Fig. 1d,g,4f in *Liu et al. Nature 2016*). These results suggest that the bottom-up retinal input is the driver of the OKR behavior which determines the basic level of OKR including its tuning curves (set point). Under the plasticity condition, the activity of NOT-DTN-projecting PNs was boosted (**Fig. 4d,e, Supplementary Fig. 8c**) and consistently silencing the visual cortex now reduced OKR gain and NOT-DTN activity much more strongly while largely preserving the OKR behavior and its turning (**Fig. 3i**, Fig. 1d,g,4f in *Liu et al. Nature 2016*). Thus, the top-down cortical input is the modulator which potentiates OKR responses around the set point. We have discussed this in **page 18, line 9-18**.

6. *“It would be helpful if the authors clearly state what are the significant advances being made in this study. How is the top-down modulation of the DTN-NOT different from what was shown earlier by Liu et al., 2016? Given that cortical control of OKR gain is weak with the vestibular system intact, the authors should discuss what the major contributor to the OKR gain is.”*

We thank the reviewer for the suggestions. One major contribution of the study of *Liu et al., 2016* is the demonstration that visual corticofugal neurons boost on the activity of NOT-DTN neurons to potentiate OKR gain. Although this previous study pinpointed the corticofugal projections and the enhanced NOT-DTN activity as the mediators of OKR potentiation, it did not examine what mechanisms could lead to enhanced cortical drive of NOT-DTN neurons. Indeed, there are at least 3 possible mechanisms: i) an increase in the firing of NOT-DTN-projecting corticofugal neurons, ii) a strengthening of the synapse between corticofugal axons and NOT-DTN neurons or iii) a decrease in feed-forward inhibition received by NOT-DTN neurons. Thus, a detailed investigation of the circuit and cellular mechanisms underlying the cortical contribution to OKR plasticity was still missing. In the present study, we examined the first possibility and found that the NOT-DTN-projecting PNs became more responsive, especially in the population preferring the temporo-nasal direction. In addition to the corticofugal plasticity, understanding the cortical contribution to OKR potentiation also requires the knowledge of the nature of the signals flowing through the corticofugal pathway and the connectivity of the corticofugal-brainstem circuits, which had neither been fully addressed by previous studies, including *Liu et al., 2016*. We found that the corticofugal projection synapsed on the NOT-DTN neurons projecting to the IO and through this disynaptic pathway the signals encoding temporo-nasal motion were conducted to the IO. In comparison to the study of *Liu et al., 2016*, the novelty of the present study is manifested in three aspects. First, the plasticity of NOT-DTN-projecting PNs now explains for the first time how the visual cortex enhances the NOT-DTN activity and in turn potentiates OKR gain (**page 17, line 10-21**). Second, the temporo-nasal bias and direction selective plasticity in NOT-DTN-projecting PNs highlight the exquisite functional specificity of cortico-brainstem projections and the importance of the behaviorally relevant corticofugal signals in the cortical modulation of innate behaviors (**page 16, line 10-23; page 17, line 10-31; page 18, line 1-8**). Finally, the discovery of the disynaptic pathway connecting the visual cortex to the IO provides a new perspective on the sources of retinal slips, which are essential for the cerebellum-dependent OKR plasticity, and suggests that the visual cortex may also have an impact on the cerebellum-dependent OKR potentiation (**page 17, line 10-**

31; page 18, line 1-8). We revised the introduction and results to clarify the questions which had not been completely answered before and we are trying to tackle in this study (**page2, line 30-32; page 3, line 1-4, line 17-20; page 9, line 16).**

Please see our response to the relatively weak cortical control of OKR gain with intact vestibular system in comment 1. Please see our discussion of what the driver and modulator of OKR behavior are in comment 5.

Reviewer #3 (Remarks to the Author):

In their study, Liu et al. describe the organization and function of a direction-selective cortico-brainstem pathway in mice. The authors find that corticofugal layer 5 pyramidal neurons (L5 PN) labeled retrogradely from NOT and DTN, two brainstem nuclei that prefer temporo-nasal (TN) visual motion and control horizontal gaze-stabilizing eye movements via the optokinetic reflex (OKR), share the direction preference and strong selectivity of their targets. Liu et al. show that L5 PNs providing input to NOT/DTN are enriched in anterior primary visual cortex (V1) and posterior higher visual areas (HVAs). Consistent with this enrichment, they demonstrate that optogenetic silencing of posterior HVAs reduces the OKR gain more than optogenetic silencing of anterior HVAs. Next, Liu et al. discover that the OKR potentiation observed after prolonged visual stimulation is correlated with an increase in the responses of NOT/DTN-projecting L5 PNs, particularly those preferring TN motion. Neurons in NOT/DTN project to several downstream targets. Liu et al. show that the DTN/NOT neurons that receive input from L5 PNs selectively innervate one of these, the inferior olive (IO). All IO-projecting NOT/DTN neurons prefer TN motion. Liu et al. pharmacogenetically suppress the IO-projection NOT/DTN neurons, which cancels the impact of optogenetically silencing visual cortex on the OKR gain. Together these findings indicate that visual cortex modulates the OKR gain through a direction-selective (TN) pathway from visual cortex via NOT/DTN to IO. Finally, the authors use a computational model to explore how directional bias and response enhancement of L5 PNs may drive OKR potentiation through the downstream pathway.

Overall, this exciting study reveals the organization and function of a direction-selective pathway from visual cortex via NOT/DTN to IO that modulates a highly conserved gaze-stabilizing reflex. The authors combine an impressive array of sophisticated techniques, and the quality of the data and their analysis is exceptional. The results are presented clearly in the figures and the text. Addressing the following specific comments would help to clarify a few important points.

Specific comments

1. *“Do retinal ganglion cells and L5 PNs converge onto IO-projecting NOT/DTN neurons? The discussion suggests they do. Has this been shown in mice? If so, what is the spatial and cell-type distribution of the retinal ganglion cells, and does it align with the retinotopic position of the L5 PNs in the anterior V1? A retrograde tracing strategy could address these questions (e.g., injecting a retrogradely transported helper AAV into IO followed by injection of a rabies virus into NOT/DTN).”*

We thank the reviewer for bringing out this important question. The convergence of retinal and cortical inputs onto IO-projecting NOT-DTN neurons hadn't been shown in mice before. Following reviewer's suggestion, we carried out rabies transsynaptic tracing to examine presynaptic neurons innervating the IO-projecting NOT-DTN neurons (starter cells) (**Fig. 5j, Supplementary Fig. 9e**). We uncovered transsynaptically labeled RGCs in contralateral retina (named as NOT-DTN-innervating RGCs) and PNs in L5 of ipsilateral visual cortex (named as NOT-DTN-innervating PNs, **Supplementary Fig. 9f,i, page 11, line 14-32; page 12, line 1-4**), which indicates

that RGCs and L5 PNs do converge onto IO-projecting NOT-DTN population.

We also examined the spatial distribution of NOT-DTN-innervating RGCs and PNs and the dendritic morphology of these RGCs. First, the NOT-DTN-innervating PNs showed a similar distribution across the visual cortex to the NOT-DTN-projecting cortical neurons: they were more densely packed in anterior V1 and posterior HVAs (**Supplementary Fig. 9g,h**). Second, the NOT-DTN-innervating RGCs were clearly more populated in the superior portion of the retina than the inferior portion (**Supplementary Fig. 9i,j**). This distribution pattern retinotopically aligns with the enrichment of the NOT-DTN-innervating PNs in the anterior V1, since both superior retina and anterior V1 monitor the lower visual space. Last, by analyzing dendritic stratification in ON and OFF sublamina layers of the IPL, we found that the NOT-DTN-innervating RGCs consisted of both ON and ON-OFF types of RGCs (**Supplementary Fig. 9k-m**), which is consistent with a previous report that both ON and ON-OFF direction selective RGCs project to the NOT-DTN (*Dhande et al., J Neurosci 2013*). Please see details in **page 11, line 19-32; page 12, line 1-4**.

2. “The authors show that pharmacogenetic silencing of IO-projecting NOT/DTN neurons suppresses the impact of cortical silencing on the OKR gain. How does silencing the IO-projecting NOT/DTN neurons alone affect the OKR gain and its stimulus-drive plasticity?”

This question was also asked by reviewer 1. To answer this question, we took an optogenetic approach to silence the IO-projecting NOT-DTN neurons, which allows switching the circuit on and off alternately in a trial-by-trial basis. The rapidity and reversibility of optogenetic silencing control for trial to trial variability and, most importantly, for potential drifts in OKR gain during long recording sessions, and thus this method can reveal relatively small changes in OKR gain. We conditionally expressed inhibitory opsin ArchT in IO-projecting NOT-DTN neurons in the left hemisphere. Optogenetically silencing this projection-specific population reduced the gain of OKR driven by the ipsiversive/temporo-nasal motion, but not by the contraversive/naso-temporal motion ($14.4 \pm 3.6\%$ vs $-0.07 \pm 1.5\%$, **Fig. 6f**), which is consistent with the effects of unilateral cortical silencing (**Supplementary Fig. 7d-f**). In addition, we chemogenetically silenced the IO-projecting NOT-DTN neurons during continuous OKR stimulation. This circuit perturbation prevented the OKR potentiation. Overall, these results suggest that the NOT-DTN to IO projection plays an essential role in OKR plasticity by adjusting the strength of OKR according to behavioral needs and visual experience. We have added the results of these two experiments to **page 13, line 23-27** and **page 14, line 1-7**.

3. “The authors show that OKR plasticity is accompanied by enhanced responses in TN-preferring DTN/NOT-projection L5 PNs. They show that both phenomena require patterned visual stimulation (i.e., a horizontally oscillating grating vs. a gray screen). Are both phenomena motion axis selective, or do vertically oscillating gratings have the same effects, does the stimulus selectivity of the two phenomena diverge for vertically oscillating gratings?”

Following the reviewer’s suggestion, we have done a new control experiment to address whether the OKR potentiation and the cortical plasticity depend on the direction of grating motion. We presented a horizontal drum grating oscillating vertically to head-fixed mice continuously for 45 min. This visual exposure did not increase the gain of horizontal OKR, neither the activity of NOT-DTN-projecting cortical neurons in response to temporo-nasal motion (**Supplementary Fig. 8d-f; page 10, line 9-11**). These results indicate that both the OKR potentiation and the cortical plasticity depend on the direction of visual motion which induces OKR potentiation.

REVIEWERS' COMMENTS

Reviewer #1 (Remarks to the Author):

I am impressed by the rigor and quality with which the authors have responded to my comments. I have only minor points to be clarified. I list them point by point following the replies given by the authors

4 Page 3 line 2-4 this may be true for the mouse but not as a general statement. Thus it remains an open question in the mouse

8 Page 3 line 28: you could add to following OKR potentiation already here by a drum grating oscillating bidirectionally along the azimuth to avoid any misunderstanding like mine
11 + 20 page 17 line 8: It is a common assumption that the OKR and VOR function as a push pull system. This would work without a suppressive effect from cortex on contraversive OKR. Reducing the drive in one direction(i. e. ipsiversive) will automatically enhance the opposite direction.

12 I am impressed by the additional information.

15 least squares is the non-optimal fit because cortical plasticity index as well as OKR potentiation are variable. So you should apply 2 dimensional regression. You can omit least squares.

16 very nice

17+18 Inhibitory NOT cells do not only project to the SC. It was shown in cats that so called jerk cells in the NOT (responding to saccade like image displacements) are GABAergic and project to the LGN. This may be of interest for you ms in preparation. Wahle et al. 1994 EJM 6 454-460

19 if the NOT – IO projection plays only a modulatory and pivotal role in OKR plasticity which NOT ipsiversive output drives OKN ?? You showed that NPH projections are sparse (probably feeding into the integrator). Since abolishing NOT ipsiversive activity knocks out ipsiversive OKN there must be a prominent projection (direct or indirect) onto the horizontal oculomotor nuclei. This information would be much appreciated by the oculomotor community.

Reviewer #2 (Remarks to the Author):

I have no further comments and I congratulate the authors for doing a commendable job in addressing previous critiques.

Reviewer #3 (Remarks to the Author):

The authors have added an impressive set of new experiments to improve an already strong manuscript. I have no further significant comments or concerns. Congratulations on an insightful study!

Here are our replies to reviewers' comments.

Reviewer #1 (Remarks to the Author):

I am impressed by the rigor and quality with which the authors have responded to my comments. I have only minor points to be clarified. I list them point by point following the replies given by the authors

1. *"4 Page 3 line 2-4 this may be true for the mouse but not as a general statement. Thus it remains an open question in the mouse"*

We thank the reviewer for the comment and suggestion. We have modified the statement accordingly. Please see the revision in **line 57-58**.

2. *"8 Page 3 line line 28: you could add to following OKR potentiation already here by a drum grating oscillating bidirectionally along the azimuth to avoid any misunderstanding like mine"*

We have added the details of the OKR potentiation protocol as the reviewer suggested. Please see the change in **line 83-84**.

3. *"11 + 20 page 17 line 8: It is a common assumption that the OKR and VOR function as a push pull system. This would work without a suppressive effect from cortex on contraversive OKR. Reducing the drive in one direction(i. e. ipsiversive) will automatically enhance the opposite direction."*

We thank the reviewer for providing the insight of the how the OKR circuits work in general. In the push-pull mechanism, the OKR is driven by the activity of operating NOT-DTN (ipsiversive, push), but suppressed by the activity of non-operating NOT-DTN (contraversive, pull); in other words, OKR gain is determined by the activity difference between the two NOT-DTNs (*Distler & Hoffmann, J Neurosci 2011*). Our data suggest that the visual cortex influences the OKR actually by modulating the push-pull system. This cortical area can facilitate the ipsiversive OKR by innervating the NOT-DTN mediating the OKR behavior (push, **Supplementary figure 7d**), but suppress the contraversive OKR by innervating the non-operating NOT-DTN (pull, **Supplementary figure 7e**). Thus, the top-down input from the visual cortex utilizes the existing push-pull circuit to achieve the bidirectional modulation of OKR. We revised the manuscript accordingly (**line 484-489**).

4. *"12 I am impressed by the additional information."*

We thank the reviewer for the compliment.

5. *"15 least squares is the non-optimal fit because cortical plasticity index as well as OKR potentiation are variable. So you should apply 2 dimensional regression. You can omit least squares."*

We removed the least squares regression as the reviewer suggested.

6. *"16 very nice"*

We thank the reviewer for the compliment.

7. *"17+18 Inhibitory NOT cells do not only project to the SC. It was shown in cats that so called jerk cells in the NOT (responding to saccade like image displacements) are GABAergic and project to the LGN. This may be of interest for you ms in preparation. Wahle et al. 1994 EJM 6 454-460"*

The reviewer is right that the inhibitory NOT neurons in mice project not only to the SC but also to the LGN, as well as several other known or unknown targets (unpublished data). The anatomy and function of those

inhibitory projections will be covered by another manuscript in preparation. We really appreciate the reviewer's courtesy of suggesting a reference for that study.

8. "19 if the NOT – IO projection plays only a modulatory and pivotal role in OKR plasticity which NOT ipsiversive output drives OKN ?? You showed that NPH projections are sparse (probably feeding into the integrator). Since abolishing NOT ipsiversive activity knocks out ipsiversive OKN there must be a prominent projection (direct or indirect) onto the horizontal oculomotor nuclei. This information would be much appreciated by the oculomotor community."

We agree with the reviewer that considering the indispensable role of NOT-DTN in driving the OKR and the modulatory role of NOT-DTN to IO projection, there must be other NOT-DTN projection pathways mediating the OKR. In addition to the IO, the NOT-DTN neurons also project to many places in the midbrain, pons and medulla, including pre-oculomotor structures such as the nucleus prepositus hypoglossi (NPH), the nucleus reticularis tegmenti pontis (NRTP) and the medial vestibular nucleus (MVN), which innervate directly or indirectly the horizontal oculomotor nuclei (*Kato et al. Brain Res 1995; Gamlin, Prog Brain Res 2006*). Even though the direct experimental evidence to answer which of these projections mediate the OKR lacks, the NOT-DTN projections to those pre-oculomotor structures are likely to drive the OKR behavior. We revised the manuscript accordingly (**line 527-530**).

Reviewer #2 (Remarks to the Author):

I have no further comments and I congratulate the authors for doing a commendable job in addressing previous critiques.

We thank the reviewer again for the comments and suggestions, which certainly improved the quality of this study.

Reviewer #3 (Remarks to the Author):

The authors have added an impressive set of new experiments to improve an already strong manuscript. I have no further significant comments or concerns. Congratulations on an insightful study!

We thank the reviewer again for the comments and suggestions, which certainly improved the quality of this study.